# Engineering immunomodulatory and osteoinductive implant surfaces via mussel adhesion-mediated ion coordination and molecular clicking

Tao Wang[1,2,3,5], Jiaxiang Bai [4,5], Min Lu[1], Chenglong Huang[3], Dechun Geng[4], Gang Chen[3], Lei Wang[1], Jin Qi[1✉], Wenguo Cui [1✉] & Lianfu Deng[1✉]

Immune response and new tissue formation are important aspects of tissue repair. However, only a single aspect is generally considered in previous biomedical interventions, and the synergistic effect is unclear. Here, a dual-effect coating with immobilized immunomodulatory metal ions (e.g., $Zn^{2+}$) and osteoinductive growth factors (e.g., BMP-2 peptide) is designed via mussel adhesion-mediated ion coordination and molecular clicking strategy. Compared to the bare $TiO_2$ group, $Zn^{2+}$ can increase M2 macrophage recruitment by up to 92.5% in vivo and upregulate the expression of M2 cytokine IL-10 by 84.5%; while the dual-effect of $Zn^{2+}$ and BMP-2 peptide can increase M2 macrophages recruitment by up to 124.7% in vivo and upregulate the expression of M2 cytokine IL-10 by 171%. These benefits eventually significantly enhance bone-implant mechanical fixation (203.3 N) and new bone ingrowth (82.1%) compared to the bare $TiO_2$ (98.6 N and 45.1%, respectively). Taken together, the dual-effect coating can be utilized to synergistically modulate the osteoimmune microenvironment at the bone-implant interface, enhancing bone regeneration for successful implantation.

[1] Department of Orthopaedics, Shanghai Key Laboratory for Prevention and Treatment of Bone and Joint Diseases, Shanghai Institute of Traumatology and Orthopaedics, Ruijin Hospital, Shanghai Jiao Tong University School of Medicine, 197 Ruijin 2nd Road, 200025 Shanghai, P. R. China. [2] Department of Orthopaedics, Shanghai General Hospital, Shanghai Jiao Tong University School of Medicine, 85 Wujin Road, 200080 Shanghai, P. R. China. [3] Jiaxing Key Laboratory of Basic Research and Clinical Translation on Orthopedic Biomaterials, Department of Orthopaedics, The second Affiliated Hospital of Jiaxing University, 1518 North Huancheng Road, 314000 Jiaxing, P. R. China. [4] Department of Orthopaedics, The First Affiliated Hospital of Soochow University, 188 Shizi Street, Suzhou 215006 Jiangsu, P. R. China. [5] These authors contributed equally: Tao Wang, Jiaxiang Bai. ✉email: jinjin838@hotmail.com; wgcui80@hotmail.com; lf_deng@126.com

Bone implantation using nonliving materials (e.g., metals, polymers, and ceramics) as bone substitutes has proven to be an efficient clinical method for bone fracture fixation, joint arthroplasty, spinal reconstruction, and so on[1–3]. Clinically, the success rate of integration between implants and bone tissues exceeds 95%[4,5], while the failure rate of early implantation was about 1.2% due to the development of fibrous tissue between the implants and the surrounding bone tissues in the healing period[6]. The general problem in these exogenous biomaterials is their bio-inertness, lacking in bioactivities to completely adapt to the complex physiological bone regeneration process. Tissue regeneration involves three indispensable stages: (i) immune action, (ii) cell proliferation and new tissue formation, and (iii) remodeling and maturation[7,8]. The bio-inertness of exogenous biomaterials thus is inclined to lead to fibrotic encapsulation, implant loosening, or implantation failure. Traditional studies regarding bone implants predominantly focused on optimizing the osteogenic capacity, with some inert bone implants designed to evade immune response and others introduced with various bioactive moieties (e.g., peptides, growth factors, protein, and even ions) for promoting osteogenesis in vitro and bone-to-implant osseointegration[9–11]. However, these implants may not completely adapt to the in vivo microenvironment, thus leading to some inconsistent results in vivo[12] majorly due to the uncontrolled local immune responses triggered by exogenous biomaterials. Thus in order to develop implants with efficient bone regeneration, it is inadequate to merely emphasize the direct osteogenesis while ignoring the local immune microenvironment. To design an ideal bone implant, the synergy of direct osteogenicity and immunomodulatory function should be considered to precisely match the mechanisms of bone regeneration process and achieve satisfactory osteogenesis at the bone-to-implant interfaces.

Recent studies on osteoimmunology further revealed that immune microenvironments also play an important role in bone tissue formation[7,13,14]. In different microenvironments, macrophages polarize into classically activated macrophages (M1) or alternatively activated macrophages (M2). The pro-inflammatory M1 macrophages activate inflammation and promote fibrosis while the pro-healing M2 macrophages coordinate tissue healing processes by activating stem/progenitor cells and remodeling extracellular matrix for regeneration[15,16]. As is found in previous studies, an efficient and timely switch from M1 to M2 macrophage phenotype was essential for bone healing and osteointegration around bone implants, creating a favorable osteoimmune environment via the increased production of anti-inflammatory (e.g., IL-10) and pro-osteogenic (e.g., BMP-2 and VEGF) cytokines[17]. To this end, various efforts on surface bioengineering of the bone implants, such as optimizing surface physical properties[18–21] (e.g., topography, wettability, charge, etc.) and introducing cytokines[22–24] (e.g., interleukin-4 (IL-4), lipoxin A4 (LXA4), etc.) or active metal ions[25–28] (e.g., $Ca^{2+}$, $Zn^{2+}$, $Sr^{2+}$, etc.), have been made to regulate the local immune microenvironment and ameliorate the final outcomes of bone regeneration and osseointegration around bone implants in vivo.

Surface bioengineering possesses remarkable superiority in introducing osteoinductive and immunomodulatory activities onto a bone implant because it allows surface modification with various bioactivities for diversified requirements[29–32]. Multi-functional bone implants capable of co-regulating stem/progenitor cells and immune cells could conceivably be readily obtained by surface co-modification of relevant biological signals, which may construct a benign microenvironment suitable for complete bone regeneration. To date, various physical or chemical means have been vigorously developed for surface bioengineering. For instance, layer-by-layer assembly, Langmuir-

Blodgett deposition silanization, anodization, acid etching, and ion doping were widely used to add implant surfaces with different bioactive moieties (e.g., peptides, proteins, and even ions)[33–36]. Since physical methods have persistent drawbacks of serious molecular leakage and the lack of long-term activity, current surface bioengineering strategies for bone implants mainly rely on chemical conjugations[37,38]. Traditional chemical methods, however, mostly involve tedious chemical reactions as well as complicated surface treatment technologies[1,39]. Apart from potential damage to the bioactive molecules, the complex procedures of these methods also hinder their application in multi-modification due to the low controllability and poor operability. Therefore, simple and biocompatible surface approaches capable of efficiently conjugating multiple bioactivities are highly desired particularly for the design of dual-functional bone implants with both osteoinductive and immunomodulatory functions to match the mechanism of bone regeneration.

Biomimetic strategies have emerged as a promising approach for the multi-modification. In 2007, Lee et al. developed a novel surface chemistry via dopamine polymerization[40]. The method was inspired by the molecular adhesion mechanism of marine mussel foot proteins (Mfps), in which repetitive catecholic amino acids (3,4-dihydroxy-L-phenylalanine, DOPA) contribute to strong surface adhesion[41]. Likewise, catechol-rich poly(-dopamine) can also achieve robust molecular adhesion to virtually all kinds of substrates[42]. In addition, the catechol residues on the surface enable not only simple conjugations of amino- or thiol-containing biomolecules via Michael addition or Schiff base reaction but also spontaneous coordination with bioactive metal ions[43–45]. These advantages indicate the mussel-inspired surface strategy has the potential to co-modify bone implants with osteoinductive biomolecules and immunomodulatory active ions. However, the critical problem of this strategy is the random consumption of active groups (e.g., amino and thiol), which would interfere with the functions of conjugated biomolecules[38,46]. Recently, Pan et al.[1] designed two mussel-derived biomimetic peptides for simple biomodification of Ti implants through robust catechol/titanium dioxide ($TiO_2$) coordinative interactions. The highly biomimetic peptides capped with RGD- or OGP-derived sequences could improve not only the biocompatibility of Ti implants but also the efficiency of osteogenicity, osseointegration, and mechanical stability in vivo. The strategy provides a clear chemical binding on implants surfaces yet the uncontrolled biomolecular conjugation, particularly for multi-modification, which is not conductive to the reproducibility of a multi-bioactive surface. Given this, we designed an improved biomimetic strategy by combining mussel-like adhesion with bioorthogonal click reaction, a specific and biocompatible chemistry[47–49]. We hypothesize this strategy which involves a bioclickable way for biomolecular conjugation and a coordination means for ion loading would provide a promising solution for surface engineering of osteoinductive and immunomodulatory bone implants.

In this work, we chose medical titanium (Ti) screw as implant model since Ti materials are widely used in orthopedic and dental surgery. To integrate the mussel adhesion mechanism with bioorthogonal click chemistry, we first synthesize a DOPA-containing peptide with bioclickable group dibenzylcyclooctyne (DBCO)[50] (Fig. 1A). Next, the clickable mussel-derived peptide is stably bound onto Ti screws via metal-catechol coordination. Then, an immunoactive metal zinc ion ($Zn^{2+}$) capable of polarizing macrophages to the anti-inflammatory M2 phenotype is coordinated with the catechol residues to generate an immunoactive surface. For direct osteogenicity, a BMP-2-derived synthetic peptide capped with azido group is synthesized and

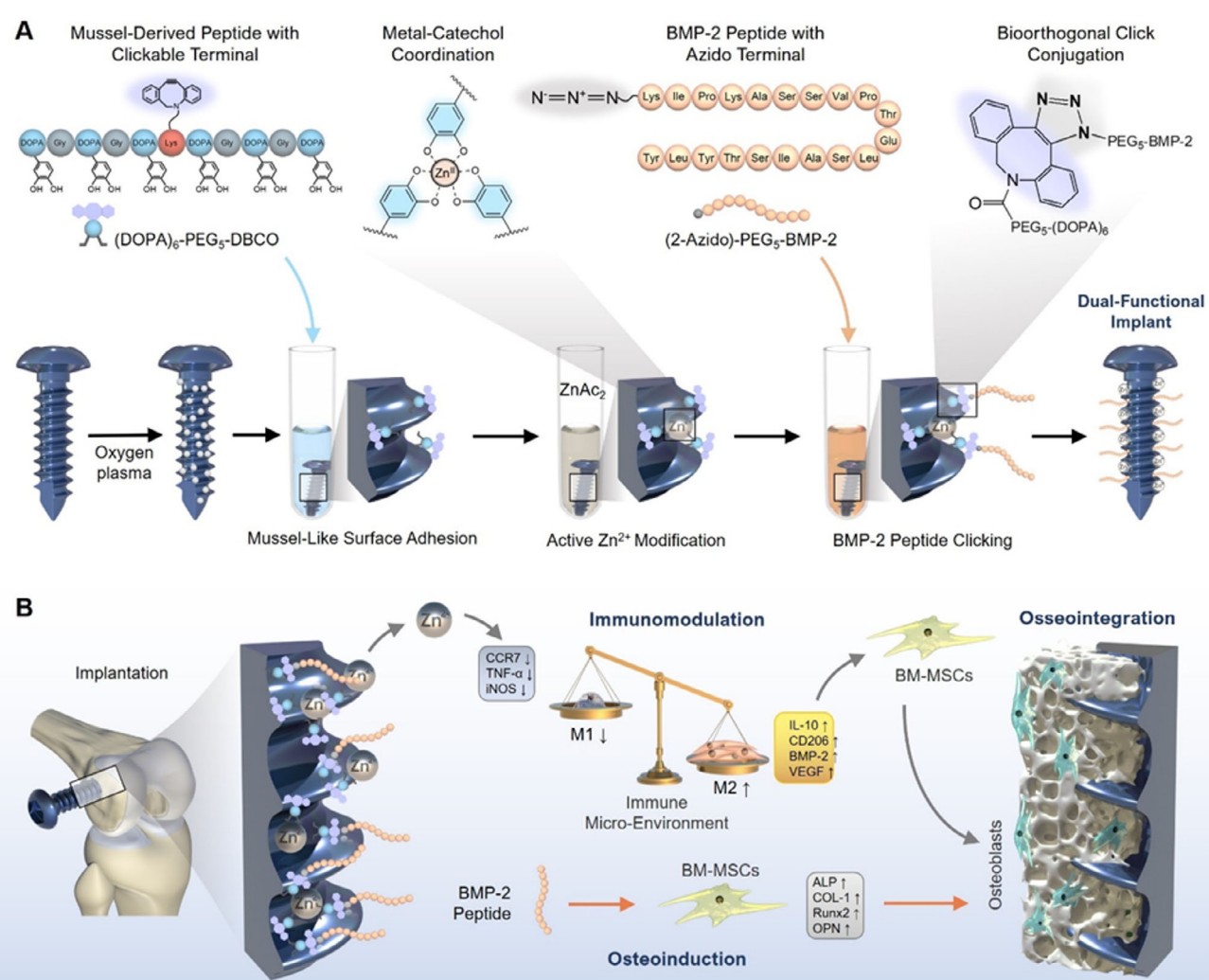

**Fig. 1 Design strategy for engineering immunomodulatory and osteoinductive implant surfaces. A** Schematic illustration of the mussel-derived peptide for ion coordination and biomolecular click conjugation on a medical Ti screw. **B** In a bone implant model, the $Zn^{2+}$ and BMP-2 peptide co-modified Ti screw shows osteoinductive and immunomodulatory dual functions in vivo, synergistically enhancing the interfacial osteogenesis and the intra-bone implant integration after implantation.

conjugated with surface DBCO groups via a bioorthogonal cycloaddition chemistry. After successful synthesizing the implants, the efficiency of mussel adhesion-mediated ion coordination and molecular clicking between $Zn^{2+}$ modification and BMP-2 peptide conjugation is evaluated, respectively. Then, the synergetic effect of immunoactive $Zn^{2+}$ and osteoinductive BMP-2 peptide on macrophage polarization and interfacial osteogenesis in vitro and in vivo are investigated and discussed. Overall, this study elucidates the synergy of direct osteogenicity and immunomodulatory function on osteogenesis. It may also provide a simple and efficient solution for the engineering of osteoinductive and immunomodulatory implants to precisely adapt to the favorable microenvironment in vivo for bone regeneration.

## Results and discussion

**Mussel-derived peptide synthesis and surface modification.** The mussel-derived peptide with clickable DBCO group was synthesized by solid-phase peptide synthesis strategy with a Mfps-like peptide mimicking method[39,47]. A commercially available acetonide-protected Fmoc-DOPA (acetone)-OH was used to introduce catecholic amino acid DOPA into the peptide sequence. To preserve enough catechol groups for $Zn^{2+}$ coordination and

accessible DBCO groups for biomolecular clicking after peptide adhesion on Ti surfaces, hexavalent DOPA units with one amino acid interval and DBCO with a long polyethylene glycol (PEG) chain were used to prepare the mussel-derived peptide Ac-(DOPA)-G-(DOPA)-G-(DOPA)-K[(PEG$_5$)-(Mpa)-(Mal-DBCO)]-(DOPA)-G-(DOPA)-G-(DOPA) ((DOPA)$_6$-PEG$_5$-DBCO) (Fig. 2A and Supplementary Fig. 1). In addition, a peptide (KIPKASSVPTELSAISTLYL) derived from the 73 to 92 amino acid fragment of the BMP-2 finger epitope was conjugated with two-azidoacetic acid and PEG$_5$-carboxyl coupling, respectively (Fig. 2B and Supplementary Fig. 1). The obtained azido-capped BMP-2-derived peptide (Azido-KIPKASSVPTELSAISTLYL, (2-Azido)-PEG$_5$-BMP-2) thus could be easily connected with (DOPA)$_6$-PEG$_5$-DBCO-bound surfaces, providing a flexible surface modification strategy. The two synthesized peptides were first purified by high-performance liquid chromatography (HPLC, purity > 95%) (Supplementary Fig. 2). Electrospray ionization mass spectrometry (ESI-MS) was further used to confirm their molecular structures. The monoisotopic mass $[M + 2H]^{2+}$ of (DOPA)$_6$-PEG$_5$-DBCO and $[M + 2H]^{2+}$ of (2-Azido)-PEG$_5$-BMP-2 were found at 1036.69 Da and 1247.63 Da, which corresponded to the theoretical molecular weight at 2070.18 Da and 2492.86 Da, respectively (Fig. 2C, D). It is worth

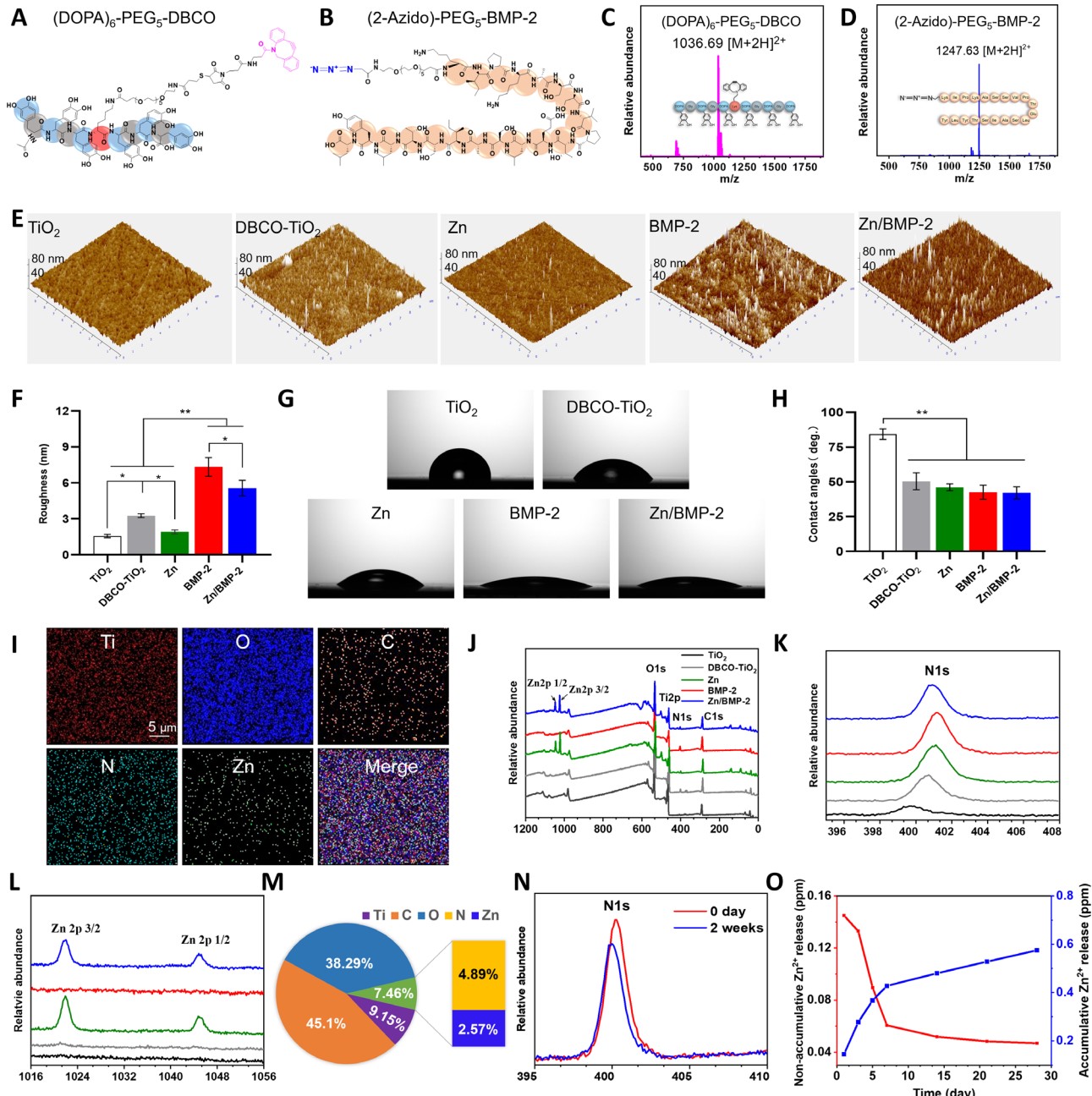

**Fig. 2 Characterizations of clickable peptide and Zn$^{2+}$ and BMP-2 peptide co-modified surface. A, B** The molecular structures of (DOPA)$_6$-PEG$_5$-DBCO and (2-Azido)-PEG$_5$-BMP-2. **C, D** ESI-MS spectra of the two synthetic peptides. **E** AFM images of different surfaces and (**F**) quantitation of the suface roughness with the different modified surfaces ($n = 3$ independent samples per group, by a one-way ANOVA with a Tukey's post hoc test for multiple comparisons). Data are reported as mean ± SD, *$p < 0.05$, **$p < 0.01$. **G** The water contact angles of different surfaces and (**H**) the quantitative results ($n = 3$ independent samples per group, by a one-way ANOVA with a Tukey's post hoc test for multiple comparisons. Data are reported as mean ± SD, **$p < 0.01$). **I** SEM-EDS elemental mapping for the Zn$^{2+}$ and BMP-2 peptide co-modified surface (Zn/BMP-2) (scar bar = 5 μm, three independent experiments). **J–L** XPS analysis of the bare and modified TiO$_2$ surface (DBCO-TiO$_2$, Zn, BMP-2 and Zn/BMP-2. **M** Quantitative elemental analysis according to XPS. **N** Changes of N 1 s signal in the XPS spectrum of the Zn/BMP-2 surface after incubated in DMEM for 2 weeks. **O** Zn$^{2+}$ release profiles of the Zn/BMP-$^2$ surface in PBS solution; red (left) and blue (right) represent the non-accumulative and accumulative Zn$^{2+}$ release, respectively. Exact $P$ values were given in the Source Data file.

mentioning that the azido-capped osteogenic peptide (2-Azido)-PEG$_5$-BMP-2 still showed excellent osteogenic activity (Supplementary Fig. 3). Incubating bone marrow mesenchymal stem cells (BM-MSCs) with (two-Azido)-PEG$_5$-BMP-2 peptide could elicit potent alkaline phosphates (ALP) activity and matrix mineralization, demonstrating the efficient osteoinductive ability of this molecularly modified BMP-2 peptide. These results indicated the

successful preparation of the clickable mussel-derived peptide and azido-capped osteogenic biomolecule.

With the two peptides, surface coating on Ti-based substrates was then performed to prepare a Zn$^{2+}$ and BMP-2 peptide co-modified surface. TiO$_2$-deposited quartz substrate (noted as TiO$_2$), the control group in the following experiments, was used to mimic the surface of medical Ti biomaterials in all in vitro

investigations. (DOPA)$_6$-PEG$_5$-DBCO-coated substrates (noted as DBCO-TiO$_2$) were prepared by incubating the TiO$_2$-deposited quartz in phosphate buffer saline solution (PBS, 0.02 mM, pH = 7.2) with the mussel-derived peptide. The catechol residues on peptide-coated substrates were then coordinated with Zn$^{2+}$ after immersion into zinc acetate (ZnAc$_2$) solution to obtain Zn$^{2+}$-loaded surface (noted as Zn). Finally, (2-Azido)-PEG$_5$-BMP-2 was conjugated through bioorthogonal click chemistry to prepare a Zn$^{2+}$ and BMP-2 peptide co-modified surface (noted as Zn/BMP-2). A BMP-2-modified surface without loading Zn$^{2+}$ (noted as BMP-2) was also prepared as a control. Quartz-crystal microbalance (QCM) was used to monitor the peptide grafting densities of (DOPA)$_6$-PEG$_5$-DBCO and (2-Azido)-PEG$_5$-BMP-2[51]. As shown in Supplementary Fig. 4A, (DOPA)$_6$-PEG$_5$-DBCO could be steady bound onto the QCM chips and the maximal grafting density was about 489 ng/cm$^2$, indicating the high efficiency and spontaneous adhesion onto TiO$_2$-deposited quartz substrate surface. The grafting density in our study was higher than that of Pan's work (363 ng/cm$^2$)[1], mainly due to the improved binding affinity of mussel-adhesion peptide resulting from the increased number of catechol groups. Then, the DBCO-modified TiO$_2$ substrates were incubated with azido-capped BMP-2-derived peptides for bioorthogonal conjugation (Supplementary Fig. 4B). The click reaction started in a few minutes, and the maximal grafting density for (2-Azido)-PEG$_5$-BMP-2 was 140 ng/cm$^2$, which was comparable to the results in previous reports on the immobilization of BMP-2 on chitosan-grafted Ti surfaces (50 ng/cm$^2$)[52] and the polydopamine-coated nanofibers (124 ng/cm$^2$)[53], respectively. The changes of surface roughness after peptide modification were first checked by atom force microscope (AFM) (Fig. 2E). A clear change was observed after (DOPA)$_6$-PEG$_5$-DBCO peptide modification. Although the Zn$^{2+}$-modification led to a negligible change, the second peptide modification with (2-Azido)-PEG$_5$-BMP-2 resulted in a significant increase in surface roughness. Quantitative analysis confirmed the results (Fig. 2F), indicating the efficiency of peptide modification through mussel-like adhesion and subsequent peptide clicking. Likewise, the surface wettability was significantly improved after Zn$^{2+}$ or BMP-2 peptide modification (Fig. 2G, H), probably due to the hydrophilicity of surface chelated Zn$^{2+}$ and the amino acid sequence of BMP-2 peptide. The successful Zn$^{2+}$-modification was confirmed by energy-dispersive X-ray spectrometry (EDS) elemental mapping and X-ray photoelectron spectroscopy (XPS). Although scanning electron microscopy (SEM) analysis showed no significant differences on different surfaces after co-modification (Supplementary Fig. 5), EDS elemental mapping revealed homogeneously distributed zinc on the surface after treatment with Zn$^{2+}$ (Fig. 2I). Surface elemental compositions were further determined by XPS to further confirm Zn$^{2+}$ and BMP-2 peptide co-modification (Fig. 2J–L). On the TiO$_2$-deposited surface (TiO$_2$ group), only the signal peaks of carbon, Ti and oxygen were found, while in the groups of Zn and Zn/BMP-2, Zn 2p3/2 and Zn 2p1/2 signal peaks at 1021.75 Da and 1044.85 Da were found (Fig. 2L). In addition, the 1 s signal of nitrogen (N 1 s)was found on the DBCO-TiO$_2$, BMP-2 and Zn/BMP-2 groups (400.13 eV). A gradual increase in N 1 s was also observed when the DBCO-TiO$_2$ surface was further modified with (2-Azido)-PEG$_5$-BMP-2 (i.e., the BMP-2 and Zn/BMP-2 groups) (Fig. 2K). To be exact, the N/Ti atomic ratio increased from 0.052 (the TiO$_2$ group) to 0.686 (the BMP-2 group); the Zn/Ti atomic ratio increased from 0.000 (the TiO$_2$ group) to 0.003 (the BMP-2 group) (Supplementary Table 1). Quantitative analysis revealed that atom percentages of zinc and nitrogen on the Zn/BMP-2 surface were 2.57% and 4.89%, respectively, indicating the efficiency of Zn$^{2+}$ ion and BMP-2 peptide co-modification (Fig. 2M). Then, the durability of BMP-2

on the surface was evaluated by incubating the Zn/BMP-2 substrate in Dulbecco's modified Eagle's medium/F12 (DMEM/F12, 37 °C) for 2 weeks. As shown in Fig. 2N, the intensity of N 1 s signal in XPS showed a slight decrease of <15%. In addition, the durability of the coated clickable peptide labelled by a FITC probe was further checked to confirm the bioactivity. Despite of incubation in DMEM/F12 for 2 weeks, the intensity of fluorescence on the clickable peptide-modified TiO$_2$ surface (Zn/BMP-2 group) did not show significant reduction (Supplementary Fig. 6). Thus, it could be concluded that immobilized BMP-2 peptide is highly stable, probably due to the covalent bonding between DBCO group and azido group via bioorthogonal click chemistry. In addition, the release of Zn$^{2+}$ from the Zn/BMP-2 co-modified surface to PBS was determined by an inductively coupled plasma-atomic emission spectrometry (ICP-AES, JY2000-2, France). The Zn$^{2+}$ release was also comparable to previous reported Zn$^{2+}$-modified surface by sequential sulfonation and magnetron sputtering[28]. Specifically, a burst Zn$^{2+}$ release was observed on the first day (0.145 ppm), and the release slowed down in the following days and reached a steady state (0.04 ppm) lasting 3–4 weeks. Furthermore, the zinc release from FBS-free DMEM and 10% FBS-coanting DMEM were investigated. During the 4-week observation, all the cumulative profiles (Supplementary Fig. 7) showed similar release characteristics. Therefore, the Zn$^{2+}$ concentration in the local microenvironment around the Zn/BMP-2 co-modified Ti implant in vivo is probably at a similar level as in vitro. In a word, these results collectively indicated that the TiO$_2$ based surfaces were successfully co-modified with Zn/BMP-2 and had potential to show long-term bioactivity.

**Surface cytocompatibility in vitro**. After passage 3, the cells were detached with trypsin, and then the cell surface markers were determined by flow cytometry (LSRFortessaTM X-20, BD, USA) analysis to identify the purity of BM-MSCs. As shown in Supplementary Fig. 8, high expressions of CD29 (99.8%) and CD90 (98.9%) and extremely low expressions of the hematopoietic marker CD45 (1.7%) and CD34 (3.1%) were detected, indicating the high purity of BM-MSCs as well as the feasibility of employing these cells in following studies.

The in vitro biocompatibility of Zn$^{2+}$ or BMP-2 modified surfaces was investigated by seeding leukemia cells in mouse macrophage (RAW 264.7 cells) and BM-MSCs on the surfaces. The Live/Dead staining was first carried out and a slight reduction in the number of dead cells was observed on the Zn$^{2+}$, BMP-2 peptide, or their co-modified surfaces compared to the bare TiO$_2$ surface (Fig. 3A). To further quantify the cell viability, RAW264.7 and BM-MSCs were stained by Annexin V/ Propidium iodide (PI) staining and analyzed with the flow cytometry (LSRFortessaTM X-20, BD, USA). These results showed the mean living cell (Annexin V-, PI-) percentage on Zn/BMP-2 co-modified surface was up to 94% (RAW264.7) and 97.9% (BM-MSCs), respectively (Supplementary Fig. 9). In addition, lactic dehydrogenase (LDH) released from cells incubated with Zn$^{2+}$ or BMP-2 modified TiO$_2$ surfaces were detected to determine the cytotoxicity of the materials. After 24 h incubation, the amount of LDH from these cells was slightly lower than that in the bare TiO$_2$ group, indicating that there was no cytotoxicity (Fig. 3E). These implied the poor biocompatibility of the bare TiO$_2$ surface was significantly improved by Zn$^{2+}$ modification and BMP-2 peptide conjugation, promising further application in bone-implants. Furthermore, the morphology of adherent BM-MSCs on different surfaces was also investigated to evaluate the surface cytocompatibility. SEM images showed that BM-MSCs on the TiO$_2$ surface exhibited limited cell spreading

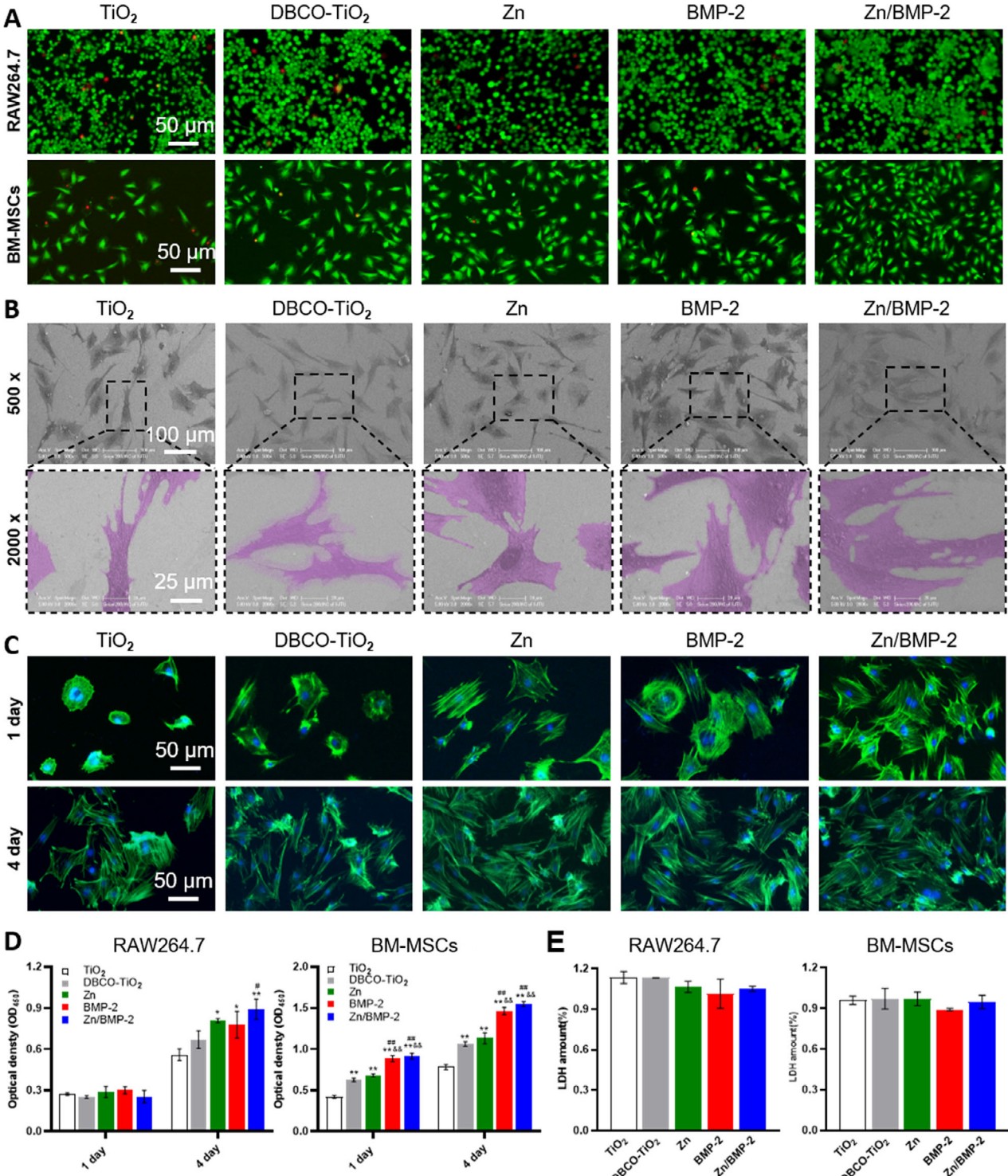

**Fig. 3 Biocompatibility properties of Zn$^{2+}$ and BMP-2 co-modified TiO$_2$ surfaces. A** Live/Dead staining of BM-MSCs and RAW264.7 on the bare and modified TiO$_2$ surface (DBCO-TiO$_2$, Zn, BMP-2 and Zn/BMP-2) (scar bar = 50 μm, three independent experiments). **B** Scanning electron images of BM-MSCs on different surfaces (×500 scar bar = 100 μm and ×2000 scar bar = 25 μm, three independent experiments). **C** The cytoskeleton staining (FITC-phalloidin/DAPI) of BM-MSCs on different surfaces for 1 and 4 days (scar bar = 50 μm, three independent experiments). **D** Cell viability of BM-MSCs and RAW264.7 on different surfaces for 1 and 4 days (CCK-8) (**D** $n = 3$ independent samples per group, by a one-way ANOVA with a Tukey's post hoc test for multiple comparisons). **E** Cell cytotoxicity of BM-MSCs and RAW264.7 on different surfaces for 24 h (**E** $n = 3$, independent samples per group, by a one-way ANOVA with a Tukey's post hoc test for multiple comparisons). Data are reported as mean ± SD, *$p < 0.05$, **$p < 0.01$ compared with the bare TiO$_2$ surface; #$p < 0.05$, ##$p < 0.01$ compared with the DBCO-TiO$_2$ surface; &$p < 0.05$, &&$p < 0.01$ compared with Zn surface. Exact P values were given in the Source Data file.

area compared with cells on the Zn, BMP-2, and Zn/BMP-2 surfaces (Fig. 3B). Further studies with cytoskeleton staining (FITC-phalloidin/DAPI) revealed that BM-MSCs on TiO$_2$ exhibited a relatively spherical morphology with almost no filopodia after a 1-day culture. On the contrary, BM-MSCs on the Zn, BMP-2 and Zn/BMP-2 surfaces exhibited better cell adhesion with polygonal shapes and high expressions of filamentous F-actin (Fig. 3C). It is noteworthy that there were no obvious differences in cell adhesion and spreading state between all groups after a 4-day culture. Nevertheless, cell counting Kit-8 (CCK-8) assay was then employed to evaluate the proliferation of BM-MSCs and RAW 264.7 cells. The results showed that the surfaces of Zn, BMP-2, and Zn/BMP-2 afforded better cell viability for both RAW 264.7 and BM-MSCs compared with others. Interestingly, the Zn$^{2+}$-containing surfaces (i.e., the groups of Zn and Zn/BMP-2) elicited the fastest proliferation of RAW 264.7 cells, while BMP-2 peptide-containing surfaces (i.e., the groups of BMP-2 and Zn/BMP-2) were inclined to enhance BM-MSCs proliferation (Fig. 3D). Impressively, Zn/BMP-2 co-modified surface exhibited better cell adhesion, spreading and proliferation, probably due to the synergistic effect of Zn$^{2+}$ and BMP-2 in promoting the affinity for cells. Taken together, the above results implied that the Zn$^{2+}$ and BMP-2 peptide co-modified surfaces had positive effects on the growth of both macrophages and BM-MSCs with no appreciable cytotoxicity. The reduced cytotoxicity, improved adhesion and proliferation of both the immune cells and multipotential stem cells would be a prerequisite to create a favorable microenvironment for tissue regeneration.

**Macrophage phenotypic switching in vitro.** The typical immunomodulatory mechanism of biomaterials for tissue repair is to switch the macrophage phenotypes from pro-inflammatory M1 to anti-inflammatory M2[21,54,55]. For bone implants, studies have showed that excessive M1 macrophages can cause bones absorption, which is an important factor leading to implant loosening or implantation failure[56,57]. It has been demonstrated that functional coatings, loaded with cytokines or active ions, could promote macrophage polarization toward an M2 phenotype both in vitro and in vivo and improve the integration of implants into bone tissue[24,28]. Regarding bioactive ions, zinc, an essential trace element for some key enzymes and transcription factors, is considered to be indispensable for the development of the adaptive immune system[58,59]. A clinical study reported that patients with inflammatory diseases (e.g., rheumatoid arthritis, RA) had low serum levels of zinc and corresponding increased levels of pro-inflammatory TNF-α; the process could be reversed by the supplementation of zinc[60]. Besides, some previous reports have already suggested that zinc exerts modulatory effects on macrophage phenotype from M1 to M2, inducing anti-inflammatory responses and inhibiting the pro-inflammatory[27,28]. Therefore, the addition of zinc has a potential positive effect on osteoimmunomodulation. In this work, we successfully employed mussel adhesion-mediated ion coordination and molecular clicking strategy to incorporate Zn$^{2+}$ and BMP-2 peptide onto bone implants. To investigate the effect of our strategy on the regulation of immune microenvironment, the polarization of macrophages on these surfaces was then studied. Macrophage-like cells RAW264.7 in the resting state (M0) were first stimulated by lipopolysaccharide (LPS), and their phenotypes were then evaluated after culture on different surfaces (Fig. 4A). As shown, macrophages (M0) were round, and cells stimulated by LPS developed into the pro-inflammatory M1 phenotype with a pancake-like shape (Fig. 4B). Due to different surface modifications, the morphology of macrophages on different surfaces showed different states (Fig. 4C). RAW cells on the surfaces without Zn$^{2+}$ coordination (i.e., the TiO$_2$,

DBCO-TiO$_2$, and BMP-2 groups) were predominantly pancake-shaped. On the contrary, macrophages switched their morphology into an elongated shape on the Zn$^{2+}$ modified groups (i.e., the Zn and Zn/BMP-2 groups). The significant increase of elongated cells on Zn and Zn/BMP-2 surfaces preliminarily indicated the polarization of macrophages from pro-inflammatory M1 phenotype to anti-inflammatory M2 phenotype (Fig. 4D). In addition, to define the shape of macrophages, the degree of cell elongation was further quantified by the ratio of the long axis to the short axis length[61,62]. The macrophages treated with Zn$^{2+}$-containing coatings showed a significant higher rate of cellular elongation than those on the bare TiO$_2$ surface (control group) (Supplementary Fig. 10). Together, these data suggested that Zn$^{2+}$-containing surfaces could influence the macrophages morphology and might have an impact on their macrophage phenotypic conversion.

As we know, the M1 and M2 phenotypes have distinguishable markers on cell surfaces and cytokine secretion profiles[13,63]. Thus, the cytokine secretion and the relative expression levels of inflammatory gene markers were determined with enzyme-linked immunosorbent assay (ELISA) and real-time polymerase chain reaction (RT-PCR). Clearly, the secretion of pro-inflammatory cytokine TNF-α in groups of TiO$_2$, DBCO-TiO$_2$, and BMP-2 was significantly higher than that in the Zn$^{2+}$-modified groups (Zn and Zn/BMP-2) (Fig. 4E), indicating the predominant existence of M1 macrophages on these surfaces. In contrast, the secretion of anti-inflammatory cytokine IL-10 in groups of Zn and Zn/BMP-2 efficiently increased, showing potent inflammation-attenuating effect of Zn$^{2+}$ on macrophage phenotypic switching (Fig. 4F). To further evaluate the macrophages polarization status, the M1 surface markers (CD86 and iNOS) and M2 surface markers (CD206 and Arg-1) in RAW 264.7 macrophages were further labelled by immunofluorescence staining. As shown in Fig. 4G–L, LPS stimulation upregulated the proportion of M1 macrophages (F4/80/CD86$^+$ and F4/80/iNOS$^+$, red) on the surfaces of TiO$_2$, DBCO-TiO$_2$ and BMP-2 groups. In contrast, anti-inflammatory M2 macrophages on the Zn and Zn/BMP-2 surfaces were dominant (CD206$^+$ and Arg-1$^+$, red). Interestingly, the highest percentage of CD206 and Arg-1 positive cells and the lowest percentage of iNOS and CD86 negative cells were found in the Zn/BMP-2 group. It was also worth mentioning that the M1 phenotype markers (iNOS and CD86) were slightly down-regulated while the the M2 phenotype markers were slightly upregulated in BMP-2 group compared with the controls (TiO$_2$ and DBCO-TiO$_2$), probably due to the potential of BMP-2 protein for regulating local osteoimmune microenvironment[64].

Quantitative RT-PCR analysis also showed similar results. The gene expression levels showed downregulation for pro-inflammatory TNF-α and upregulation for anti-inflammatory IL-10 in the case of Zn$^{2+}$ modification, further indicating the potential of Zn$^{2+}$-modified groups for the switch of macrophage to M2 phenotype (Fig. 4M, N). The expression levels of surface markers associated with M1 (e.g., CCR7) and M2 (e.g., CD206) phenotypes were further determined. The efficiently suppressed expression of *Cd206* and significantly increased expression of *Ccr7* in Zn and Zn/BMP-2 groups further indicated the positive immunomodulatory function of Zn$^{2+}$-modified groups for M2 phenotype polarization (Fig. 4O, P). It is interesting to note that the above results all reflected the Zn$^{2+}$ and BMP-2 peptide co-modified surface elicited the most efficient M2 phenotype polarization, probably due to the potential immunomodulatory role of BMP-2 peptide which may motivate the immunoactivity of Zn$^{2+}$ [64]. In addition, the gene expression levels of osteogenic cytokines secreted from the M2 macrophages (e.g., *Bmp-2* and *Vegf*) were also upregulated on the Zn$^{2+}$-containing surfaces, with the highest expression in the Zn/BMP-2 surface (Fig. 4Q, R). The Zn$^{2+}$-enhanced osteogenic cytokine secretion together with

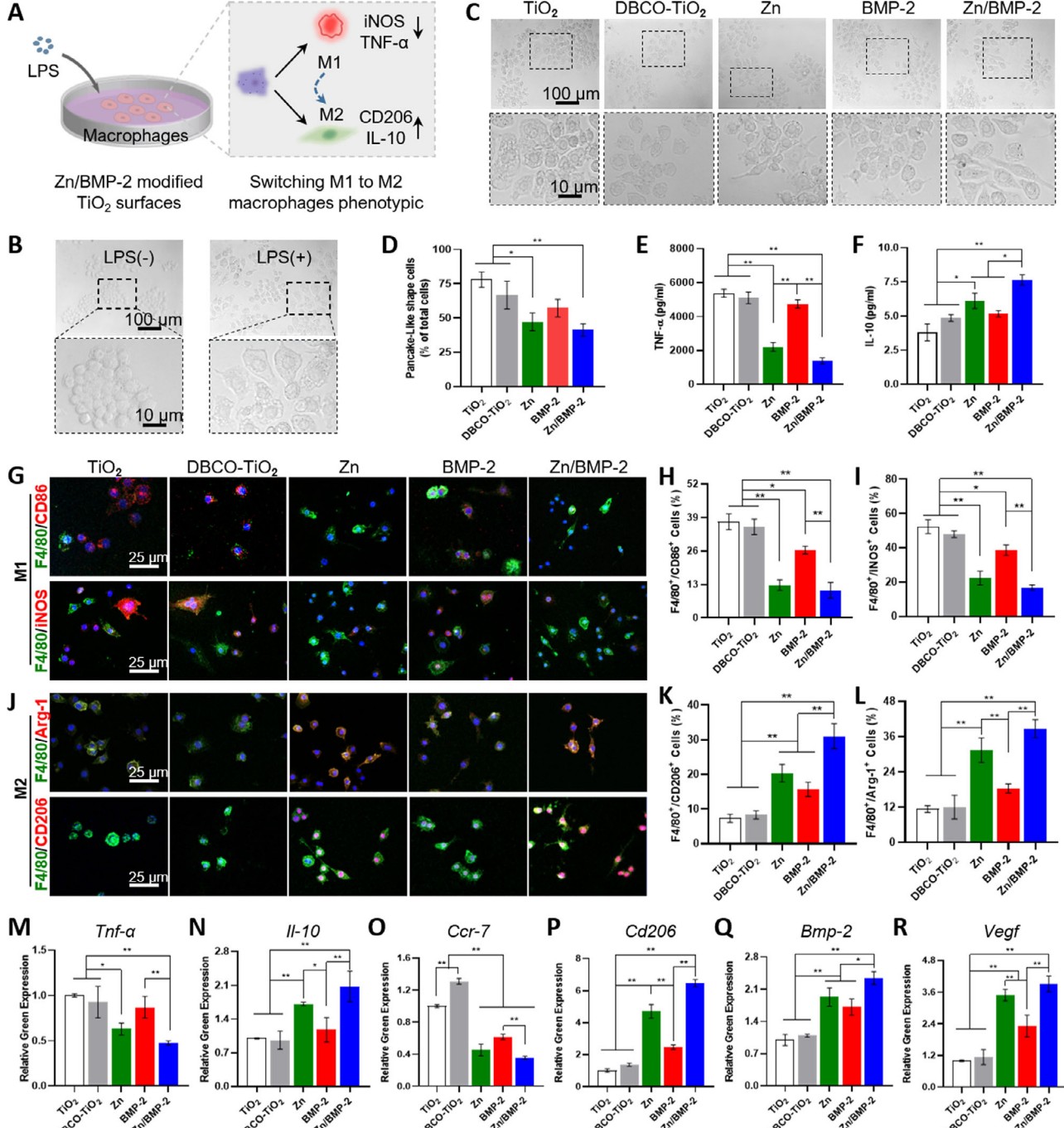

**Fig. 4 Zn²⁺ and BMP-2 peptide co-modified TiO₂ surface regulate macrophages polarization in vitro. A** The illustration of experimental design; **B** The morphology of RAW246.7 was stimulated by LPS or without LPS; **C** The morphology of RAW264.7 was cultured on different surfaces (TiO₂, DBCO-TiO₂, Zn, BMP-2, and Zn/BMP-2 surface) (scar bar = 100 μm and 10 μm, three independent experiments) and (**D**) Quantitative results of pancake-like shape cells (M1) as a proportion of total cells (**D** *n* = 3 biologically independent samples per group, by a one-way ANOVA with a Tukey's post hoc test for multiple comparisons); **E** TNF-α and (**F**) IL-10 cytokine secretion by ELISA (**E**–**F** *n* = 3 biologically independent samples per group, by a one-way ANOVA with a Tukey's post hoc test for multiple comparisons). **G**, **J** Immunofluorescent staining results for RAW264.7 cultured on different surfaces: red (M1 marker: CD86 or iNOS and M2 marker: CD206 or Arg-1), green (F4/80, a monoclonal antibody specifically directing against the mouse macrophage), and blue (nuclei) (scar bar = 25 μm, three independent experiments); Corresponding percentage of double-positive macrophages M1 (**H**, **I**) and M2 (**K**, **L**) (**H**–**I**, **K**–**L** *n* = 3 biologically independent samples per group, by a one-way ANOVA with a Tukey's post hoc test for multiple comparisons); **M**–**R** RT-PCR results of *Tnf-α, Il-10, Ccr7, Cd206, Bmp-2, and Vegf* respectively (**M**–**R** *n* = 3, biologically independent samples per group, by a one-way ANOVA with a Tukey's post hoc test for multiple comparisons). Data are reported as mean ± SD, *\*p* < 0.05, *\*\*p* < 0.01. Exact *P* values were given in the Source Data file.

the BMP-2 peptide-enhanced immunomodulatory function collectively revealed the different but overlapping roles of $Zn^{2+}$ and BMP-2 peptide in immunomodulation and osteoinduction. Therefore, their combination (i.e., Zn/BMP-2) may provide a more favorable microenvironment and create improved bone repair potentials at bone-to-implant interfaces.

Although zinc plays an important and beneficial role in immune functions, its effects depend on the concentration of zinc[58]. A study reported that $1.25 \times 10^{-6}$ M (0.08 ppm) zinc was enough to inhibit the expression of pro-inflammatory cytokines; however, a low concentration of $Zn^{2+}$ (<6.5 ppm) had no osteoclast activity, while higher $Zn^{2+}$ amounts increased osteoclast activity[65]. In the present study, zinc concentration reached 0.145 ppm on the first day, and the total zinc ion released from Zn/BMP-2 dual-effect coating was 0.575 ppm after 28 days (Fig. 2O), which is much lower than 6.5 ppm. Furthermore, zinc ions at the concentration of 0.145 ppm could elevate the proliferation of both macrophages and BM-MSCs (Fig. 3), activate the macrophage phenotypic switch from M1 to M2 (Fig. 4) and enhance the osteogenic differentiation (Fig. 5).

**Immunomodulation-enhanced osteogenic differentiation in vitro**. A larger number of cells, including immune cells and BM-MSCs, are recruited to the bone implant's surface after implantation. The osteogenic differentiation of BM-MSCs in vivo is regulated by not only surface properties of implants but also surrounding immune microenvironment. Accordingly, the immunomodulatory effect of Zn/BMP-2 surface on in vitro osteogenic differentiation was further investigated by using macrophage conditioned medium (MCM)[20,27,28] (Fig. 5A). At day 14, the expressions of three osteogenic-related proteins, including ALP, calcium binding proteins and osteopontin (OPN), were determined to evaluate the efficiency of osteogenic differentiation. The ALP staining clearly showed that significantly higher ALP activity was detected in the MCMs derived from $Zn^{2+}$ or BMP-2 peptide modified surfaces (i.e., the Zn, BMP-2, and Zn/BMP-2 groups) compared with the $TiO_2$ and DBCO-$TiO_2$ group (Fig. 5B). Quantitative analysis of ALP staining revealed that the ALP activity in the Zn/BMP-2 derived MCM were 5.06, 4.22, 2.00, and 1.80-fold higher than that of $TiO_2$, DBCO-$TiO_2$, Zn and BMP-2 groups, respectively (Fig. 5E, F). Meanwhile, ALP immunofluorescence images further confirmed the results that the Zn/BMP-2 derived MCM could induce the highest level of ALP activity (Supplementary Fig. 11), indicating the Zn/BMP-2 co-modified surface could regulate the immunoregulatory effect on osteogenic differentiation of BM-MSCs. In addition to ALP protein level, similar trends were also found in the expressions of other osteogenesis-related proteins (calcium binding proteins and OPN). For example, Alizarin Red S (ARS) staining for calcium binding proteins in the mineralized matrix at day 14 showed that the size and quantity of the mineral nodules in Zn/BMP-2 group was 1.50 and 1.33-fold higher than that of the Zn and BMP-2 groups, indicating the most efficient calcium deposition in BM-MSCs cultured with Zn/BMP-2 derived MCM (Fig. 5C–G). OPN immunofluorescence staining also confirmed the most efficient enhancement of osteogenic differentiation in Zn/BMP-2 derived MCM, in which the relative OPN expression increased by 46.10% and 34.11% compared with the Zn and BMP-2 groups, respectively. Apart from the above osteogenesis-related proteins, we also investigated the expression levels of osteogenesis-related genes, including *Alp*, runt-related transcription factor 2 (*Runx2*), type I collagen (*Col1a1*) and *Opn* (Fig. 5I–L). As expected, the mRNA expressions of these osteogenesis-related genes in BM-MSCs cultured with Zn/BMP-2 derived MCM were all significantly higher than others,

confirming the potent ability of Zn/BMP-2 derived MCM to enhance osteogenic differentiation. Overall, these results demonstrated that the combination of immunoactive $Zn^{2+}$ and osteoinductive BMP-2 peptide would be more conducive to the regulation of macrophage phenotypic switch from M1 to M2, and the secreted cytokines would provide an optimum osteoimmunomodulatory microenvironment and lead to immuno-enhanced osteogenesis.

**Macrophage phenotypic switching in vivo**. Macrophages play a pivotal role in osteoimmunomodulation due to their plasticity and direct function in the inflammatory process of early neo-bone formation. A switch to the M2 phenotype has been shown to be essential for bone healing and osteointegration around implants[17]. In this study, the above in vitro studies have already demonstrated that the Zn/BMP-2 co-modified surface possessed excellent properties including improved cytocompatibility to accelerate cell growth, efficient macrophage phenotype regulation from M1 to M2, and potent osteoimmunomodulatory activity to enhance osteogenicity. These advantages guide us to further investigate the immunomodulatory activity and osteogenicity in vivo. A commercially available cortical bone self-tapping Ti-based screw was used as the model implant. The screws were treated the same as that of $TiO_2$-deposited substrates to obtain different surface modification (named as DBCO-$TiO_2$, Zn, BMP-2 and Zn/BMP-2, respectively). The untreated Ti screw, marked as the control group in vivo, was also named as $TiO_2$ owing to the tight and continuous $TiO_2$ layer on it. Screw implantation in the femoral condyles of rats was then performed according to standard surgical protocol. To minimize the damage to surface layer during implantation, a drill with a diameter between the concave and convex thread of screw was used in animal experiments (the yellow line in Supplementary Fig. 12). 4 days after implantation, the rat femoral condyles containing the implanted Ti screws were harvested for histological hematoxylin-eosin (H&E) and Goldner's trichrome staining. H&E images showed that the tissues near the screws in the groups of Zn, BMP-2, and Zn/BMP-2 exhibited milder inflammatory response, thinner fibrous layers and more complete bone structures as compared with that of the $TiO_2$ and DBCO-$TiO_2$ groups (Fig. 6A). The thinnest fibrous layer was observed in the Zn/BMP-2 group, while the milder inflammatory response in Zn and Zn/BMP-2 screws were comparable (Fig. 6D, E). Goldner's trichrome analysis highlighted the calcified bone (green) significantly increased at the implantation site of Zn/BMP-2 co-modified group in comparison with others, indicating a higher extent of integration and a larger amount of newborn trabecular structures adjacent to the implant (Supplementary Fig. 13). These results implied that the Zn/BMP-2 co-modified screw could create a favorable immune microenvironment for enhancing osteointegration between bone tissues and implants. It was worth mentioning that a slightly less improvement on inhibition of fibrosis and inflammatory response was shown in the BMP-2 group, still indicating the potential immunoactive of the BMP-2 peptide[64]. On the contrary, the thick fibrous layers and large-scale infiltration of inflammatory cells around the bare screw indicate the reason of most implantation failures. In addition, the phenotypic conversion of macrophages around the screws was further evaluated with immunofluorescence staining. Inflammatory cell infiltration was determined by labelling CD68, and the M1 to M2 phenotypic switching was determined by labelling CCR7 and CD206-positive cells, respectively (Fig. 6B). Immunofluorescence images showed CD206-positive cells (i.e., M2 phenotypic macrophages) around Zn, BMP-2 and Zn/BMP-2 screws were significantly more than those around the control screws in $TiO_2$ and DBCO-$TiO_2$.

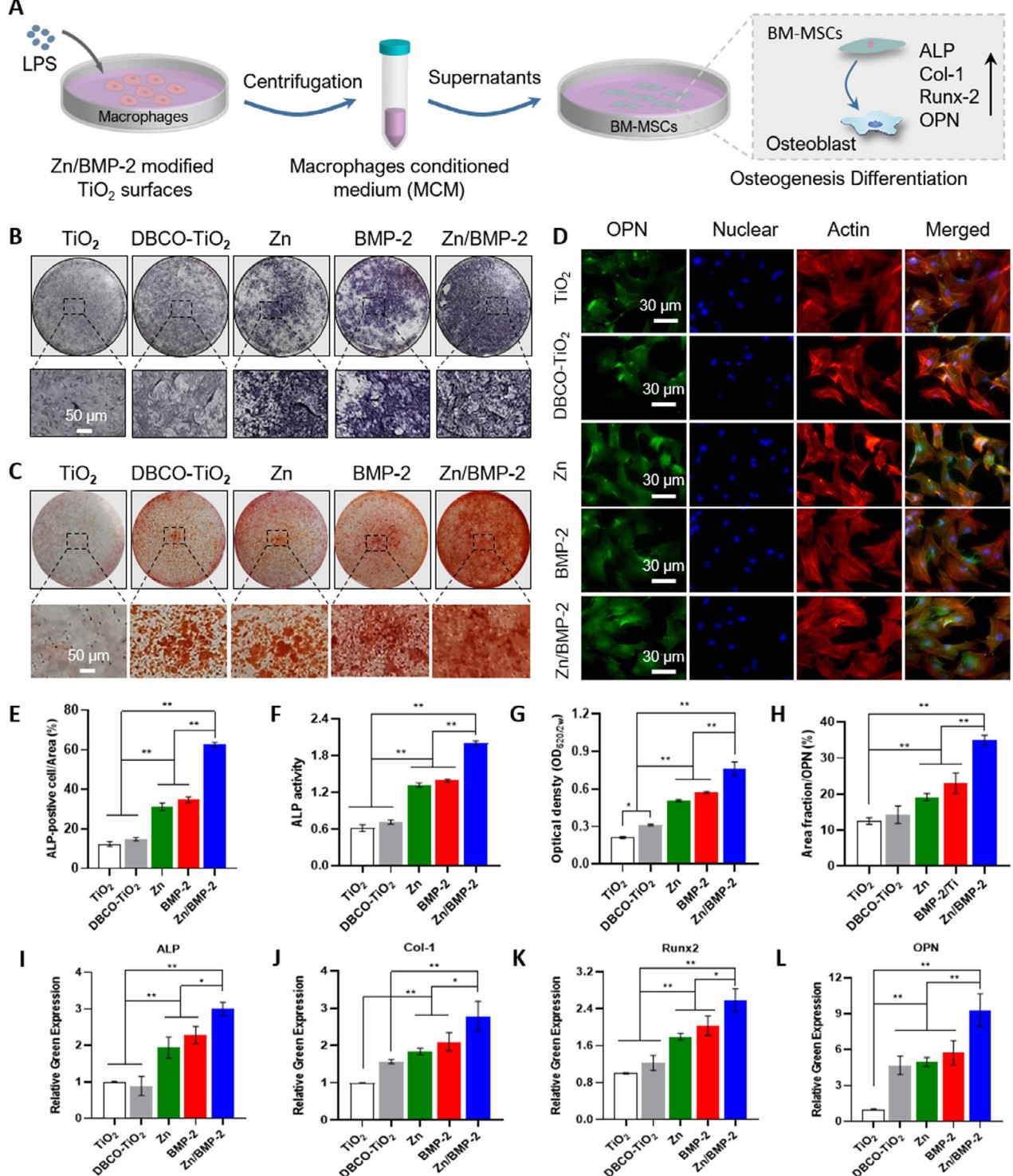

**Fig. 5 Zn²⁺ and BMP-2 peptide co-modified TiO₂ surfaces enhance osteogenic differentiation in vitro. A** The illustration of experimental design; **B**, **E** ALP staining and (**C**, **G**) ARS staining of BM-MSCs cultured in osteogenic medium supplemented with MCM (**B–C** scar bar = 50 µm, three independent experiments); **F** ALP activity assay of the BM-MSCs on the different surfaces; **D** Images of the BM-MSCs after immunofluorescent staining:(green: OPN; red: cytoskeleton and blue: nuclei) (scar bar = 30 µm, three independent experiments) and (**H**) quantitative results; **I–L** Osteogenesis-related genes expression of the BM-MSCs cultured in MCM detected by RT-PCR (*Alp, Runx2, Col1a1 and Opn*). (**E–L** $n = 3$ biologically independent samples per group, by a one-way ANOVA with a Tukey's post hoc test for multiple comparisons). Data are reported as mean ± SD, *$p < 0.05$, **$p < 0.01$. Exact $P$ values were given in the Source Data file.

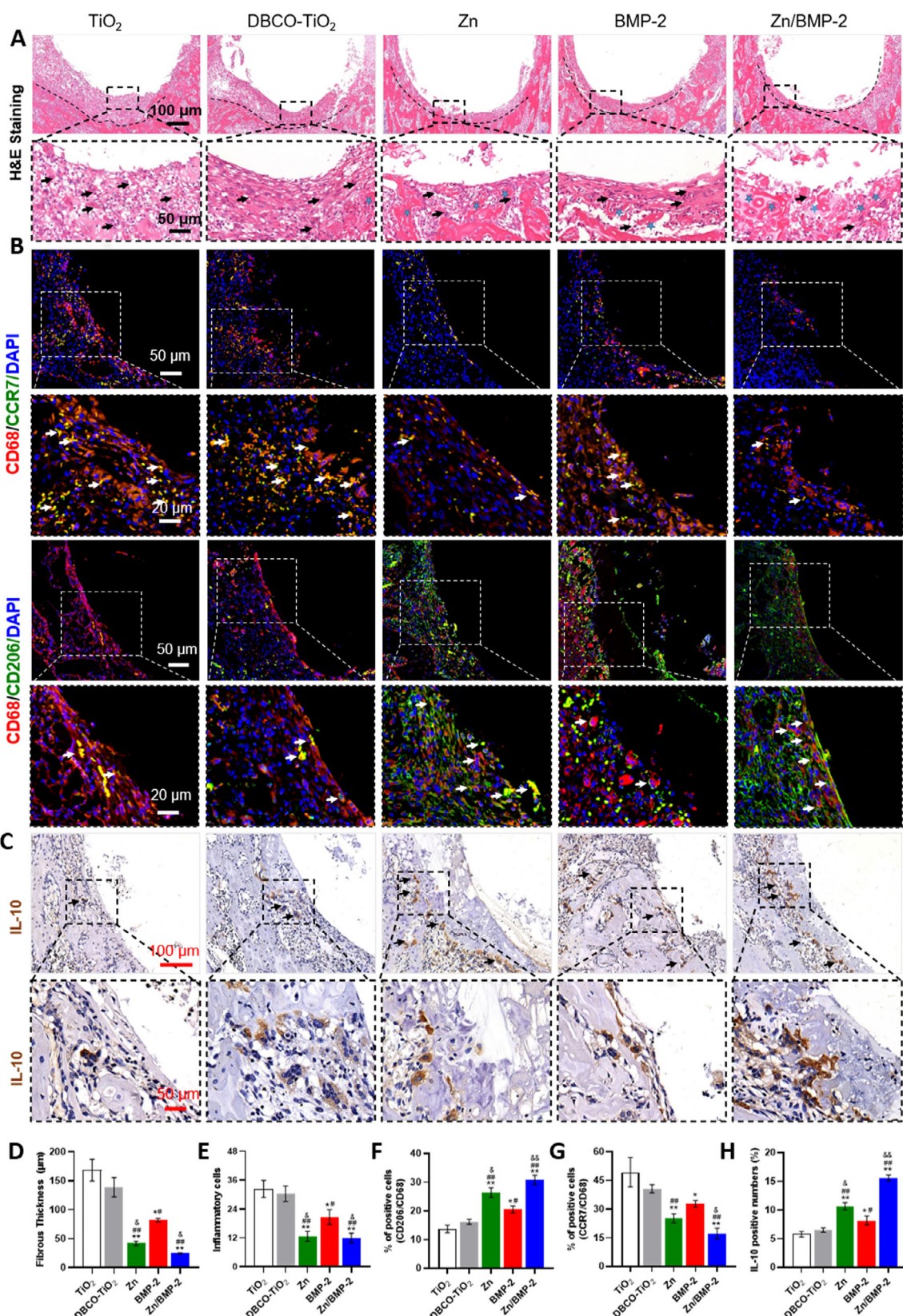

Quantitative analysis showed that the proportion of M2 macrophages in Zn, BMP-2 and Zn/BMP-2 groups was about 1.92, 1.50, and 2.25 folds higher than that in the TiO$_2$ group, respectively (Fig. 6F). In contrast, the number of CCR7-postive cells (M1 phenotypic macrophages) in the TiO$_2$ and DBCO-TiO$_2$ groups was higher than that in others, with the lowest number of M1

macrophages in the Zn/BMP-2 group (Fig. 6G). Immunohistochemical staining further revealed that the deposition of anti-inflammatory cytokine IL-10 dramatically increased in the Zn (10.63%), BMP-2 (8.16%), and Zn/BMP-2 (15.61%) groups as compared with the TiO$_2$ control (5.76%) (Fig. 6C–H). This result, together with the reduced number of infiltrated macrophages and

**Fig. 6 Zn$^{2+}$ and BMP-2 peptide co-modified Ti screws regulate macrophages polarization in vivo. A** H&E staining images of the peri-implant tissue (scar bar = 100 μm and 50 μm, three independent experiments) and quantified with (**D**) fibrous layers and (**E**) infiltration inflammatory cells; **B** Coimmunostaining images of the peri-implant tissue: green (M1 marker, CCR7 and M2 marker, CD206), red (CD68, rat macrophage-specific antigen marker), and blue (nuclei) with white arrows indicating the double-positive cells (scar bar = 50 μm and 20 μm, three independent experiments) and (**F, G**) Quantitative double-positive macrophages; **C** Images of immunohistochemical staining of IL-10 in the peri-implant tissue (scar bar = 100 μm and 50 μm, three independent experiments) and (**H**) quantification of IL-10 positive cells as a proportion of total cells. (**D**–**H** $n = 3$ biologically independent samples per group, by a one-way ANOVA with a Tukey's post hoc test for multiple comparisons). Data are reported as mean ± SD, $^*p < 0.05$, $^{**}p < 0.01$ compared with the bare TiO$_2$ surface; $^#p < 0.05$, $^{##}p < 0.01$ compared with the DBCO-TiO$_2$ surface; $^&p < 0.05$, $^{&&}p < 0.01$ compared with BMP-2 surface. Exact $P$ values were given in the Source Data file.

increased conversion of M1 to M2 phenotypes, confirmed the most potent immunomodulatory activity of Zn/BMP-2 co-modified screw in vivo and implied its potential to reverse the excessive inflammation caused by injury and enhance interfacial osseointegration around the implants.

**Osseointegration in vivo**. As we know, successful osteointegration requires a favorable immune microenvironment, which promotes osteogenesis differentiation and neo-bone formation around the implants[21]. Osteoimmunomodulation determines the capacity of implantable biomaterials on modulating the osteoimmune environment, thus regulating neo-bone formation. As we have verified that the Zn/BMP-2 surface possessed the optimal immunoactivity to regulate macrophage M1/M2 polarization in vitro and in vivo and significantly enhanced osteogenesis in vitro, the new bone formation and osseointegration at the screw-to-bone interface was further evaluated 8 weeks after implantation. The in vivo toxicity reaction of the implanted screws on heart, liver, spleen, lung, and kidney were first evaluated, and no significant tissue toxicity was observed, indicating the cytocompatibility of our strategy (Supplementary Fig. 14). Then, Micro-CT 3D reconstruction and histological analysis of the harvested bone tissues with screws were performed. The reconstructed micro-CT 3D images showed that the highest amount of newly formed bone tissue around the screws was observed in the Zn/BMP-2 group, while few disconnected bone tissues were found in the TiO$_2$ control (Fig. 7A). Quantitative analysis further confirmed this result. Tissues in the Zn/BMP-2 group had the highest percentage of bone mineral density (BMD) and bone volume to tissue volume (BV/TV) and exhibited the best trabecular structural features of the new bone under the same volume of interest (VOI) (Fig. 7B). The value of BV/TV was 1.37 and 1.26-fold higher in the co-modified Zn/BMP-2 group (82.06 ± 1.46%) as compared to that in Zn (59.77 ± 3.89%) and BMP-2 groups (65.35 ± 3.63%), respectively. In addition, the values of trabecular separation (Tb. Sp), trabecular thickness (Tb. Th), and trabecular number (Tb. N) in the Zn/BMP-2 group were 70.92%, 128%, and 162% of that in Zn group whereas 84.28%, 137%, and 143% of that in BMP-2 group, respectively. As expected, the Zn/BMP-2 co-modified surface showed the best osseointegration between bone tissue and implants, probably due to the synergy of immunoactive Zn$^{2+}$ ion and osteoinductive BMP-2 peptide, while a mono-modification (e.g., the Zn or BMP-2 group) might not provide the most favorable immunomodulatory microenvironment for bone regeneration. Sequential fluorescence labelling was further performed by using Calcein (green) and Alizarin Red (red) to mark the newly formed bone, and similar results were obtained (Fig. 7C, D). A large area of new bone mineralization on the screw surface was observed in Zn/BMP-2 group (18.80%), while a gradually decreased bone mineralization was found in the groups of BMP-2 (11.10%), Zn (8.22%), DBCO-TiO$_2$ (4.20%) and TiO$_2$ (3.74%). Quantitative analysis showed that the bone-implant contact ratio (BIC) of Zn/

BMP-2 group was significantly higher (more than twofolds in the value) than the other groups (Fig. 7E).

Since the stable connections between the implants and surrounding bone tissue are closely related to the clinical outcomes of implantation, a biomechanical pull-out experiment was then applied to test the anchorage force of Ti screws in bone tissue. As shown in Fig. 7F and Supplementary Fig. 15, the maximal pull-out forces in the Zn, BMP-2, and Zn/BMP-2 groups all significantly improved as compared with that of the TiO$_2$ control, indicating the excellent mechanical stability. To be exact, the highest maximum pull-out force (203.3 ± 14.3 N) was found in the Zn/BMP-2 group, which was nearly 2.1-fold higher than that of the TiO$_2$ control (98.6 ± 16.0 N). These results confirmed that the Zn/BMP-2 surface could significantly promote interfacial osteogenesis and enhance osseointegration in vivo. Overall, we verified that both the immunomodulatory function and direct osteogenicity were crucial to bone implants, according to the overlapping but distinct stages in tissue regeneration (i.e., immune responses and healing processes). In this work, an osteoinductive and immunomodulatory dual-effect implant was readily obtained by co-modifying immunoactive Zn$^{2+}$ and osteoinductive BMP-2 peptide with a mussel adhesion-mediated ion coordination and molecular clicking method. The dual-effect implant enabled not only M2 phenotypic switching but also direct osteoinductivity, which synergistically created a favorable microenvironment in vivo for bone regeneration (Fig. 7G).

Up to now, no relevant study has successfully combined metal ions and growth factors, and the synergistic effect on osteogenic differentiation or osseointegration is unclear. In this study, we reported on the use of a mussel-like surface coating with immobilized immunomodulatory metal ions (e.g., Zn$^{2+}$) and osteoinductive growth factors (e.g., BMP-2-derived peptide), and we demonstrated the improved in vivo outcomes. Zn/BMP-2 co-modified Ti implants increased bone formation by up to 80% at 8 weeks, which was significantly higher than either stimulator alone[66,67]. For example, some biomimetic implants have improved bone formation (BV/TV) by 28.8% in vivo[66] at 4 weeks via Zn$^{2+}$ delivery; BMP-2 delivery strategies have likewise increased bone formation by 55%[67]. The possible reasons are as follows. Zinc has been delivered as an acute released ion that diffuses to act on cells (e.g., immune cells) distant from the implant, providing a favorable immunomodulatory microenvironment. Besides, BMP-2 peptide has been fixed to the implants chemically, so side effects associated with unexpected changes in release rates are prevented[68,69], and BMP-2 peptide will only act upon recruited/activated cells (e.g., osteogenic cell) that contact implant surfaces directly, therefore improving the osseointegration at bone-to-implant interfaces.

The outlook of this study is that it provides a novel solution in a dual-functional implants with both osteoinductive and immunomodulatory activity for improving osseointegration by a mussel adhesion-mediated ion coordination and molecular clicking strategy to effectively improve mechanical fixation of the bone implants. This strategy involves combining the metal ion

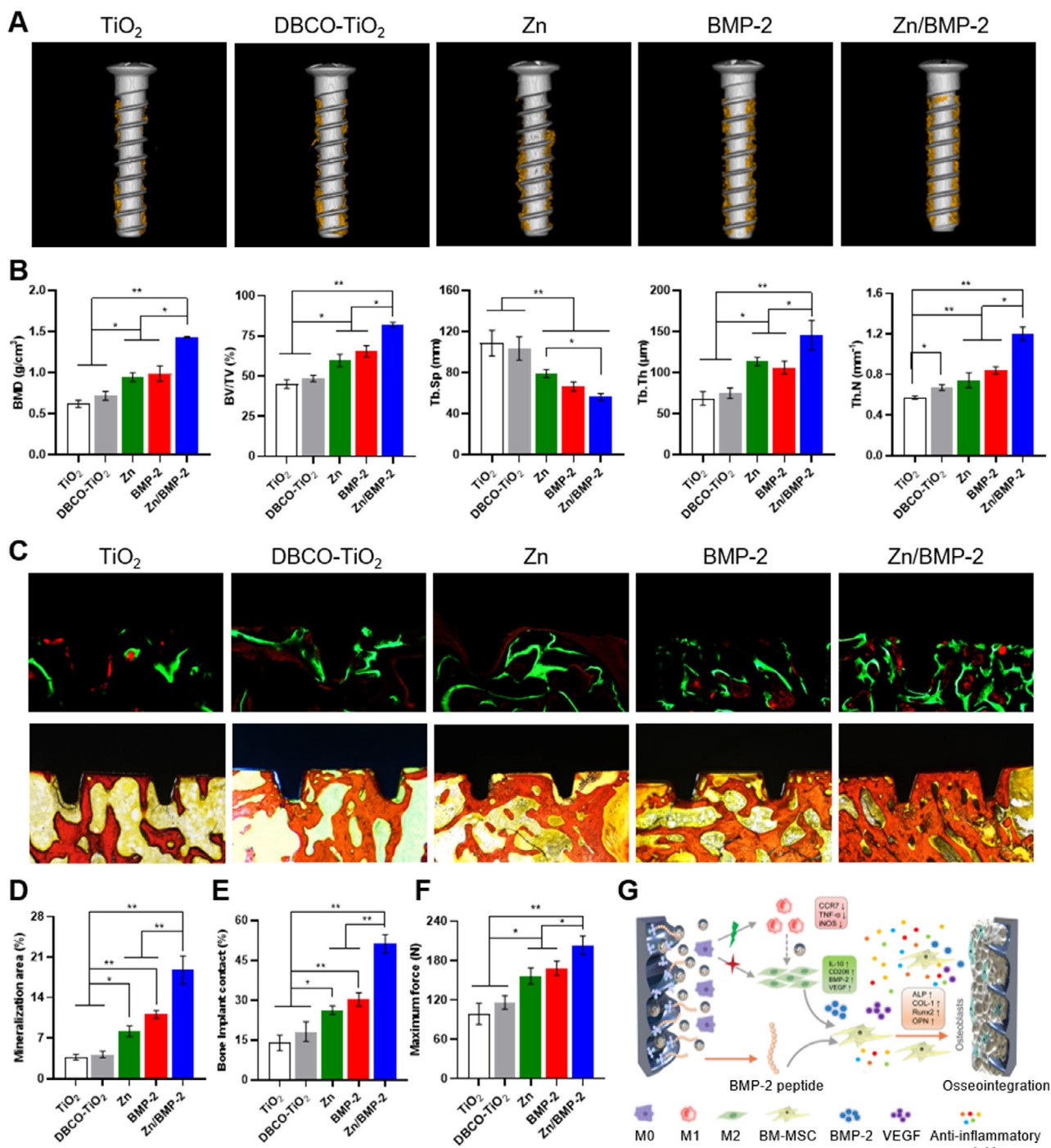

**Fig. 7 Micro-CT and histological analysis on osseointegration between bone tissue and screw. A** Micro-CT 3D reconstructed images and (**B**) quantitatively evaluating the peri-implant bone generation according to the BMD, BV/TV, Tb.Sp, Tb.Th and Tb.N (**B** $n$ = three biologically independent samples per group, by a one-way ANOVA with a Tukey's post hoc test for multiple comparisons). **C–E** Calcein-Alizarin Red staining for the newly formed bone and (**D**) quantitative staining analysis; Van Gieson and bone implant contact (BIC). **F** Maximum fixation force in different groups determined by pull-out testing. **G** Schematic of the bone regeneration mechanism by enabling M2 phenotype switching and osteoinductivity (**D–F** $n$ = 3 independent samples per group, by a one-way ANOVA with a Tukey's post hoc test for multiple comparisons). Data are reported as mean ± SD; *$p < 0.05$, **$p < 0.01$. Exact $P$ values were given in the Source Data file.

(e.g., $Zn^{2+}$) with bioactive peptide (e.g., BMP-2-derived peptide) to overcome the shortcomings of the traditional methods. It not only ensures the long-term bioactivity of peptide, but also combines unique biological activities of the inorganic metal ions with bioactive peptide to meet the various needs of biological materials. The two-step approach has successfully combined

$Zn^{2+}$ with BMP-2 derived peptide, acting as a distant modulator for immune cytokine production in the first stage and achieving local activation/differentiation of recruited target cell clusters (e.g., osteogenic cells instead of immune cells) in the second stage. Given that previous works on bone-implant surface modification focus either on osteoinduction or immunomodulation, our work

is the first to combine the dual functions of molecules and ions for biomodification on bone implants via a mussel adhesion-mediated approach. In addition, BMP-2-derived peptide or zinc ion could be replaced by other biomimetic peptides (e.g., VEGF, AMP) and metal ions (e.g., $Cu^{2+}$, $Mg^{2+}$) to synthesize varieties of multifunctional coatings for satisfying different clinical requirements. Although further exploration is still needed to understand the potential mechanisms of osteoimmunomodulation, these results have demonstrated a promising strategy toward bone regeneration and bone-implant osseointegration, which is in all probability utilized in future clinical practice and applied to orthopedic research. Furthermore, our mussel adhesion-mediated and molecular bioclickable strategy provides a favorable ossteointegration approach to clinical applications in osteoporosis, diabetes, infection, and poor bone healing. The combination of inorganic metal ions with bioactive peptides and biomaterials will provide more opportunities for developing a new generation of engineering bone implants for orthopedic medicine.

In summary, we here reported a dual-effect coating on bone implants with both immunomodulatory and osteoinductive activities by a mussel adhesion-mediated ion coordination and molecular clicking strategy. The strategy could provide a simple method for co-modification of Ti bone implants with immunoactive metal ions (e.g., $Zn^{2+}$) and osteoinductive growth factors (e.g., BMP-2 peptide). The $Zn^{2+}$ and BMP-2 peptide co-modified implants could elicit a favorable osteoimmune microenvironment by macrophage switch from M1 to M2 phenotypes that facilitates the osteogenic differentiation of BM-MSCs, thus enhancing osseointegration at the bone-implant interface and improving their mechanical stability in vivo. Overall, the dual-effect coating could be utilized to regulate macrophage phenotypic conversion and create a favorable immunomodulatory microenvironment for bone regeneration and osseointegration, providing a new idea of bone tissue engineering implants with immunoactivity and osteoinductivity.

## Methods

**Materials**. The clickable mussel-derived peptides were prepared using standard Fmoc solid-phase synthesis[47]. Quartz wafers (10 mm or 15 mm in diameter) with 80–100 nm $TiO_2$ layer were prepared by the key Laboratory of Advanced Technologies of Materials, Southwest Jiaotong University (Chengdu, China). For in vivo experiments, pure Ti screws (2 mm × 10 mm) were purchased from Tianjin Zhengtian Medical Device Company (Tianjin, China). Cell counting kit-8 (CCK-8) kit, a Live/Dead cell staining kit and FITC-labeled Phalloidin staining were purchased from Yeasen Biotechnology Co. (Shanghai, China). Alkaline phosphatase (ALP) kit, ARS kit and Triton X-100 were obtained from Beyotime Biotechnology Co. (Jiangsu, China). The other chemical reagents or antibodies unless mentioned elsewhere were almost purchased from Sigma, Abcam or Invitrogen.

**Surface modification**. The peptide coating was performed by immersing the clean $TiO_2$-coated quartz wafers or medical Ti screws in the PBS solution of $(DOPA)_6$-$PEG_5$-DBCO (0.01 mg/mL) for 24 h. Note that the peptide solution needs to be first purged with nitrogen ($N_2$) for 15 min to reduce the oxidation of catechols. The $(DOPA)_6$-$PEG_5$-DBCO-coated substrates or screws were then incubated with 0.05 M zinc acetate (Zn $(CH_3COO)_2$, $ZnAc_2$) in MiniQ water (18.2 MΩ cm) for 12 h. Finally, Zn-loaded substrates or screws were incubated with (2-Azide)-$PEG_5$-BMP-2 (0.1 mg/mL) in PBS solution for 12 h. The $Zn^{2+}$ and BMP-2 peptide co-modified substrates or screws were carefully rinsed with MiniQ water and dried with $N_2$ for further use. All experiments were performed at room temperature.

**Characterizations**. The two synthetic peptides were first purified by high-performance liquid chromatography (HPLC, Agilent system with Kromasil 100-5C18 column) and then evaluated by electrospray ionization mass spectrometry (ESI-MS, Sciex API 150EX LC/MS with Agilent 1100 HPLC). The surface morphology of modified or unmodified $TiO_2$ substrates was evaluated by field emission scanning electron microscope (FE-SEM, Sirion 200, FEI) and atomic force microscopes (AFM, NT-MDT, Russia). The chemical composition of different samples was characterized by energy-dispersive X-ray spectrometry (EDS, Sirion 200, FEI) and X-ray photoelectron spectroscopy (XPS, AXIS Ultra DLD, Japan). Surface wettability of different samples was analyzed by a contact angle instrument Theta Lite (Biolin scientific, Finland). QCM-D (Q-sense AB, Sweden) was used to

determine the mass of peptides modified on the $TiO_2$ surfaces. The concentrations of $(DOPA)_6$-$PEG_5$-DBCO and (2-Azide)-$PEG_5$-BMP-2 used for QCM-D analysis were the same as those used for the peptide coating and bio-orthogonal co-grafting process. To explore the stability of (2-Azide)-$PEG_5$-BMP-2 graft on the $TiO_2$ surface, first (2-Azide)-$PEG_5$-BMP-2 was labeled with 5-FITC, then the labeled peptides was further used to prepare the Zn/BMP-2 coating. The fluorescence distribution was observed by a fluorescence microscope (PCOM, Nikon, Japan) after co-culturing Zn/BMP-2 with DMEM for 14 days, and the data was analyzed by ZEN imaging software (Zeiss, Germany). The release behavior of $Zn^{2+}$ from the Zn/BMP-2 samples in PBS, FBS-free DMEM and 10% fetal bovine serum (FBS)-DMEM was analyzed by an inductively coupled plasma-atomic emission spectrometry (ICP-AES, JY2000-2, France).

**Cell culture**. RAW264.7 (ATCC, TCM13, Shanghai, China) was provided by the Soochow University (Suzhou, China) as a gift. RAW264.7 cells were cultured in alpha-minimum essential medium (α-MEM, HyClone) supplemented with 10% FBS and incubated at 37 °C under 5% $CO_2$ atmosphere. The cell culture medium of α-MEM was refreshed every two days. Cells ($2 \times 10^4$ cells/well) were seeded in 24-well plates for the subsequent experiments in vitro.

Bone marrow-derived mesenchymal stem cells (BM-MSCs) were isolated from the 4-week-old male Sprague Dawley (SD) rats (Shanghai Jihui Experimental Animal Center, Shanghai, China) according to a previous protocol[70]. All animal experiments were approved by the Animal Research Committee of Shanghai Jiaotong University School of Medicine. Briefly, the femur and tibia were collected and separated form muscle and connective tissue. After cutting off both ends of the bone, the bone marrow suspensions were flushed out and suspended in Dulbecco's modified Eagle's medium: F-12 (DMEM/F12, HyClone) containing 10% FBS (Gibco) and 100 U/ml of penicillin/streptomycin. The cell suspension was filtered by 70 μm filters (Millipore, Ireland). The cells were incubated at 37 °C under 5% $CO_2$ atmosphere, the medium was refreshed every 2–3 days. When the cells arrived at 80–90% confluence, the cells were detached from the culture dish by 0.25% trypsin/EDTA. BM-MSCs were evaluated by flow cytometry in identifying the surface specific markers and confirming their purity before any experiment in vitro. Briefly, passage 3 BM-MSCs were suspended in PBS (pH = 7.2) at a density of $1 \times 10^6$ cells/ml and then were stained with FITC anti-rat/mouse CD90 (BioLegend, 202503), PE anti-mouse/rat CD29 (BioLegend, 102207), FITC anti-rat CD45 (BioLegend, 202205) and APC mouse anti-rat CD34 (Novus, NB600-1071) flow cytometry antibodies at 4 °C for 30 min. Cells without any staining was used as a negative control. Cells were washed in PBS and stained with 7-amino-actinomycin D (7AAD, 559925, Biosciences, BD) according to the manufacture instructions. Quantitative fluorescence analysis was performed with a flow cytometer (LSRFortessa$^{TM}$ X-20, BD, USA), and FlowJo$^{TM}$ software (Version 10.7.1) was used to quantified the expression levels of surface markers.

**Cytocompatibility**. BM-MSCs and RAW264.7 cells were separately cultured on samples with different surface treatments for 24 h and gently rinsed with sterilized PBS solution for three times. A Live/Dead cell staining kit (Yeasen, China) consisting of Calcein-AM (green fluorescence) and Propidium iodide (PI, red fluorescence) was used to assess the cell viability. The fluorescent images were acquired by a fluorescence microscope (Nikon, Japan). To further investigate cell viability, cells were collected and incubated with Annexin V-APC and PI. After incubation, binding buffer was added and the cells were analyzed by flow cytometry. All data were analyzed using FlowJo$^{TM}$ software. Lactate dehydrogenase (LDH, Beyotime, China) and cell counting kit-8 (CCK-8, Yeasen, China) assays were performed to evaluate the cytotoxicity of samples in the cell proliferation. To investigate the adhesion and morphology of BM-MSCs on the samples, FITC-labeled Phalloidin staining (Yeasen, China) and SEM (Sirion 200, FEI) were used in our study. After 24 h incubation, the samples were fixed with 4% paraformaldehyde and permeabilized with 0.1% (v/v) Triton X-100, followed by blocking with 4% bovine serum albumin (BSA) and staining with FITC-labeled Phalloidin (Yeasen, Shanghai, China) and 4′,6-diamidino-2-phenylindole hydrochloride (DAPI, Beyotime, China). Cytoskeletal actin and cell nuclei were observed by a fluorescence microscope (PCOM, Nikon, Japan). Besides fluorescence staining, SEM analysis was also used to investigate the cell adhesion and morphology. After incubating 24 h, cells on different samples surfaces were fixed with glutaraldehyde (2.5% v/v) for 2 h and dehydrated in gradient ethanol at 30, 50, 70, 85, 90, and 100 v/v%. Then, the samples were observed by SEM (Sirion 200, FEI).

**Macrophage polarization in vitro**. In order to investigate the effect of $Zn^{2+}$-modified surfaces in regulating the polarization of macrophage, $2 \times 10^4$ cells/well RAW264.7 cells were cultured on different surfaces in 24-well plate. Then cells were stimulated with 100 ng/mL lipopolysaccharide (LPS, Sigma) for 8 h to induce M1 phenotype. After rinsed in PBS for three times, the complete medium was changed with fresh α-MEM medium. After 24 h culturing, adherent cells were observed by an Olympus CK40 culture microscope (Tokyo, Japan) to investigate the morphology change. The medium supernatants were collected to analyze the secretion of inflammatory cytokines (TNF-α and IL-10) by a commercial mouse cytokine ELISA kits (BD Bioscience, USA). Quantitative real-time polymerase chain reaction (qRT-PCR) was also used to quantify the expression levels of

inflammatory cytokines (TNF-α and IL-10), surface marker (CCR7, M1 macrophage marker and CD206, M2 macrophage marker), and the osteogenic factors (BMP-2 and VEGF). The total RNA in cells was extracted by Trizol reagent (Invitrogen, USA) according to the manufacturer's instructions. RNA concentrations were quantified by a NanoDrop spectrophotometer (Thermo, USA). Immediately, cDNA was synthesized from 1 µg RNA by reverse transcription reaction using a PrimeScript$^{TM}$ RT reagent Kit (Takara, Japan). Gene expression analysis was performed by a 7500 Real-Time PCR System (Thermo, USA) using SYBR® Premix Ex Taq$^{TM}$ (Takara, Japan). The sequences of primers were exhibited in the Supporting Information (Supplementary Table 2). In addition, immunofluorescence staining was used to evaluate the polarization of macrophage from M1 to M2 phenotypes. The expression of M1 macrophage-related markers (CD86, iNOS) and M2 macrophage-related markers (CD206, Arg-1) were confirmed by double-staining as previously reported[31]. Briefly, RAW264.7 cells were fixed with 4% paraformaldehyde (PFA, Sangon Biotech) overnight at 4 °C, permeabilized with 0.2% (v/v) Triton X-100 (Sigma, US) for 5 min, blocked with 2% bovine serum albumin (BSA, Sigma, US) for 1 h and incubated with primary antibodies at 4 °C overnight. The primary antibodies included F4/80 (Abcam, ab6640), CD86 (Abclonal, A16805), iNOS (Abcam, ab3523), CD206 (Abcam, ab64693), and Arg-1 (Abcam, ab91279). After incubation, the samples were rinsed with PBS for 5 min and incubated with the goat anti-rabbit IgG H&L (Alexa Fluor 647, red; Abcam, ab150079) and goat anti-rat IgG H&L (Alexa Fluor 488, green; Abcam, ab150165) or goat anti-rabbit IgG H&L (Alexa Fluor 488, green; Abcam, ab150077) antibodies for 1 h at room temperature. 4,6-diamidino-2-phenylindole (DAPI) was used for nucleus counterstaining and then the samples were imaged by a laser confocal microscopy (LSCM, Zeiss, Germany). Coverslips ($n = 3$) in each group were included for semi-quantitative analysis; and three different subregions were randomly selected. Positive cells and images were analyzed by the Image J (version 1.51a, NIH) software.

**Osteogenic differentiation in vitro.** To investigate whether $Zn^{2+}$-loaded substrates could affect the differentiation of BM-MSCs though modulating the polarization of macrophage, the supernatants of RAW264.7 cells cultured on different surfaces in inflammatory conditions was collected. Then, the supernatant was centrifuged at $\times 60\,g$ for 5 min to move any remained cells and frozen at −80 °C for further use. In addition, the supernatant filtered with a 0.22 µm filter (Millipore, Ireland) was mixed with fresh DMEM/F-12 medium at a ratio of 1:2 to obtain MCM. BM-MSCs was cultured with a density at $2 \times 10^4$ cells/well in DMEM/F-12 for 12 h, then the medium was replaced by MCM with osteogenic components (10 mM β-glycerophosphate, 0.1 µM dexamethasone and 0.25 mM ascorbate) for further culturing. After culturing for 14 days, ALP staining and activity were performed as previous description[31]. In addition, the ECM mineralization was evaluated by the Alizarin Red Staining (40 mM, pH = 4.2, Cyagen, China). Total RNA was extracted from the treated BM-MSCs to measure the amount of osteogenesis-related genes expression (e.g., *Alp, Runx2, Col1a1, and Opn*) by qRT-PCR. The primer sequences for the target genes were listed in Supplementary Table 2. Two osteogenic-related proteins (e.g., ALP, OPN) were evaluated by immunofluorescence staining. Briefly, the cells were fixed in 4% paraformaldehyde and permeabilized with 0.2% Triton-X 100, followed by blocking with 2% BSA and incubating with primary antibodies of ALP (PA5-106391, Invitrogen) and OPN (ab63856, Abcam). Subsequently, the cells were respectively incubated with secondary antibody, phalloidin and DAPI. Finally, the cells were observed by a fluorescence microscope (PCOM, Nikon, Japan).

**Animal models.** All the animal experiments were approved by the Animal Research Committee of Shanghai Jiaotong University School of Medicine, and all the operation procedures were conducted according to National Institutes of Health Guide. Sprague Dawley rats (SD, male, 6–8 weeks old) were randomized into five groups (TiO$_2$, TiO$_2$-DBCO, Zn, BMP-2, and Zn/BMP-2). The flat lateral surfaces of the femoral condyles were selected as the surgical site. After proper anesthesia and disinfection, the lateral femoral condyle was gradually exposed and was drilled with a 1.5 mm kirschner wire under the strict asepsis procedure. The Ti screws (2.0 mm × 10 mm, the height of thread is 0.3 mm, Zhengtian, Tianjing) with Zn or BMP-2 co-modified surface were placed vertically in bilateral femoral condyles in each animal ($n = 10$ per group). Then, the incision was sutured in separate layers and all the animals was intramuscularly injected with antibiotic for 3 days after surgery.

**Macrophage polarization in vivo.** Four days after implantation surgery, the animals ($n = 5$ per group) were sacrificed. The bilateral femurs with screws were collected and fixed in 4% PFA for 48 h, and then decalcified in 10% ethylenediaminetetraacetic acid (EDTA, Macklin) for following 30 days. After removing the Ti screws, the femurs were embedded in paraffin. Paraffin sections was processed by microtome (Leica, RM2255, Germany) with 5 µm thickness for the further staining. H&E and Goldner's trichrome staining were used to evaluate the inflammation and new collage synthesis around the implants. To evaluate the polarization of macrophages around the implants, the abovementioned sections were stained with CD68 (a pan-macrophages marker; Abcam, ab201340), CCR7 (M1 maker; Servicebio, GB11502), and CD206 (M2 maker; Servicebio, GB13438). In addition, the

standard immunohistochemistry was performed to identify IL-10 with primary antibodies against IL-10 (1:100 dilution, Servicebio) according to the manufacturer's instructions. To semi-quantitative analysis, we randomly chose three dependent sections from three different animals in each group to calculate the number of positive cells for CD68, CCR7, CD206, and IL-10 by Image J (version 1.51a, NIH) software.

**Micro-CT evaluation.** After 8 weeks of implantation, the rats ($n = 5$ per group) were sacrificed. The femurs were isolated and fixed in 70% alcohol for 3 days. Then, those femurs were scanned by a Micro-CT system (SkyScan1172 Ex-Vivo Micro-CT, Belgium) with a rotation step of 0.15°, a pixel size of 17.81 µm, 80 kV source voltage, 112 µA source current, 0.5 mm Al filter optimizing the contrast, and 370 ms exposure time. Multilevel thresholding procedure (threshold for new bone = 50–80, threshold for implant = 130) was applied to distinguish bone from other tissues. The VOI included the trabecular compartment between the outer diameter and inner diameter from the longitudinal axis of the screw. Specifically, the VOI was selected in an axisymmetric cuboid with a circular plane 2 mm ($d = 17.81\,\mu m \times 113$ layer) from the top view (B3 in Supplementary Fig. 12) and a depth of 6 mm (L = $17.81\,\mu m \times 340$ layer) along the longitudinal axis of the screw (B1 and B2 in Supplementary Fig. 12). The 3D images were reconstructed by NRecon software (Version1.7.3.0, Bruker, Kontich, Belgium) with correction for misalignment and ring artefacts. Diversity index of bone regeneration including bone tissue volume/total tissue volume (BV/TV), BMD, trabecular thickness (Tb.Th), trabecular number (Tb.N), and trabecular separation (Tb.Sp) were calculated by supporting analyzing software (CTAn, Bruker, Kontich, Belgium). To further investigate the maximum load of different screws, a biomechanical pull-out testing was performed[1]. The screw was pulled out with a displacement rate of 3 mm/min by a material testing system (HY1080, China). The test was stopped when the screw was completely separated from the bone. Meanwhile, the maximum load was recorded and used to evaluate the mechanical stability. Each groups had three parallel replicates.

**Histological evaluation.** To evaluate bone mineralization in vivo, calcein (20 mg/kg, Sigma) and alizarin red (30 mg/kg, Sigma) were intraperitoneally injected into the rats, 4 and 6 weeks after the surgery, respectively. After micro-CT examination and fixnation in 70% alcohol, those femurs around the implantation site were dehydrated in alcohol and acetone. Then these femurs were embedded in methyl methacrylate solution, and processed to get femoral sections with 5 µm thickness. All the fluorescent-labeled bone sections were visualized by LSCM (Zeiss, Germany). In addition, the sections were stained with van Gieson dye and observed by an optical microscope (Nikon, Japan) in the BIC. Finally, images were acquired and analyzed by Image J software to determine the mineralization rate and BIC.

**Statistics and reproducibility.** The results were presented as the means ± SD for each group. The data were analyzed by a one-way ANOVA followed by Tukey's post hoc test for multiple comparisons using the GraphPad Prism software 8.0 (Solvusoft, US). Differences between two groups were considered significant when the $p$ value is <0.05. All micrograph assays were carried out at least three independent times with similar results.

**Reporting summary.** Further information on research design is available in the Nature Research Reporting Summary linked to this article.

## Data availability
The authors declare that all data supporting the findings of this study are available within this paper and its Supplementary Information. All data is available from the corresponding authors upon reasonable request. The Source data underlying Figs. 2C, D, F, H, J–L, N, O, 3D, E, 4D–F, H, I, K, L, M–R, 5E–L, 6D–H, and 7B, D–F; Supplementary Figs. 2, 3C, D, 4, 6B, 7, 9B, C, 10B, 11B and 15 are provided as a Source data file. Source data are provided with this paper.

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

## Acknowledgements

This work was supported by the National Natural Science Foundation of China (81972134 and 82102535), National Key Research and Development Program of China (2018YFC1106200), Shanghai Municipal Education Commission-Gaofeng Clinical Medicine Grant Support (20171906), Shanghai Jiao Tong University "Medical and Research" Program (YG2021ZD06 and ZH2018ZDA04), China Postdoctoral Science Foundation Funded Project (2019M661560), and GuangCi Professorship Program of Ruijin Hospital Shanghai Jiao Tong University School of Medicine.

## Author contributions

T.W. and J.B. performed materials characterization, cell culture, in vitro experiment. L.W. and G.C. guided the in vivo experiments. T.W., C.H., and M.L. performed in vivo implantation. D.G. and J.Q. provided assistant in Micro-CT evaluation and histology analysis. W.C. and T.W. designed and synthesized the peptides. W.C. and L.D. conceived, directed and supervised the study. W.C. and L.D. analyzed the data and wrote the paper with help from all authors. All authors discussed the results and commented on the paper.

## Competing interests

The authors declare no competing interests.
