## [Peer Review File · Nature Communications]

REVIEWER COMMENTS

Reviewer #1 (Remarks to the Author):

This study demonstrated the addition of Zn²⁺ and BMP-2 on the surface of titanium implant promote osteogenesis and osteointegration. It is proposed that Zn²⁺ can modulate the polarization of macrophage while BMP-2 can facilitate the osteogenic differentiation of BM-MSCs. Although the osteogenic performance of Zn²⁺ and BMP2 co-modified coating was well supported by both in vitro and in vivo data, the study was unable to demonstrate any novelty or scientific advancement. This study might contribute to the development of metallic implant coating when the underlying mechanism had been studied. E.g., how this co-factored surface modification contributes to immune-regulated osteogenesis. Giving that there are a number of concerns in particular to the hypothesis of this study, I feel difficulty to recommend this article to the editorial board for publication. Lastly, my comments have been listed below for author's consideration.

1. The design of Mussel adhesion-mediated surface modification has been extensively used in medical implants and summarized elsewhere (Chen, Xu, et al. *Smart Materials in Medicine*, 2020). Moreover, both Zn²⁺ and BMP2 have been widely used in the modification of orthopedic/dental biomaterials because of their well-known osteogenic properties. Thus, it's not surprising that they both contribute to increased osteointegration of bone to the titanium implant. It seems the authors fail to provide very strong evidence to show the advancement of mussel adhesion-mediated Zn²⁺ or BMP2 surface modification over the others.
2. The study implies Zn²⁺ and BMP2 can sequentially modulate early immune response and late direct bone modeling/remodeling process, however, relevant data to support this hypothesis is extremely insufficient. Especially the release kinetics of BMP2 was not given.
3. It is proposed that the release of Zn²⁺ from the implant can polarize macrophage into M2 phenotype, which contradicts to the general observations that Zn²⁺ promote the release of pro-inflammatory cytokines such as TNF- α , IL-1 β , and IL-6 (Gao H, Dai W, Zhao L, Min J, Wang F. *Journal of immunology research*. 2018 Oct;2018). Although the author provided the Zn²⁺ release profile in PBS, it seems that the concentration of Zn²⁺ in the cell culture or animal study has been neglected. Therefore, the immunomodulatory effects of Zn²⁺ in this study has to be carefully verified.
4. M2 macrophages are known to be able to contribute to tissue healing through the release of pro-regenerative cytokines including BMP-2, if Zn-Ti can contribute to polarization of macrophages towards anti-inflammatory M2 phenotypes, the addition of BMP-2 seems to be redundant. Moreover, it is reported that BMP-2 can regulate the polarization of macrophages, so the role of BMP-2 on the early inflammation stage should not be ignored.
5. The study is unable to identify the key factors in the supernatant of macrophages that contribute to the osteogenic differentiation of BM-MSC. Additional experiment using siRNA, neutralizing antibody, or specific inhibitor should be conducted in order to verify the hypothesis. Moreover, the detailed parameters (e.g., speed and time) for centrifugation should have been included in order to exclude the possible involvement of exosomes or microRNA in the indirect coculture.
6. In Fig.4d, OPN was shown highly expressed in nuclei, which is uncommon seen. Please provide the information of the antibody used and explain why OPN is primarily found in the nuclei.
7. A higher magnification image of Fig.5b should be provided to verify the immunofluorescent staining. It seems the colocalization of the pan-marker for macrophage and the makers for M1/M2 macrophage are rare. Why is that?
8. The use of chloralhydrate for animal anesthesia is not acceptable! Not only in terms of animal welfare but also in terms of their potential interference of experiment (Ren, Yu, et al. *The Journal of the American Society of Anesthesiologists* 111.1 (2009): 209-210).
9. The quantification of IL-10, CCR7, CD206 and CD68 should have relied on biological replicates (three different animals), instead of technical replicates (three different areas in one photo) as used in this study.
10. The methods for the quantification of CD86, iNOS, CD206, and Arg-1 were not given. Is it from one biological sample or just one image? The sample size for each experiment should be clearly given throughout the manuscript.
11. The grammar of the manuscript should be thoroughly edited and proofread

- a. "...these exogenous biomaterials are hardly to completely adapt the tissue injury-triggered cell responses,..."
 - b. "...RAW264.7 cells (ATCC, TCM13, Shanghai, China) were kindly a gift from Soochow University ..."
 - c. "...the medium was replaced by refresh α -MEM..."
12. The VOI for the quantification of bone clearly involves cortical bone (Fig.S6), thus the use of parameters like Tb.N, Tb.Th, and Tb.Sp need to be reconsidered.
13. The details of the quantification of mineralization rate was not given. Is it based on Calcein or Alizarin red?

Minor issues:

1. It's difficult to tell whether Zn²⁺ modification is successful because the signal of Zn²⁺ is so weak (weaker than N) in EDS mapping. Perhaps it can be just noise. Moreover, EDS is not an adequate way for the detection of C;
2. The y-axis of Fig.1o disappeared;
3. Fig.5H was never referred in the main text;
4. " α -MEM" is the abbreviation of "alpha-minimum essential medium", not "alpha-modified DMEM medium";
5. The catalogue number for CD206 antibody was not given as other antibodies used;
6. Does "Col-1" mean COL1A1? The gene names presented in this study need to be checked thoroughly and appear in italic font;
7. What does "I.T." mean in Fig.S6? Should it be "i.p. injection"?

Reviewer #2 (Remarks to the Author):

Recommended decision: Major revision, pending recommended changes.

Key results and Validity

As a short summary of the results, the authors have translated a common approach to biomimetic biomaterials, using an adhesive protein/peptide, in an innovative way. They have used the adhesive properties to attach to a medical implant (Ti screws), and chemically crosslink bioactive (BMP, bone morphogenetic protein) and immunomodulatory molecules (Zn), acting together as a coating, to produce very significant changes in healing (bone volume) and inflammation (cytokine profile, macrophage type/recruitment towards Th2/M2, fibrotic capsule reduction). The results are particularly striking, and very convincing. While BMP and ion/ Zn delivery/ release are not novel, and the developed "material" itself is only partially novel, this new biomaterial appears to have high translation value, and the authors have designed a very eloquently study using a widely applicable chemistry approach.

1. Validation of their approach- the authors show convincing evidence (HPLC/mass spectrometry, and EDS/ESCA) that their synthesis and modification approach works.
2. While their characterization of the primary BMSC cells used is lacking, they do provide adequate validation of appropriate cell behavior (proliferation and differentiation in Figure 2 and 4). The authors should provide additional details on how they characterized the purity and behavior of the BMSCs they isolate and how they were isolated (e.g. how pure was the BMSC population)?
3. The authors provide clear and strong evidence that their material/synthesis approach can modulate both inflammatory polarization (using complimentary assessment of gene and protein expression, and surface markers in Figure 3), and BMSC cell differentiation (similar techniques shown in Figure 4). The reported differences between groups are significant, and large enough to be convincing to other scholars and clinicians, particularly providing support for their later in vivo results. Appropriate controls are included to convince the reader.
4. The authors also provide clear, strong, and eloquent evidence that their approach yields large improvements, in vivo, in bone healing and immune modulation (Figure 5 and 6). The drastic

reduction in fibrotic capsule thickness, clear immune cell changes via IHC and histology, and very clear differences in bone growth between groups strongly support their findings, without any unexplained or contradictory findings.

5. The authors fail to provide sufficient information to validate their microCT results, the reader cannot determine what resolution (voxel size) their data is, the accuracy, and how well they could actually distinguish between bone and screw. This MUST be fixed before publication (more detail in "methods" section below).

Significance

1. The clinical problem is significant and unmet (poorly healing bone?), and the scientific approach is novel and innovative. However, this is not well explained in the introduction. In fact, this study is very similar to the first reference (Pan et al.) and, while I do not see a hindering loss of novelty in this, the authors are missing out on a critical opportunity: to build on this prior work! They do not explain what Pan et al. discovered, and how they have expanded on, or differ from, that work. They do not even compare their results to other authors/publications. For example, would it be just as effective to simply inject BMP2? What about Zn? What limitations in modern approaches make the authors' work significant/ important? Is there an actual need for osteogenic orthopedic screws? What makes their approach superior to others? They discuss the theory in this respect in the introduction, but give no tangible values (e.g. currently bone ingrowth into orthopedic screws requires 8 weeks in rodents, while in our study healing occurred in 4 weeks due to...). A reader cannot answer these question after reading this manuscript and introduction/ discussion text, which indicates that the authors have not informed their readers properly.

2. The authors must devote text to differentiate between their and Pans' (and other cited) work, as they are very similar in structure and design. This will highlight the significance of their work.

3. The true novelty of this work is the simplicity- the authors could likely easily substitute the "clicked" BMP-2 with many other pharmaceutical analogues (e.g. VEGF, PTH, etc.), and other ions can be coordinated in place of Zn. Therefore, the translational value is quite high. However, like all devices, the regulatory hurdles will be significant as this is likely a class 3 device. It would benefit the wider audience if the authors could either discuss this aspect, or at least discuss what alternative agents they might use in place of BMP2, and in which applications they might employ their approach (e.g. poorly healing bone, osteoporosis, etc.).

4. There is a lack of context (in part due to poor introduction section) for their results. For example, how does the amount of bone growth (Figure 6B-D) the authors achieve compare to other publications, other approaches, or to simply administering BMP alone? What is the big picture take away for readers? Would this replace bone graft or calcium phosphate augmentation? How much BMP2 would you expect to "release", or is this osteogenic effect only present at the screw/interface where a thin layer of cells/tissue benefit from the BMP-2 coating? Would the authors recommend their approach for poorly growing bone? Is it likely to increase the amount of bone growth (higher final density), or to simply accelerate the formation of bone (same final mineral density as mature bone, just appearing sooner)? Does the device actually integrate (osseointegration) with bone tissue or simply potentiate new/repared trabecula to grow towards and inter-digitate into the screw threads?

Data

1. In supplementary figure S3 it is unclear how strong the osteogenic response is. Please include picture of control well (cells + osteogenic media without BMP). Including the control picture should be standard, and the reader should easily be able to exclude the possibility of pathological cell behavior (e.g. false positive from overactive osteogenesis, which can only be determined by comparing treated to controls). The authors do a good job of showing this in Figure 4C, but it feels incomplete in Figure S3. It would also help the reader if you had a quantitative analysis- approximately how much more did the BMP2 peptide stimulate ALP, or Alizarin-chelated mineralization, and how much did the conjugated BMP-2 peptide stimulate ALP/Alizarin binding? In the results section this information should be written at the point where Figure S3 is first mentioned.

2. SEM/EDS: EDS is typically only semi-quantitative, unless using very rigorous methodology and calibration standards. Please include the standard deviation and number of measurements for all EDS results in the text.

3. Figure 10: In the results text please explicitly state what the minimum concentration needed to elicit a positive response, in vitro and in vivo, so the reader can immediately understand the

significance of your release concentrations (based on prior literature). Ideally, in Figure 10, put a baseline (dotted or dashed line) at the PPM value of Zn that elicits a relatively well known in vitro response, and one for in vivo response (e.g. concentration where >40% of culture Raw cells would be expected to convert to M2 cytokine secretion). At this point in the text the reader may ask, is releasing 0.04 PPM of Zn as steady state good enough to elicit an in vitro, and an in vivo change?

4. The authors should make a supplementary figure showing representative force/displacement curves (Figure 6F), ideally superimposing all curves from each group, in a separate graph for each group. That way the reader can get more information about the mechanical behavior during pull-out, including stiffness, proper failure mechanics, etc. This does not require additional testing- you should already have this data saved in your mechanical test machine, and it should only require exporting, and replotting in excel/origin/or any software.

Methodology

1. In the cell culture, please explicitly describe how you selected for BM-MSCs from the many cell populations collected from the bone marrow. Did you use magnetic beads? Was it a mixed population of cells? How did you confirm the purity, using FACS? The methodology in this section is lacking. Please provide more detailed information, and if this is a routine or common procedure in your laboratory, please reference/cite your prior work so the reader can get some idea of the purity of your MSC population.
2. Cytocompatibility Figure 2- how long did you let cells attach, did you wash off unattached cells?
3. Osteogenic differentiation- just to clarify, did you collect a large pooled stock of conditioned media, freeze it, and thaw/use every 2-3 days during the differentiation process? Did you centrifuge it and filter through a 0.2um filter to ensure that no large cell debris carried over? Mention these details!
4. In the microCT section the methods information is lacking, and this is quite troubling! What settings did you use: voxel size? source voltage? current? Filter material and thickness??? exposure time?? did you frame average? Did you use rotation during scanning? What threshold settings did you use to differentiate between bone and metal? What reconstruction software did you use, and what version of software? How can others considering reproducing or building on your work without this kind of information? This is very important because it is easy to unintentionally reach erroneous conclusions on how much bone has formed with incorrect thresholding, especially if the metal screw cause artifacts (very common). Usually, you cannot correctly threshold/differentiate any closer than 10-100um from the screw surface, but as a reader I cannot even determine whether the results are correct/reliable, because you have not provided the resolution/voxel size! In authors figures, Supplementary Figure 6-B3, it very clearly appears that there is some artifacts arising from the screw. The authors must include a new image in S.Figure 6, "B4" showing a representative image (3D and 2D reconstruction) that indicates any artifacts, and shows exactly their choice of thresholding and where the delineation of bone/screw occurs. Please include in that image a scale bar that is the VOXEL size. This does not require any new experiments.

Analytical approach and statistics: No comments, authors were rigorous and comprehensive in their analytical approach.

Suggested improvements: See details provided in other sections.

Clarity and context: The results and discussion text was clear and concise, but lacked a broader context (e.g. compare to the results of other publication, similar models, use of same drug via alternative methods, etc.). The introduction and abstract must be revised. The conclusion should also be revised.

References: The references are sufficient.

Specific suggestions

- Abstract

- o "We anticipate this study would provide new ideas and solutions for engineering implants with immunoactivity and tissue inductivity to precisely adapt tissue regeneration microenvironment." Strongly recommend avoiding such broad claims, especially in the abstract. The abstract is your one chance to give specific information BEFORE readers will read your full manuscript. The prior text in the abstract suggest you are using well understood approaches to improve the material properties (e.g. releasing bioactive or immunomodulatory ions), how does your work actually provide a new idea, or solution? Explain, specifically, how it does that. For example, you use click

chemistry- but click chemistry is routinely employed to make new materials. How, specifically, have you created a new solution and idea using click chemistry?

- Introduction

- o "Previous considerations for bone implants, however, mostly took a one-sided approach by either to minimize the immune actions or to induce direct osteogenesis at the bone-to-implants interfaces".

- o Did the authors do a pull-out test or push-out mechanical test? In one part they mention "push out" testing. Re-read the manuscript and check for these errors!

- Rewriting suggestions:

- o I hesitate to make specific suggestions as this is a personal choice, but please consider my suggestions below on how to improve your abstract and introduction sections. These are optional and not "required" revisions, though these text sections must be revised and improved in some way.

- o First in abstract:

- Why did you invent this "device", try to describe a specific clinical problem that would benefit (if possible).

- Specify the benefits (results) of your device- e.g. "Zn ion release increase M2 macrophage recruitment by up to X% in vivo, expression of M2 cytokine IL-10 by Y%, and increased total bone volume by Z%, while dual functionalized implants containing both Zn and BMP2 increased M2 macrophage recruitment by up to X% in vivo, expression of M2 cytokine IL-10 by Y%, and increased total bone volume by Z%. Similar benefits were also observed in vitro." You only need to pick your most important result, in this case bone volume, mechanical strength, and inflammatory cell recruitment or expression, in vivo.

- o In the introduction:

- In the first sentence the authors introduce the most relevant work, Pan et al. but do not compare or contrast their work. The reader loses out of a lot of potential information. I recommend restructuring the introduction like so:

- What is the problem you seek to address? You write that orthopedic materials can fix some problems, but are not tailored to the specific needs of each clinical case (e.g. modulating the inflammatory response). Is this really a problem? You do not make clear to the reader that this is an unmet need/problem by citing literature that shows X number of patients could benefit from shifting the immune response towards M2.

- Your assertions are broad. "These evidences indicated the two-sidedness of immune actions, in which macrophages and other immune cells can not only clear cell debris, combat microbes, activate inflammation and promote fibrosis, but also coordinate tissue healing processes by activating stem/progenitor cells and remodeling extracellular matrix for regeneration." This is informative, but a slightly improved statement might say something like, "Macrophages and other immune cells serve dual roles, activating inflammation and promoting fibrosis (M1 response?), but also coordinating tissue healing processes by activating stem/progenitor cells and remodeling extracellular matrix for regeneration (M2 response?)." Since your whole design relies on these 2 response type dichotomy, you should define and explain them, and why you want to trigger an M2 or shift from M1 to M2. That is unclear.

- In paragraph 3 and 4 this would be a perfect place to discuss how your approach differentiates from Pan et al., and other studies in the field.

Reviewer #3 (Remarks to the Author):

General comments:

In this manuscript, the Ti substrate coated with Zn²⁺ and BMP-2 peptide was used to achieve dual functions of (1) immunomodulation and (2) osseointegration. The study introduced commonly used chemical surface coating method of mussel-mediated chemistry, but the difference was emphasized by using bioorthogonal click reaction to solve the random consumption of active groups on peptides (e.g., amino and thiol). Despite advantages, some of results and discussion

needs supportive experiments or further explanation. Please find the detail comments.

Major comments:

- I am not convinced with the statement of “the current biomaterial design is trapped into a one-sided consideration with either focusing on the regulation of immune response or paying attention to induction of new tissue formation.”. Previously, several approaches already endeavored to address both immune response and bone formation at the same time for natural tissue-like bone regeneration.
- Recently, a number of studies on osteoimmunomodulation have been developed by immobilizing osteoinductive factors and immunomodulatory factors at the same time (e.g. growth factors, polyphenols, metal ions). In contrast to those, the significant difference of this study is not clear.
- Why did the authors choose ‘Zn²⁺’ as an immunomodulatory factor? Excluding the other factors such as Sr²⁺ or immune cytokines, which were addressed in the introduction section. The authors should elaborate the function of Zinc ion.
- Were the peptides ((DOPA)6-PEG5-DBCO and (2-Azido)-PEG5-BMP-2) stably coated on Ti surface? Although the chemical analysis including EDS and AFM demonstrated the successful coating initially, the long term evaluation is needed to prove the stability of the peptides coating (release or detachment).
- In page 10, line 6, XPS results are not sufficient to explain changes in amount of immobilized BMP-2 because DOPA also showed similar N1s peak as shown in figure 1K. Release profile of BMP-2 would be more appropriate for this purpose.
- What is ‘Zn concentration’ in Figure 1O? Additionally, explanation about different y-axis for each line should be mentioned.
- In 2.2., further quantitative data such as DNA assay or cell counting should be presented to confirm reduction in dead cells from surface modified groups, which was only explained via live/dead staining.
- During bone regeneration process including immunomodulation, tissue formation, and remodeling, the dose of adapted biomolecules is an important parameter to enhance the natural-like tissue reconstruction. Overdose of inductive proteins or peptides often caused the abnormally and bulky new bone formation. So what is the coating density of Zn²⁺ and (2-Azido)-PEG5-BMP-2) on the Ti surface? The author would provide quantified amount of each. What is your comment on the concentration-dependent immunomodulation or osteogenesis?
- Why the initial cell adhesion and spreading of BM-MSCs were enhanced in Zn, BMP-2, Zn/BMP-2 groups than TiO₂ group? Was it associated with the Zn²⁺ and BMP-2, or other parameter? The authors should elaborate the increase in cell adhesion.
- In in vivo experiments, the inflammatory response was derived from the surgical process during the implantation of the screws because the host femoral bone was damaged, however, it was different from the immune response derived from LPS, which was a case of infection and demonstrated in in vitro analysis. Could Zn²⁺ reduce the inflammatory response in both cases with the same mechanism?
- In page 11, line 4, authors’ implication about Figure 2D was not convincing because enhanced proliferation does not guarantee differentiation and immunoactivity of stem cells.
- In page 11, line 6, positive effect of Zn²⁺ and BMP-2 peptide could not be explained through Figure 2E because there were no significant differences between groups.
- There is no criterion to decide pancake-like cells in this manuscript. It was too subjective that other objective method such as elongation factor would be proper to this purpose.
- In figure 5A, structure of bone was hard to distinguish that other staining method to show mature bone tissue structure, i.e. Goldner’s trichrome staining would be appropriate for this study.
- All the genes written in this manuscript should follow general nomenclature.
- As a minor comment, the forward primer information of Runx-2 and CCR7 in Table S2 was duplicated.
- In figure legend for Figure 2A, typo in living/dead staining.
- In figure legend for Figure 6D, Van Gieson straining should be changed to Van Gieson

Answers to reviewers:

Reviewer #1 (Remarks to the Author):

Comment 1

This study demonstrated the addition of Zn^{2+} and BMP-2 on the surface of titanium implant promote osteogenesis and osteointegration. It is proposed that Zn^{2+} can modulate the polarization of macrophage while BMP-2 can facilitate the osteogenic differentiation of BM-SCs. Although the osteogenic performance of Zn^{2+} and BMP2 co-modified coating was well supported by both *in vitro* and *in vivo* data, the study was unable to demonstrate any novelty or scientific advancement. This study might contribute to the development of metallic implant coating when the underlying mechanism had been studied. E.g., how this co-factored surface modification contributes to immune-regulated osteogenesis. Giving that there are a number of concerns in particular to the hypothesis of this study, I feel difficulty to recommend this article to the editorial board for publication. Lastly, my comments have been listed below for author's consideration.

Reply: Many thanks for the reviewer's critical comments. **In this context, we design a simple and biocompatible surface approach capable of efficient conjugating biomimetic peptide and metal ions for design of dual-functional bone implants with both osteoinductive and immunomodulatory functions to adapt mechanism of bone regeneration.** In our opinions, novelties are summarized as follows. Giving that previous works of surface modification on bone-implants focus either on osteoinduction or immunomodulation, **it is the first time to combine the molecule and ion dual-functions in the field of biomodification on bone-implants by mussel adhesion-mediated approach in our work.** In addition, it is difficult to co-grafted zinc ions and bioactive peptides with traditional methods^{1, 2, 3}. In this study, we combined the metal ion (e.g., Zn^{2+}) with bioactive peptide (e.g., BMP-2-derived peptide) using a mussel adhesion-mediated ion coordination and molecular clicking strategy to overcome the shortcomings of the traditional methods. It not only ensures the long-term bioactivity of peptide, but also combine unique biological activities of the inorganic metal ions with bioactivity peptide to meet the various needs of biological materials. What's more, **BMP-2-derived peptide or zinc ion could be replaced by other biomimetic peptides (e.g., VEGF, AMP) and metal ions (e.g., Cu^{2+} , Mg^{2+}) to synthesize varieties of multifunctional coatings for satisfying different clinical requirements.** In a word, our study provides a promising solution for engineering implants with immunoactivity and tissue inductivity to precisely adapt tissue regeneration microenvironment.

To be better illustrate the innovation of this study, we have added relevant contents in the revised manuscript.

Changed in the revised manuscript (Page 27 Line 554-572, highlight):

The outlook of this study is that it provides a novel solution in a dual-functional implants with both osteoinductive and immunomodulatory activity for improving osseointegration by a mussel adhesion-mediated ion coordination and molecular clicking strategy to effectively improve mechanical fixation of the bone implants. This strategy involves combining the metal ion (e.g., Zn^{2+}) with bioactive peptide (e.g., BMP-2-derived peptide) to overcome the shortcomings of the traditional methods. It not only ensures the long-term bioactivity of peptide, but also combines unique biological activities of the inorganic metal ions with bioactive peptide to meet the various needs of biological materials. Additionally, BMP-2-derived peptide or zinc ion could be replaced by other biomimetic peptides (e.g., VEGF, AMP) and metal ions (e.g., Cu^{2+} , Mg^{2+}) to synthesize varieties of multifunctional coatings for satisfying different clinical requirements. Although further exploration is still needed to understand the potential mechanisms of osteoimmunomodulation, these results have demonstrated a promising strategy towards bone regeneration and bone-implant osseointegration, which is in all probability utilized in future clinical practice and applied to orthopedic research. Furthermore, our mussel adhesion-mediated and molecular bioclickable strategy provides a favorable osseointegration approach to clinical applications in osteoporosis, diabetes, infection, and poor bone healing. The combination of inorganic metal ions with bioactive peptides and biomaterials will provide more opportunities for developing a new generation of engineering bone implants for orthopedic medicine.

Comment 2: The design of Mussel adhesion-mediated surface modification has been extensively used in medical implants and summarized elsewhere (Chen, Xu, et al. Smart Materials in Medicine, 2020). Moreover, both Zn^{2+} and BMP2 have been widely used in the modification of orthopedic/dental biomaterials because of their well-known osteogenic properties. Thus, it's not surprising that they both contribute to increased osseointegration of bone to the titanium implant. It seems the authors fail to provide very strong evidence to show the advancement of mussel adhesion-mediated Zn^{2+} or BMP-2 surface modification over the others.

Reply: Thank you for your comment and suggestions. As you mentioned, both Zn^{2+} and BMP-2 have been widely used in the modification of orthopedic/dental biomaterials by a

variety of ways ¹. These approaches, however, are limited by their deficiencies. Since physical methods always suffer from serious molecular leakage and the lack of long-term activity, current surface bioengineering strategies for bone implants mainly relies on chemical conjugations ^{1, 2}. These traditional chemical methods, however, mostly involve tedious chemical reactions as well as sophisticated surface treatment technologies ^{2, 3}. Apart from potential damage towards the bioactive molecules, the complex procedures also make them to be hardly applied for multicomponent modification due to low controllability and poor operability. What's more, it is difficult to co-grafted zinc ions and BMP-2 peptide. For example, previous studies reported that zinc-doped bone implants mostly need tedious chemical reactions as well as sophisticated surface treatment technologies, such as magnetron sputtering ⁴, acid-etching ⁵, and ion-doping ⁶. However, those traditional chemical methods dose not adopt to graft a bioactive peptide (e.g., BMP-2 peptide) because it destroys its bioactivity. In this context, we design a simple and biocompatible surface approach capable of efficient conjugating biomimetic peptide and metal ions for design of dual-functional bone implants with both osteoinductive and immunomodulatory functions to adapt mechanism of bone regeneration. This novel possesses numerous merits such as ease of operation, high efficiency and specificity, non-use of organic solvents as well as the uniform modification boding well the implants with irregular shapes. It not only ensures the long-term bioactivity of peptide, but also combine unique biological activities of the inorganic metal ions (Zn^{2+}) with BMP-2 peptide to meet the vary needs of biological materials.

In addition, the mussel-inspired peptide biomimetic strategy, which Pan et.al ³ reported, has been extensively used in the medical implants. However, the critical problem of this strategy is the random consumption of active groups (e.g., amino and thiol), which would impede the functions of conjugated biomolecules ^{2, 7}, leading to be unable to display long-term bioactivity. In addition, the second-step chemical conjugation with Michael addition or Schiff base has low specificity and efficiency, taking a toll on the reproducibility and controllability (e.g., heterogeneous molecular conjugation and random molecular orientation) ⁸. What's more, the mussel adhesion-mediated surface modification strategy doesn't take the unique biological activities of the inorganic metal ions into consideration. The novelties of our research are specifically illustrated in **comment 1**.

Comment 3: The study implies Zn^{2+} and BMP2 can sequentially modulate early immune response and late direct bone modeling/remodeling process, however, relevant data to support this hypothesis is extremely insufficient. Especially the release kinetics of BMP2 was not

given.

Reply: Thank you for your comment and suggestions. Broadly, at the early stage of bone healing, inflammatory cytokines peak at 24 h and start to decline at 3-5 days, followed by the initiation of bone repair^{9,10}. In this aspect, our data is consistent with the previous conclusion. At early stage, Zn²⁺ can modulate immune response and then BMP-2 can direct bone modeling/remodeling process at the late stage. On the other hand, Zn²⁺ and BMP-2 play their roles through different mechanisms. The switch from M1 to M2 was induced by free-Zn ion released by this co-modified coating and the bone remodeling process was contributed to direct contact with BMP-2 peptide coated on the surface of implant. Our data showed that the concentration of Zn²⁺ peaked at 24 h *in vitro* and performed therefore early immunomodulation (Figure 1O).

In our study, BMP-2 peptide stably grafted on the surface of titanium. And it doesn't have a clear release profile like zinc ion as you mentioned. BMP-2 peptide grafted could recruit BM-MSCs and promote their adhesion and spreading for better osseointegration on the surface. We also identified the durability of surface modified BMP-2 peptide as follows. The Zn/BMP-2 substrate was incubated in DMEM for 2 weeks. The result, which was the intensity of N 1s signal was tested by XPS, showed just a slight decrease of the intensity of N 1s (less than 15 %) (Figure 1N), indicating a long-term stability of the surface modified BMP-2 peptide. To further confirm the preservation of bioactivity, we labelled the coated clickable peptide with a FITC probe. Despite of the 2 weeks-incubation in DMEM, the intensity of fluorescent on the clickable peptide-modified TiO₂ surface (Zn/BMP-2 group) did not show impressive reduction (Figure S6). Based on above-mentioned results, we believe that the rapid release of immunoactive Zn²⁺ can lead to early immune response and stably modified BMP-2 peptide can direct late bone modeling/remodeling process.

Figure 1. (N) Changes of N 1s signal in the XPS spectrum of the Zn/BMP-2 surface after incubated in DMEM for 2 weeks.

Figure S6. (A) Images of FITC-(2-Azido)-PEG₅-BMP-2 and Zn²⁺ co-modified surfaces:(green: (2-Azido)-PEG₅-BMP-2) and (B) Intensity profile with regions of interest.

As concern to the mechanism of BMP-2, it could play its role in bone development and regeneration through active Smad-dependent pathways whether it is dissociated or not. As reported, Signaling by BMP-2 involves two types of transmembrane serine/threonine kinases, termed type I (BRI) and type II (BRII) receptors on BM-MSCs¹¹. Receptors of both types are needed to form a functional complex to initiate further signaling events. Binding of BMP-2 to the type II receptor induces oligomerization of the receptor complex, resulting in phosphorylation of the type I receptor and recruitment of downstream signaling protein, Smad1, Smad5, and Smad8. Type I BMPR-phosphorylated Smad1 heterodimerizes with Smad4, translocate to the nucleus to act as a transcription factor, and then induces genes that mediate the biological activity of BMP-2¹².

Changed in the revised manuscript (Page 11 Line 231-237, highlighted):

2. Results and discussion

2.1. Mussel-Derived Peptide Synthesis and Surface Modification

Then, the durability of BMP-2 on the surface was evaluated by incubating the Zn/BMP-2 substrate in Dulbecco's modified Eagle's medium/F12 (DMEM/F12, 37 °C) for 2 weeks. As shown in **Figure 1N**, the intensity of N 1s signal in XPS showed a slight decrease of less than

15 %. In addition, the durability of the coated clickable peptide labelled by a FITC probe was further checked to confirm the bioactivity. Despite of incubation in DMEM/F12 for 2 weeks, the intensity of fluorescence on the clickable peptide-modified TiO₂ surface (Zn/BMP-2 group) did not show significant reduction (**Figure S6**). Thus, it could be concluded that immobilized BMP-2 peptide is highly stable, probably due to the covalent bonding between DBCO group and azido group via bioorthogonal click chemistry

Figure S6. (A) Images of FITC-(2-Azido)-PEG₅-BMP-2 and Zn²⁺ co-modified surfaces:(green: (2-Azido)-PEG₅-BMP-2) and (B) Intensity profile with regions of interest.

Comment 4: It is proposed that the release of Zn²⁺ from the implant can polarize macrophage into M2 phenotype, which contradicts to the general observations that Zn²⁺ promote the release of pro-inflammatory cytokines such as TNF- α , IL-1 β , and IL-6 (Gao H, Dai W, Zhao L, Min J, Wang F. Journal of immunology research. 2018 Oct;2018). Although the author provided the Zn²⁺ release profile in PBS, it seems that the concentration of Zn²⁺ in the cell culture or animal study has been neglected. Therefore, the immunomodulatory effects of Zn²⁺ in this study has to be carefully verified.

Reply: We really appreciate the reviewer for the carefulness. We totally agree with the viewer. We found that this cytokines induction effect was discussed under the stimulation of LPS in this review as you mentioned (Gao H, Dai W, Zhao L, Min J, Wang F. Journal of immunology research. 2018 Oct;2018). Incubation with LPS triggers an increase in

intracellular available zinc. Zinc ions have been primarily regarded as a static component of zinc enzymes and transcription factors. The increased intracellular zinc would promote LPS-induced MAPK and NF- κ B activation and transcription and release of proinflammatory cytokines, e.g., TNF- α , IL-1 β , and IL-6.

It was shown that zinc influences the production of these proinflammatory cytokines, but those effects were somewhat contradictory depending on the concentration of zinc that was used. Previous researches showed that zinc concentration was significant in its interaction with the immune systems, which could induce pro-inflammatory responses at higher concentration (e.g., $>100 \times 10^{-6}$ M, 6.25 ppm) while anti-inflammatory responses at lower concentration (e.g., $1.25 \times 10^{-6} \sim 100 \times 10^{-6}$ M, 0.08~6.25 ppm)^{13, 14}. Zn concentration in our study was 0.145 ppm, which can be considered to decrease the expression of pro-inflammatory cytokines and polarize macrophage into M2 phenotype. Some studies have reported similar results to ours^{4, 15, 16}.

In addition, per the Reviewer's suggestion, we have assessed the Zn²⁺ release profile in cell culture conditions of FBS-free DMEM and 10% FBS-containing DMEM as below. It seems that the Zn²⁺ release profile in these conditions were similar compared with PBS group (Figure S7).

Figure S7. Zn²⁺ release profiles of the Zn/BMP-2 surface in PBS solution, FBS-free DMEM and 10% FBS-DMEM. (A) non-accumulative Zn²⁺ release; (B) accumulative Zn²⁺ release.

Changed in the revised manuscript (Page 11 Line243-246, highlighted):

The Zn²⁺ release was also comparable to previous reported Zn²⁺-modified surface by sequential sulfonation and magnetron sputtering⁴. Specifically, a burst Zn²⁺ release was observed on the first day (0.145 ppm), and the release slowed down in the following days and reached a steady state (0.04 ppm) lasting 3~4 weeks. **Furthermore, the zinc release from FBS-free DMEM and 10% FBS-coating DMEM were investigated. During the 4-week observation,**

all the cumulative profiles (**Figure S7**) showed similar release characteristics. Therefore, the Zn^{2+} concentration in the local microenvironment around the Zn/BMP-2 co-modified Ti implant *in vivo* is probably at a similar level as *in vitro*. In a word, these results collectively indicated that the TiO_2 based surfaces were successfully co-modified with Zn/BMP-2 and had potential to show long-term bioactivity.

Comment 5: M2 macrophages are known to be able to contribute to tissue healing through the release of pro-regenerative cytokines including BMP-2, if Zn-Ti can contribute to polarization of macrophages towards anti-inflammatory M2 phenotypes, the addition of BMP-2 seems to be redundant. Moreover, it is reported that BMP-2 can regulate the polarization of macrophages, so the role of BMP-2 on the early inflammation stage should not be ignored.

Reply: Thank you for your comments and suggestions. Zinc ion was initially modified to generate a favorable osteoimmune microenvironment at the early stage that facilitates the osteogenic differentiation and enhances osseointegration at the bone-implant interface. The release of pro-regenerative cytokines BMP-2 was transient. To ensure long term and late direct bone modeling/remodeling process, BMP-2 peptide was introduced onto the co-modified coating. For this aspect, the addition of BMP-2 peptide was to be better promote the osteogenesis of the bone-implant interface.

In the section of Results and Discussion (Macrophage Phenotypic Switching In Vitro), we compared the polarization of macrophages in all groups and found that the Zn^{2+} and BMP-2 peptide co-modified surface elicited the most efficient M2 phenotype polarization. BMP-2 group also suppressed the gene expression of *Tnf-a* and *Ccr7*, reduced TNF-a secretion, however, lower than the Zn group (**Figure 3E, 3M and 3O**). Therefore, our results indicated that BMP-2 might play a slight role in polarization of macrophages. The potential immunomodulatory role of BMP-2 peptide may associate with the immunoactivity of Zn^{2+} as discussed in the sections of results and discussion. In a word, our study revealed the different but overlapping roles of Zn^{2+} and BMP-2 peptide in immunomodulation and osteoinduction.

Figure 3. (E) TNF- α cytokine secretion by ELISA; RT-PCR results of *Tnf- α* (M) and *Ccr7* (O) ($n=3$, independent samples per group. Data are reported as mean \pm SD and analyzed with a one-way ANOVA with a Tukey's post hoc test, * $p<0.05$, ** $p<0.01$).

Comment 6: The study is unable to identify the key factors in the supernatant of macrophages that contribute to the osteogenic differentiation of BM-MSC. Additional experiment using siRNA, neutralizing antibody, or specific inhibitor should be conducted in order to verify the hypothesis. Moreover, the detailed parameters (e.g., speed and time) for centrifugation should have been included in order to exclude the possible involvement of exosomes or microRNA in the indirect coculture.

Reply: Thank you for your comment and suggestions. Many researchers reported that the osteogenic potential of BM-MSCs was significantly enhanced in the macrophage conditioned medium (MCM)-based inductive medium, as evidenced by the upsurge in ALP activity, increased intensity of mineralized nodules, and the augmented levels of osteogenic gene expressions^{17, 18, 19, 20}. It indicates that the MCM method is applicable for exploring cell-cell interactions *in vitro*. However, the specific mechanism is not completely cleared. Luo et.al²¹ showed that macrophages modulate BM-MSCs osteogenic differentiation via alleviation of intracellular oxidative stress. Zhang et.al²² reported that oncostatin M (OSM) and bone morphogenetic protein 2 (BMP-2) by macrophages showed correlation with MSCs gene expression levels for OSM-receptor and BMP-2, suggesting the involvement of both signaling pathways in the osteogenic differentiation of MSCs. Wang et.al²³ thought that it might be the sensitive responsiveness of osteoblasts to the pro-osteogenic cytokines released from the macrophages. Based on the results of qRT-PCR, in our study, we supposed that BMP-2 and VEGF, which were mainly in the supernatant of macrophages medium, might play a key role in the osteogenic differentiation of BM-MSC. To confirm this hypothesis, we quantified the secretion level of BMP-2 and VEGF in the MCM medium by ELISA. The results indicated

that the level of BMP-2 and VEGF in Zn/BMP-2 group were higher than in the control group (TiO₂ group) (**Figure A and B**). In order to further verify this hypothesis, BM-MSCs were cultured in MCM containing the BMP-2 inhibitor²⁴ (Noggin, abs01032, Absin) or VEGF inhibitor²⁵ (sVEGFR1, ab282387, Abcam), and then the gene expression associated with osteogenesis, including *Alp*, *Colla1*, *Runx2* and *Opn*, was further detected using qRT-PCR analysis. The gene expression level of the four major osteogenic-related genes is shown as below (**Figure C**). *Alp*, *Colla1*, *Runx2* and *Opn* were obviously decreased in inhibitor-treated groups compared with MCM groups. Therefore, we believed that the pro-osteogenic cytokines (e.g., BMP-2 and VEGF) might play a key role in the osteogenic differentiation of BM-MSC.

Figure (A) BMP-2 and (B) VEGF cytokine level in MCM measured by ELISA; (C) RT-PCR results of *Alp*, *Runx2*, *Colla1* and *Opn* ($n=3$, independent samples per group. Data are reported as mean \pm SD and analyzed with a one-way ANOVA with a Tukey's post hoc test, * $p<0.05$, ** $p<0.01$).

In addition, MCM was obtained through the following processes in the context. Firstly, the supernatant of different groups collected after centrifuge at 800 rpm for 5 min and frozen at -80 °C for further use. Secondly, the supernatant, removed the debris with a 0.22 μ m filter, was mixed with fresh DMEM/F-12 medium at a ratio of 1:2 to obtain MCM. As reported, ultrahigh speed centrifugation techniques are the most common used isolation methods which is a “gold standard” for the isolation of exosomes. Exosome isolation is realized through four consequent centrifugation steps: 10 min at 300 g, 10 min at 2000 g, 30 min at 10000 g, followed by exosome pelleting by centrifugation at 100000 g for 70 min²⁶. Therefore, the centrifugation rate and time used in this study did not destroy the cells leading to the release of exosomes or microRNAs. In addition, we also referred to other studies and used their

methods to prepare conditioned media to ensure that this method did not have any other effects.

We have added these details in the section of experimental section in the revised manuscript.

Changed in the revised manuscript (Page 34 Line 723-727, highlighted):

4. Experimental Section

Osteogenic differentiation *in vitro*

To investigate whether Zn²⁺-loaded substrates could affect the differentiation of BM-MSCs though modulating the polarization of macrophage, the supernatants of RAW264.7 cells cultured on different surfaces in inflammatory conditions was collected. Then, the supernatant was centrifuged at 800 rpm for 5 min to move any remained cells and frozen at -80° C for further use. In addition, the supernatant filtered with a 0.22 μm filter (Millipore, Ireland) was mixed with fresh DMEM/F-12 medium at a ratio of 1:2 to obtain macrophage conditioned medium (MCM).

Comment 7: In Fig.4d, OPN was shown highly expressed in nuclei, which is uncommon seen. Please provide the information of the antibody used and explain why OPN is primarily found in the nuclei.

Reply: Thank you for the comments. Osteopontin (OPN) is a secreted adhesive glycoprophoprotein expressed by several cell types, and could be found in the cytoplasm and extracellular matrix in bone which is indeed uncommon seen in nuclei²⁷. In our previous Fig.4d, the primary antibody of OPN (affinity, AF0227) might have lower specificity leading to non-specific staining. In order to clearly exhibit OPN protein in BM-MSCs, we have exchanged OPN antibody with a new one (Abcam, ab63856) and stained BM-MSCs again. The new figure 4d was as follows.

Changed in the revised manuscript (Page 21, Figure 4D):

Figure 4d. Images of the BM-MSCs after immunofluorescent staining:(green: OPN; red: cytoskeleton and blue: nuclei)

Comment 8: A higher magnification image of Fig.5b should be provided to verify the immunofluorescent staining. It seems the colocalization of the pan-marker for macrophage and the makers for M1/M2 macrophage are rare. Why is that?

Reply: Thanks for the comments. The rare colocalization of the pan-marker for macrophage and the markers for M1/M2 macrophage may be due to low magnification. We are sorry for that the reviewer was confused by the low magnification images. In order to better visualize the individual cells after co-immunostaining, a higher magnification images of Fig.5B have been presented in the revised manuscript with white arrows indicating the double-positive cells as follows.

Changed in the revised manuscript (Page 24, Figure 5B):

Figure 5. (B) Coimmunostaining images of the peri-implant tissue: green (M1 marker, CCR7 and M2 marker, CD206), red (CD68, rat macrophage-specific antigen marker), and blue (nuclei) with white arrows indicating the double-positive cells.

Comment 9: The use of chloralhydrate for animal anesthesia is not acceptable! Not only in terms of animal welfare but also in terms of their potential interference of experiment (Ren, Yu, et al. The Journal of the American Society of Anesthesiologists 111.1 (2009): 209-210).

Reply: Thanks for the reviewer's constructive comments. We are sorry for that we did use chloralhydrate as an anesthetic in this set of experiments. The use of chloralhydrate for animal anesthesia referred to previously published works ²⁸. But we have lost the sight of the fact that chloralhydrate is considered unethical in many countries. Under your kind suggestions, we have reviewed *Animals Ethics* and referred to experts for additional guidance. We will consult with our ethics committee and use the American Veterinary Medical Association (AVMA) Guidelines for the Euthanasia of Animals (2020) to direct our future experiments. For your concern in their potential interference of experiment, we have shown that there is no difference between chloralhydrate and pentobarbital sodium for animal anesthesia when conducts functional assessment and tissue toxicity *in vivo* (as follow figure). It was found that there was not toxicity in two groups of different anesthesia. And then, sequential fluorescence labelling and VG straining were further performed to mark the newly formed bone, and similar results were obtained. Therefore, in order to make it more acceptable

for you and readers, we replaced “chloralhydrate” with “proper anesthesia” in the revised manuscript.

Figure. (upper) Toxicities of samples on heart, liver, spleen, lung and kidney. H&E staining of the organ tissue sections (50 μ m); (down) Calcein-Alizarin Red and Van Gieson staining for the newly formed bone.

In order to make it more acceptable for readers, the section of methods has been modified in the revised manuscript.

Changed in the revised manuscript (Page 34 Line 745-746, highlighted):

4. Experimental Section

Animal models:

The flat lateral surfaces of the femoral condyles were selected as the surgical site. **After proper anesthesia and disinfection**, the lateral femoral condyle was gradually exposed and was drilled with a 1.5 mm kirschner wire under the strict asepsis procedure.

Comment 10: The quantification of IL-10, CCR7, CD206 and CD68 should have relied on biological replicates (three different animals), instead of technical replicates (three different areas in one photo) as used in this study.

Reply: Thanks for your comments and suggestions. We are sorry for our ambiguous

description. In our work, to semi-quantitative analysis of immunofluorescence staining, we randomly chose three dependent sections from three different animals in each group to calculate the number of positive cells for IL-10, CCR7, CD206 and CD68 by Image J (version 1.51a, NIH) software.

Changed in the revised manuscript (Page 35 Line 763-766, highlighted):

To semi-quantitative analysis, we randomly chose three dependent sections from three different animals in each group to calculate the number of positive cells for CD68, CCR7, CD206 and IL-10 by Image J (version 1.51a, NIH) software.

Comment 11: The methods for the quantification of CD86, iNOS, CD206, and Arg-1 were not given. Is it from one biological sample or just one image? The sample size for each experiment should be clearly given throughout the manuscript.

Reply: Thanks for your comments and suggestions. We are sorry for our ambiguous description. Coverslips (n=3) in each group were included for semi-quantitative analysis; and three different subregions were randomly selected. Positive cells and images were analyzed by the Image J (version 1.51a, NIH) software. According to your comment, the method for the quantification was added in revised manuscript. It was from one dependent sample not one image. We also included the sample size (n) used for statistical evaluation to figure legend, and highlighted in the revised manuscript.

Changed in the revised manuscript (Page 33 Line 716-718, highlighted):

4. Experimental Section

Macrophage polarization in vitro

4,6-diamidino-2-phenylindole (DAPI) was used for nucleus counterstaining and then the samples were imaged by a laser confocal microscopy (LSCM, Zeiss, Germany). Coverslips (n=3) in each group were included for semi-quantitative analysis; and three different subregions were randomly selected. Positive cells and images were analyzed by the Image J (version 1.51a, NIH) software.

Comment 12: The grammar of the manuscript should be thoroughly edited and proofread.

a. "...these exogenous biomaterials are hardly to completely adapt the tissue injury-triggered cell responses,..."

b. "...RAW264.7 cells (ATCC, TCM13, Shanghai, China) were kindly a gift from Soochow University ..."

c. "...the medium was replaced by refresh α -MEM..."

Reply: We really appreciate your help in pointing out those errors. We have rewritten the manuscript with the help of an English native speaker. In addition, we have proofread our manuscript thoroughly, and corrected the errors in our revised manuscript. All the changes have been highlighted in the context as below.

a. The general problem in these exogenous biomaterials is their bio-inertness, lacking in bioactivities to completely adapt to the complex physiological bone regeneration process.

b RAW264.7 (ATCC, TCM13, Shanghai, China) was provided by the Soochow University (Suzhou, China) as a gift.....

c. The cell culture medium of α -MEM was refreshed every two days.

Changed in the revised manuscript (Page 3 and 30, highlighted):

Introduction: (Page 3 Line 44-46, highlighted)

..... The general problem in these exogenous biomaterials is their bio-inertness, lacking in bioactivities to completely adapt to the complex physiological bone regeneration process. Tissue regeneration involves three indispensable stages: (i) immune action, (ii) cell proliferation and new tissue formation and (iii) remodeling and maturation.

4. Experimental Section (Page 31 Line 638-641, highlighted)

Cell Culture

RAW264.7 (ATCC, TCM13, Shanghai, China) was provided by the Soochow University (Suzhou, China) as a gift. RAW264.7 cells were cultured in alpha-minimum essential medium (α -MEM, HyClone) supplemented with 10% FBS and incubated at 37°C under 5% CO₂ atmosphere. The cell culture medium of α -MEM was refreshed every two days. Cells (2×10^4 cells/well) were seeded in 24-well plates for the subsequent experiments *in vitro*.

Comment 13: The VOI for the quantification of bone clearly involves cortical bone (Fig.S6), thus the use of parameters like Tb.N, Tb.Th, and Tb.Sp need to be reconsidered.

Reply: Thank you for your advice and pointing out the mistake. In this study, the VOI was selected in an axisymmetric cuboid with a 2 mm ($D=17.81 \mu\text{m} \times 113$ layer) a circular plane

from the top view (B3 in original Figure S6) and a depth of 6 mm ($L=17.81 \mu\text{m} \times 340$ layer) along the longitudinal axis of the screw (B1 and B2 in original Figure S6). Therefore, there was a mismatch when we marked the scope. To make clearer and more comprehensive for readers, we have reset the VOI as in original Figure S6. The new picture (**Figure S12**) is as follows.

Figure S12. (B) At each 3D location (shown as cross-hair), three orthogonal views are retrieved from the whole dataset and shown as (B1) coronal view (the normal images, in x - y plane), (B2) transaxial view (x - z plane), (B3) sagittal view (z - y plane).

In the process of testing, this volume of interest will be easier to observe and qualitative evaluate the size of bone volume in the 3D reconstructed images from the lateral view. Multilevel thresholding procedure (threshold for new bone=50-80, threshold for implant=130) was applied to distinguish bone from other tissues. In fact, however, the threshold for cortical bone is over 90; the calculation criteria did not affect the amount of new bone formation in our study. Therefore, the use of parameters like Tb.N, Tb.Th, and Tb.Sp are still accurate and may be not to be reconsidered.

Changed in the revised supplementing information (Page 13, Figure S12):

Figure S12. (A) Scheme for implantation surgery for *in vivo* tests and treatment process; (B) At each 3D location (shown as cross-hair), three orthogonal views are retrieved from the whole dataset and shown as (B1) coronal view (the normal images, in x-y plane), (B2) transaxial view (x-z plane), (B3) sagittal view (z-y plane); (B4) Upper: Semi-automated image segmentation was used to define the boundary where new bone occurs (yellow) on a 2D tomogram. The volume of interest is outside the boundary; Lower: 3D rendering of the entire volume of interest and a corresponding longitudinal cut-away view for a representative specimen.

Comment 14: The details of the quantification of mineralization rate was not given. Is it based on Calcein or Alizarin red?

Reply: Thank you for the comment. In this work, to assess bone mineralization *in vivo*, calcein (20 mg/kg, Sigma) and alizarin red (30 mg/kg, Sigma) were injected intraperitoneally

to double fluorescent label of the new bone at 4 and 6 weeks after the surgery, respectively. Therefore, we calculated the mineralization rate based on the sum of red fluorescence area and green fluorescence area in our study and used image analysis tool (Image J, version 1.51a, NIH).

Minor issues:

Comment 1: It's difficult to tell whether Zn^{2+} modification is successful because the signal of Zn^{2+} is so weak (weaker than N) in EDS mapping. Perhaps it can be just noise. Moreover, EDS is not an adequate way for the detection of C;

Reply: Thank you for the comment. We have performed EDS mapping again and achieved clearer pictures as follows (**Figure 1I**). Meanwhile, surface elemental compositions were further determined by XPS to quantify the percentage of Zn^{2+} , and the results showed that Zn 2p_{3/2} and Zn 2p_{1/2} signal peaks at 1021.75 Da and 1044.85 Da could be found as compared with the TiO₂, TiO₂-DBCO and BMP-2 group (**Figure 1J**). Combined with Zn^{2+} release curve (**Figure 1O**), it's safe to assume that Zn has been successfully coated on the TiO₂ surface.

Changed in the revised manuscript (Page 8, Figure 1I)

Figure 1. (I) SEM-EDS elemental mapping for the Zn^{2+} and BMP-2 peptide co-modified surface (Zn/BMP-2)

Figure 1. (J) XPS analysis of the bare and modified TiO₂ surface (DBCO-TiO₂, Zn, BMP-2 and Zn/BMP-2); (O) Zn²⁺ release profiles of the Zn/BMP-2 surface in PBS solution; red (left) and blue (right) represent the non-accumulative and accumulative Zn²⁺ release, respectively.

Comment 2: The y-axis of Fig.1o disappeared;

Reply: Thanks for your carefulness. We have corrected and presented a new image in the revised manuscript.

Changed in the revised manuscript (**Page 8, Figure 10**):

Figure 10. Zn²⁺ release profiles of the Zn/BMP-2 surface in PBS solution. Red (left) and blue (right) represent the non-accumulative and accumulative Zn²⁺ release, respectively.

Comment 3: Fig.5H was never referred in the main text;

Reply: We thank the reviewer for the notice. We have added it in the revised manuscript.

Changed in the revised manuscript (**Page 23 Line 488, highlighted**):

2. Results and discuss:

2.5. Macrophage Phenotypic Switching *In Vivo*

Studies on immunohistochemical staining further revealed that, the deposition of anti-inflammatory cytokine IL-10 dramatically increased in the Zn (10.63%), BMP-2 (8.16%) and Zn/BMP-2 (15.61%) groups as compared with the TiO₂ control (5.76%) (**Figure 5C and 5H**).

Figure 5. (C) Images of immunohistochemical staining of IL-10 in the peri-implant tissue and (H) quantification of IL-10 positive cells as a proportion of total cells. ($n=3$, independent samples per group, Data are reported as mean \pm SD and analyzed with a one-way ANOVA with a Tukey's post hoc test; * $p<0.05$, ** $p<0.01$ compared with the bare TiO₂ surface; # $p<0.05$, ## $p<0.01$ compared with the DBCO-TiO₂ surface; & $p<0.05$, && $p<0.01$ compared with Zn surface).

Comment 4: “ α -MEM” is the abbreviation of “alpha-minimum essential medium”, not “alpha-modified DMEM medium”;

Reply: Many thanks for the reviewer's carefulness. In agreement with the Reviewer's opinion, we have corrected in the revised manuscript.

Changed in the revised manuscript (Page 31 Line 639, highlighted):

4. Experimental Section

Cell Culture

RAW264.7 (ATCC, TCM13, Shanghai, China) was provided by the Soochow University (Suzhou, China) as a gift. RAW264.7 cells were cultured in **alpha-minimum essential medium (α -MEM, HyClone)** supplemented with 10% FBS and incubated at 37°C under 5% CO₂ atmosphere. The cell culture medium of α -MEM was refreshed every two days. Cells (2×10^4 cells/well) were seeded in 24-well plates for the subsequent experiments *in vitro*.

Comment 5: The catalogue number for CD206 antibody was not given as other antibodies used;

Reply: Many thanks for the reviewer's carefulness. The catalogue number for CD206 antibody was ab64693 (Abcam). We have added the catalogue number and brand in our revised

manuscript.

Changed in the revised manuscript (Page 33 Line 711, highlighted):

4. Experimental Section

Macrophage polarization in vitro

The primary antibodies included F4/80 (Abcam, ab6640), CD86 (Abclonal, A16805), iNOS (Abcam, ab3523), CD206 (Abcam, ab64693) and Arg-1 (Abcam, ab91279).

Comment 6: Does “Col-1” mean COL1A1? The gene names presented in this study need to be checked thoroughly and appear in italic font;

Reply: Yes, it does. Thank you for your comment and we apologize for this mistake, which we have checked and corrected in our revised manuscript.

Changed in the revised manuscript (Page 34 Line 732, highlighted) and supplementary information (Table S2, highlighted):

4. Experimental Section

Osteogenic differentiation in vitro (Page 34 Line 732, highlighted)

Total RNA was extracted from the treated BM-MSCs to measure the amount of osteogenesis-related genes expression (e.g., *Alp*, *Runx2*, *Colla1* and *Opn*) by qRT-PCR.

Supporting information: (Table S2, highlighted)

Table S2. Primers used in the RT-PCR of BM-MSCs and RAW246.7 cells.

Cell	Gene	Primers Sequence (5'-3')
BM-MSCs	Alp	F: ATGCTCAGGACAGGATCAAA R: CGGGACATAAGCGAGTTTCT
	Colla1	F: AGCTCGATACACAATGGCCT R: CCTATGACTTCTGCGTCTGG
	Runx2	F: ATCATTCAGTGACACCACCA R: GTAGGGGCTAAAGGCAAAAG
	Opn	F: GAACATGAAATGCTTCTTTCTCAG R: TCCATGAAGCCACAACTAAACTA
	β-actin	F: CCTCTATGACAACACAGT R: AGCCACCAATCCACACAG

RAW264.7	Tnf-α	F: GTTCCCAAATGGCCTCCC R: GTGCTCCTCACCCACACCG
	Il10	F: CCCTTTGCTATGGTGTCTT R: GTGGCCAGTTTGTATTAT
	Cd206	F: TACTTGGACGGATAGATGGAGG R: CATAGAAAGGAATCCACGCAGT
	Ccr7	F: GGTGGCTCTCCTTGTCATTTTC R: AGGTTGAGCAGGTAGGTATCCG
	Vegf	F: AGGAGTCCCCGACGAGATAGA R: CACATCTGCTGTGCTGTAGGAA
	Bmp-2	F: AACGAGAAAAGCGTCAAGCC R: AGGTGCCACGATCCAGTCAT
	β-actin	F: GTGACGTTGACATCCGTAAAGA R: GTAACAGTCCGCCTAGAAGCAC

Comment 7: What does “I.T.” mean in Fig.S6? Should it be “i.p. injection”?

Reply: Thank you for your comment and we are sorry for our mistake. We have confused I.T with the abbreviation of intraperitoneally injection. In the section of methods, we intraperitoneally injected Alizarin and Calcein to mark the new bone *in vivo* (Page 34. Line 9-11). So “I.T.” in original Fig.S6 actually means “I.P injection” with I.P. in our revised version. Therefore, we replaced I.T. with I.P in our revised version. And the revised picture is as follows.

Changed in the revised supplementary information (Page 13, Figure S12):

Figure S12. (A) Scheme for implantation surgery for *in vivo* tests and treatment process.

Reviewer #2 (Remarks to the Author):

Recommended decision: Major revision, pending recommended changes.

Reply: We thank the reviewer's thoughtful and positive comments as well as the opportunity to respond.

Comment 1: Key results and Validity: As a short summary of the results, the authors have translated a common approach to biomimetic biomaterials, using an adhesive protein/peptide, in an innovative way. They have used the adhesive properties to attach to a medical implant (Ti screws), and chemically crosslink bioactive (BMP, bone morphogenetic protein) and immunomodulatory molecules (Zn), acting together as a coating, to produce very significant changes in healing (bone volume) and inflammation (cytokine profile, macrophage type/recruitment towards Th2/M2, fibrotic capsule reduction). The results are particularly striking, and very convincing. While BMP and ion/ Zn delivery/ release are not novel, and the developed "material" itself is only partially novel, this new biomaterial appears to have high translation value, and the authors have designed a very eloquently study using a widely applicable chemistry approach.

Reply: We really appreciate the reviewer's constructive comments, which are very helpful for improving our study. Accordingly, we have revised our manuscript carefully and the point-by-point responses are provided below.

And the novelties of our research are specifically illustrated **in comment 1 of the reviewer 1**. Briefly, giving that previous works of surface modification on bone-implants focus either on osteoinduction or immunomodulation, **it is the first time to combine the molecule and ion dual-functions in the field of biomodification on bone-implants by mussel adhesion-mediated approach in our work**. In addition, it is difficult to co-grafted zinc ions and bioactive peptides with traditional methods^{1,2,3}. In this study, we combined the metal ion (e.g., Zn²⁺) with bioactive peptide (e.g., BMP-2-derived peptide) using a mussel adhesion-mediated ion coordination and molecular clicking strategy to overcome the shortcomings of the traditional methods. It not only ensures the long-term bioactivity of peptide, but also combine unique biological activities of the inorganic metal ions with bioactivity peptide to meet the various needs of biological materials. What's more, BMP-2-derived peptide or zinc ion could be replaced by other biomimetic peptides (e.g., VEGF, AMP) and metal ions (e.g., Cu²⁺, Mg²⁺) to synthesize varieties of multifunctional coatings for satisfying different clinical requirements. In a word, our study provides a promising solution

for engineering implants with immunoactivity and tissue inductivity to precisely adapt tissue regeneration microenvironment.

Comment 2: Validation of their approach- the authors show convincing evidence (HPLC/mass spectrometry, and EDS/ESCA) that their synthesis and modification approach works.

Reply: Thank you for your recognition of our work.

Comment 3: While their characterization of the primary BMSC cells used is lacking, they do provide adequate validation of appropriate cell behavior (proliferation and differentiation in Figure 2 and 4). The authors should provide additional details on how they characterized the purity and behavior of the BMSCs they isolate and how they were isolated (e.g. how pure was the BMSC population)?

Reply: Thank you for the comment and reminding. We completely agree with this reviewer. We have provided additional details on how they characterized the purity and behavior of the BM-MSCs they isolate and how they were isolated in the revised manuscript. The corresponding results were as following.

After passage 3, the cells were detached with trypsin, and then the cell surface markers were determined by flow cytometry (LSRFortessa™ X-20, BD, USA) analysis to identify the purity of BM-MSCs. As shown in Figure S8, high expressions of CD29 (99.8%) and CD90 (98.9%) and extremely low expressions of the hematopoietic marker CD45 (1.7%) and CD34 (3.1%) were detected, indicating the high purity of BM-MSCs as well as the feasibility of employing these cells in following studies.

Figure S8. Immunophenotypic characterization of BM-MSCs. FACS results showed that these cells

were homogenously positive for mesenchymal markers CD29 and CD90; negative for hematopoietic markers CD34 and CD45.

Changed in the revised manuscript (Page 31 and 11, highlighted):

4. Experimental Section (Page 31 Line 644-663, highlighted)

Cell Culture

Bone marrow-derived mesenchymal stem cells (BM-MSCs) were isolated from the 4-week-old male Sprague Dawley (SD) rats (Shanghai Jihui Experimental Animal Center, Shanghai, China) according to a previous protocol⁶⁶. All animal experiments were approved by the Animal Research Committee of Shanghai Jiaotong University School of Medicine. Briefly, the femur and tibia were collected and separated from muscle and connective tissue. After cutting off both ends of the bone, the bone marrow suspensions were flushed out and suspended in Dulbecco's modified Eagle's medium: F-12 (DMEM/F12, HyClone) containing 10% FBS (Gibco) and 100 U/ml of penicillin/streptomycin. The cell suspension was filtered by 70 µm filters (Millipore, Ireland). The cells were incubated at 37°C under 5% CO₂ atmosphere, the medium was refreshed every 2-3 days. When the cells arrived at 80%-90% confluence, the cells were detached from the culture dish by 0.25% trypsin/EDTA. BM-MSCs were evaluated by flow cytometry in identifying the surface specific markers and confirming their purity before any experiment *in vitro*. Briefly, passage 3 BM-MSCs were suspended in PBS (pH=7.2) at a density of 1×10⁶ cells/ml and then were stained with FITC anti-rat/mouse CD90 (BioLegend, 202503), PE anti-mouse/rat CD29 (BioLegend, 102207), FITC anti-rat CD45 (BioLegend, 202205) and APC mouse anti-rat CD34 (Novus, NB600-1071) flow cytometry antibodies at 4°C for 30 min. Cells without any staining was used as a negative control. Cells were washed in PBS and stained with 7-amino-actinomycin D (7AAD, 559925, Biosciences, BD) according to the manufacture instructions. Quantitative fluorescence analysis was performed with a flow cytometer (LSRFortessa™ X-20, BD, USA), and FlowJo™ software (Version 10.7.1) was used to quantified the expression levels of surface markers.

2. Results and discussion (Page 11 Line 250-255, highlighted)

2.2. Surface Cytocompatibility *In vitro*

After passage 3, the cells were detached with trypsin, and then the cell surface markers were determined by flow cytometry (LSRFortessa™ X-20, BD, USA) analysis to identify the purity of BM-MSCs. As shown in Figure S8, high expressions of CD29 (99.8%) and CD90

(98.9%) and extremely low expressions of the hematopoietic marker CD45 (1.7%) and CD34 (3.1%) were detected, indicating the high purity of BM-MSCs as well as the feasibility of employing these cells in following studies.

Figure S8. Immunophenotypic characterization of BM-MSCs. FACS results showed that these cells were homogenously positive for mesenchymal markers CD29 and CD90; negative for hematopoietic markers CD34 and CD45.

References:

66. Zhu H, *et al.* A protocol for isolation and culture of mesenchymal stem cells from mouse compact bone. *Nat Protoc* **5**, 550-560 (2010).

Comment 4: The authors provide clear and strong evidence that their material/synthesis approach can modulate both inflammatory polarization (using complimentary assessment of gene and protein expression, and surface markers in Figure 3), and BMSC cell differentiation (similar techniques shown in Figure 4). The reported differences between groups are significant, and large enough to be convincing to other scholars and clinicians, particularly providing support for their later *in vivo* results. Appropriate controls are included to convince the reader.

Reply: Thank you for the comment and reminding. In this work, we chose medical titanium (Ti) screw as implant model, since Ti materials are widely used in orthopedic and dental surgery. *In vitro*, TiO₂-deposited quartz substrate (noted as TiO₂) was used as the control group to mimic the surface of medical Ti implants. *In vivo* experiments, the untreated Ti screw with tight and continuous TiO₂ layer was marked as the control group.

Comment 5: The authors also provide clear, strong, and eloquent evidence that their approach

yields large improvements, *in vivo*, in bone healing and immune modulation (Figure 5 and 6). The drastic reduction in fibrotic capsule thickness, clear immune cell changes via IHC and histology, and very clear differences in bone growth between groups strongly support their findings, without any unexplained or contradictory findings.

Reply: Thank you very much for your recognition of our work.

Comment 6: The authors fail to provide sufficient information to validate their microCT results, the reader cannot determine what resolution (voxel size) their data is, the accuracy, and how well they could actually distinguish between bone and screw. This MUST be fixed before publication (more detail in “methods” section below).

Reply: Thanks for the comment and advice. We are sorry for our ambiguous description. In our study, the voxel size is 17.81 μm . Image parameters were 80 kV source voltage, 112 μA source current, 0.5 mm Al filter to optimizing the contrast, and 370 ms exposure time. Multilevel thresholding procedure (threshold for new bone=50-80, threshold for implant=130) was applied to distinguish bone from other tissues.

And we have added these details in the revised manuscript as below.

Changed in the revised manuscript (Page 35, 36 Line 769-783, highlighted):

4. Experimental Section

Micro-CT evaluation:

After 8 weeks of implantation, the rats (n=5 per group) were sacrificed. The femurs were isolated and fixed in 70% alcohol for 3 days. Then, those femurs were scanned by a Micro-CT system (SkyScan1172 Ex-Vivo Micro-CT, Belgium) with a rotation step of 0.15° , a pixel size of 17.81 μm , 80 kV source voltage, 112 μA source current, 0.5 mm Al filter optimizing the contrast, and 370 ms exposure time. Multilevel thresholding procedure (threshold for new bone=50-80, threshold for implant=130) was applied to distinguish bone from other tissues. The volume of interest (VOI) included the trabecular compartment between the outer diameter and inner diameter from the longitudinal axis of the screw. Specifically, the VOI was selected in an axisymmetric cuboid with a circular plane 2 mm ($d=17.81 \mu\text{m} \times 113 \text{ layer}$) from the top view (B3 in Figure S12) and a depth of 6 mm ($L=17.81 \mu\text{m} \times 340 \text{ layer}$) along the longitudinal axis of the screw (B1 and B2 in Figure S12). The 3D images were reconstructed by NRecon software (Version1.7.3.0, Bruker, Kontich, Belgium) with correction for misalignment and ring artefacts. Diversity index of bone regeneration including bone tissue volume/total tissue volume (BV/TV), bone mineral density (BMD), trabecular thickness (Tb.Th), trabecular

number (Tb.N), and trabecular separation (Tb.Sp) were calculated by supporting analyzing software (CTAn, Bruker, Kontich, Belgium).

Significance

Comment 1: The clinical problem is significant and unmet (poorly healing bone?), and the scientific approach is novel and innovative. However, this is not well explained in the introduction. In fact, this study is very similar to the first reference (Pan et al.) and, while I do not see a hindering loss of novelty in this, the authors are missing out on a critical opportunity: to build on this prior work! They do not explain what Pan et al. discovered, and how they have expanded on, or differ from, that work. They do not even compare their results to other authors/publications. For example, would it be just as effective to simply inject BMP2? What about Zn? What limitations in modern approaches make the authors' work significant/important? Is there an actual need for osteogenic orthopedic screws? What makes their approach superior to others? They discuss the theory in this respect in the introduction, but give no tangible values (e.g. currently bone ingrowth into orthopedic screws requires 8 weeks in rodents, while in our study healing occurred in 4 weeks due to...). A reader cannot answer these questions after reading this manuscript and introduction/ discussion text, which indicates that the authors have not informed their readers properly.

Reply: Thank you for the comment. In order to answer the questions more clearly, we separate the questions and answer them point by point.

a) The clinical problem is significant and unmet (poorly healing bone?), and the scientific approach is novel and innovative. However, this is not well explained in the introduction. In fact, this study is very similar to the first reference (Pan et al.) and, while I do not see a hindering loss of novelty in this, the authors are missing out on a critical opportunity: to build on this prior work! They do not explain what Pan et al. discovered, and how they have expanded on, or differ from, that work. They do not even compare their results to other authors/publications.

Reply: We have added the significances in the section of introduction and the novelties in the section of discussion which also makes our work superior to others. They were summarized here.

Faced with poorly healing bone and bone-implant integration failing, our paper reports on an exciting research breakthrough for surface bioengineering of bone implants with both immunoactive and bone tissue inductive functions. This co-modified strategy has several advantages. On one hand, relatively simple and biocompatible surface approaches capable of

efficient conjugating multiply bioactivities are highly desired in order to overcome potential damage and complex procedures caused by conventional chemical method. On the other hand, BMP-2-derived peptide or zinc ion could be replaced by other biomimetic peptides (e.g., VEGF, AMP, etc.) and metal ions (e.g., Cu^{2+} , Mg^{2+} , etc.) to synthesize varieties of multifunctional coatings for satisfying different clinical requirements. In a word, it provides new ideas and solutions for engineering implants with immunoactivity and tissue inductivity to precisely adapt tissue regeneration microenvironment.

As regards to Pan et al 's work, they designed two mussel-derived biomimetic peptides for facile biomodification of Ti implants through robust catechol/titanium dioxide (TiO_2) coordinative interactions. The highly biomimetic peptide capped with RGD- or OGP-derived sequence improve could improve not only the biocompatibility of Ti implants but also the efficiency of osteogenicity, osseointegration and mechanical stability *in vivo*. However, the critical problem of this strategy is the random consumption of active groups (e.g., amino and thiol), which would impede the functions of conjugated biomolecules, leading to be unable to display long-term bioactivity. In addition, the second-step chemical conjugation with Michael addition or Schiff base has low specificity and efficiency, taking a toll on the reproducibility and controllability (e.g., heterogeneous molecular conjugation and random molecular orientation). Given these understandings, we designed an improved biomimetic strategy by combining mussel-like adhesion with bioorthogonal click reaction, a specific and biocompatible chemistry. It based on Alkyne–Azido cycloaddition click chemistry which possesses numerous merits such as ease of operation, high efficiency and specificity, non-use of organic solvents as well as the uniform modification boding well the implants with irregular shapes. Meanwhile it doesn't consumption of active group, ensuring the long-term bioactivity of biomimetic peptide. What's more, our work also combined unique biological activities of the inorganic metal ions (e.g., Zn^{2+}) with bioactive peptide (e.g., BMP-2 peptide) to meet the vary needs of biological materials.

b) For example, would it be just as effective to simply inject BMP2? What about Zn? What limitations in modern approaches make the authors' work significant/ important?

Reply: Currently, the most prevalent form of BMP-2 application is by mixing it with bone matrix materials such as collagen ²⁹, chitosan film ³⁰, and porous hydroxyapatite composites ³¹. However, BMP-2 deposited superficially on the material surfaces releases with an early burst and lack of long-term activity due to these methods base on weak noncovalent bonds. By comparison, chemical conjugations are much more stable. However, traditional chemical modification mostly involves tedious chemical reaction, potentially damaging

towards the bioactive molecules. As referred to Zinc, the representative article is Liu's report on a layer of zinc ions incorporated on sulfonated polyetheretherketone (SPEEK) biomaterials by using a customized magnetron sputtering technique. In terms of immune regulation, they obtained a similar result as well as us. But these methods also mostly involve tedious chemical reactions as well as sophisticated surface treatment technologies.

The two parts represent the major limitations faced with traditional chemical methods. Apart from potential damage towards the bioactive molecules, the complex procedures also make them to be hardly applied for multicomponent modification due to low controllability and poor operability. Limitations in physical methods are more obvious, such as serious molecular leakage and the lack of long-term activity. In this context, simple and biocompatible surface approaches capable of efficient conjugating multiply bioactivities are highly desired.

c) Is there an actual need for osteogenic orthopedic screws? What makes their approach superior to others?

Reply: There are actual needs in many clinic situations, especially in osteoporosis, diabetic and poorly healing bone. Because all these conditions are accompanied with persistent inflammation, which disrupts local osteoimmune balance and further undermines bone-implant integration. Based on these clinical unmet, this co-modified coating was designed to combine the function of osteoinduction and immunomodulation for better osseointegration.

d) They discuss the theory in this respect in the introduction, but give no tangible values (e.g. currently bone ingrowth into orthopedic screws requires 8 weeks in rodents, while in our study healing occurred in 4 weeks due to...). A reader cannot answer these questions after reading this manuscript and introduction/ discussion text, which indicates that the authors have not informed their readers properly.

Reply: Based on our review of other publications, we found that different amount of bone growth was achieved with different modified approaches at different timepoints. We list representative results from some of these publications. However, these comparisons seem inclusive because of varied disease models and observation times.

Rahman et.al³² reported that BMP-2 with PLGA/PEG scaffolds into mouse calvarial model and achieved 55% of new bone area after 6 weeks of implantation.

Zhao et.al³³ reported that silicon-doped titania nanotubes modified Ti screws enhanced new bone formation (BV/TV, 16%) at six weeks.

In our approach, the Zn/BMP-2 co-modified bone implants achieved over 80% of new bone formed after 8 weeks of implantation which is several folds lower than the previously mentioned approaches. The probably due to the synergy of immunoactive Zn²⁺ and

osteoinductive BMP-2 peptide, while a mono-modification (e.g., the Zn or BMP-2 group) might not provide the most favorable immunomodulatory microenvironment for bone regeneration.

Comment 2: The authors must devote text to differentiate between their and Pan's (and other cited) work, as they are very similar in structure and design. This will highlight the significance of their work.

Reply: Thank you for the comment. Actually, there is a difference between our work and Pan's. Although there are similarities in the structure and design of biomimetic peptides with Pan's work, we have improved the structure design of peptides on this basis to overcome the original shortcomings: (1) the binding capacity of the biomimetic peptides was insufficient; (2) it was unable to display long-term bioactivity because the random consumption of active groups (e.g., amino and thiol), which would impede the functions of conjugated biomolecules^{2, 7}; (3) the second-step chemical conjugation with Michael addition or Schiff base has low specificity and efficiency, taking a toll on the reproducibility and controllability (e.g., heterogeneous molecular conjugation and random molecular orientation). In our study, the result show that (DOPA)₆-PEG₅-DBCO could steady bind onto the QCM chips and the maximal grafting density was about 489 ng/cm² (**Figure S4**), indicating the high efficiency and spontaneous adhesion onto TiO₂-deposited quartz substrate surface. It is noted that the binding capacity here is higher than that of Pan's work (367 ng/cm²)³⁴, probably due to the improved catechol orientation for surface binding. In addition, this bioclickable strategy based on Alkyne–Azide cycloaddition click chemistry possesses numerous merits such as ease of operation, high efficiency and specificity, non-use of organic solvents as well as the uniform modification boding well the implants with irregular shapes. And it also doesn't consumption of active group, ensuring the long-term bioactivity of biomimetic peptide. What's more, we combined the metal ion (e.g., Zn²⁺) with bioactive peptide (e.g., BMP-2-derived peptide) using a mussel adhesion-mediated ion coordination and molecular clicking strategy. It not only ensures the long-term bioactivity of peptide, but also combine unique biological activities of the inorganic metal ions with bioactive peptide to meet the vary needs of biological materials.

Highlight:

In this context, we design a simple and biocompatible surface approach capable of efficient conjugating biomimetic peptide and metal ions for design of dual-functional bone implants with both osteoinductive and immunomodulatory functions to adapt mechanism of bone regeneration. **Firstly**, giving that previous works of surface modification on bone-

implants focus either on osteoinduction or immunomodulation, it is the first time to combine the molecule and ion dual-functions in the field of surface modification on bone-implants by mussel adhesion-mediated approach in our work. **In addition**, it is difficult to co-grafted zinc ions and BMP-2 peptide for the traditional methods^{1, 2, 3}. We combined the metal ion (e.g., Zn²⁺) with bioactive peptide (e.g., BMP-2-derived peptide) using a mussel adhesion-mediated ion coordination and molecular clicking strategy. It not only ensures the long-term bioactivity of peptide, but also combine unique biological activities of the inorganic metal ions with bioactive peptides to meet the various needs of biological materials. **Thirdly**, BMP-2-derived peptide or zinc ion could be replaced by other biomimetic peptides (e.g., VEGF, AMP) and metal ions (e.g., Cu²⁺, Mg²⁺) to synthesize varieties of multifunctional coatings for satisfying different clinical requirements.

To better understand the differences between ours and Pan's, we have added relevant content in the Part of introduction.

Changed in the revised manuscript (Page 5 Line 107-114, highlighted):

1. Introduction

In addition, the catechol residues on the surface enable not only simple conjugations of amino- or thiol-containing biomolecules via Michael addition or Schiff base reaction but also spontaneous coordination with bioactive metal ions^{43, 44, 45}. These advantages indicate the mussel-inspired surface strategy has the potential to co-modify bone implants with osteoinductive biomolecules and immunomodulatory active ions. However, the critical problem of this strategy is the random consumption of active groups (e.g., amino and thiol), which would interfere with the functions of conjugated biomolecules^{38, 46}. **Recently, Pan et.al¹ designed two mussel-derived biomimetic peptides for simple biomodification of Ti implants through robust catechol/titanium dioxide (TiO₂) coordinative interactions. The highly biomimetic peptides capped with RGD- or OGP-derived sequences could improve not only the biocompatibility of Ti implants but also the efficiency of osteogenicity, osseointegration and mechanical stability *in vivo*. The strategy provides a clear chemical binding on implants surfaces yet the uncontrolled biomolecular conjugation, particularly for multi-modification, which is not conducive to the reproducibility of a multi-bioactive surface.** Given this, we designed an improved biomimetic strategy by combining mussel-like adhesion with bioorthogonal click reaction, a specific and biocompatible chemistry^{47, 48, 49}.

Reference:

1. Pan G, *et al.* Biomimetic design of mussel-derived bioactive peptides for dual-functionalization of titanium-based biomaterials. *J Am Chem Soc* **138**, 15078-15086 (2016).
38. Yu H, *et al.* Nitric oxide-generating compound and bio-clickable peptide mimic for synergistically tailoring surface anti-thrombogenic and anti-microbial dual-functions. *Bioact Mater* **6**, 1618-1627 (2021).
46. Ding YH, Floren M, Tan W. Mussel-inspired polydopamine for bio-surface functionalization. *Biosurf Biotribol* **2**, 121-136 (2016).

Comment 3: The true novelty of this work is the simplicity- the authors could likely easily substitute the “clicked” BMP-2 with many other pharmaceutical analogues (e.g. VEGF, PTH, etc.), and other ions can be coordinated in place of Zn. Therefore, the translational value is quite high. However, like all devices, the regulatory hurdles will be significant as this is likely a class 3 device. It would benefit the wider audience if the authors could either discuss this aspect, or at least discuss what alternative agents they might use in place of BMP2, and in which applications they might employ their approach (e.g. poorly healing bone, osteoporosis, etc.).

Reply: Thank you for your evaluation and comments. The co-modified coating designed in our work was especially applicable for clinic situations like osteoporosis, diabetic and poorly healing bone. When BMP-2 and Zn²⁺ were substituted with other agents, different combinations will present more effects and benefit wider patients. For example, when BMP-2 is replaced by antimicrobial peptide (AMP), it will greatly improve antibacterial efficiency and be benefit to patients with bacterial infection. BMP-2 can also be substituted with vascular endothelial growth factor (VEGF), the potential effect of regenerative angiogenesis achieved from VEGF will benefit to patients with inadequate perfusion and impaired bone healing. Similarly, Zn²⁺ can be replaced by Cu²⁺, it could also strengthen the antibacterial efficiency.

According to your suggestion, we have added some discussion in the revised manuscript.

Changed in the revised manuscript (Page 27 Line 554-572, highlighted):

2. Results and discussion:

2.6 Osseointegration *In vivo*

The outlook of this study is that it provides a novel solution in a dual-functional implants with both osteoinductive and immunomodulatory activity for improving osseointegration by a mussel adhesion-mediated ion coordination and molecular clicking strategy to effectively improve mechanical fixation of the bone implants. This strategy involves combining the metal

ion (e.g., Zn^{2+}) with bioactive peptide (e.g., BMP-2-derived peptide) to overcome the shortcomings of the traditional methods. It not only ensures the long-term bioactivity of peptide, but also combines unique biological activities of the inorganic metal ions with bioactive peptide to meet the various needs of biological materials. Additionally, BMP-2-derived peptide or zinc ion could be replaced by other biomimetic peptides (e.g., VEGF, AMP) and metal ions (e.g., Cu^{2+} , Mg^{2+}) to synthesize varieties of multifunctional coatings for satisfying different clinical requirements. Although further exploration is still needed to understand the potential mechanisms of osteoimmunomodulation, these results have demonstrated a promising strategy towards bone regeneration and bone-implant osseointegration, which is in all probability utilized in future clinical practice and applied to orthopedic research. Furthermore, our mussel adhesion-mediated and molecular bioclickable strategy provides a favorable osseointegration approach to clinical applications in osteoporosis, diabetes, infection, and poor bone healing. The combination of inorganic metal ions with bioactive peptides and biomaterials will provide more opportunities for developing a new generation of engineering bone implants for orthopedic medicine.

Comment 4: There is a lack of context (in part due to poor introduction section) for their results. For example, how does the amount of bone growth (Figure 6B-D) the authors achieve compare to other publications, other approaches, or to simply administering BMP alone? What is the big picture take away for readers? Would this replace bone graft or calcium phosphate augmentation? How much BMP2 would you expect to “release”, or is this osteogenic effect only present at the screw/interface where a thin layer of cells/tissue benefit from the BMP-2 coating? Would the authors recommend their approach for poorly growing bone? Is it likely to increase the amount of bone growth (higher final density), or to simply accelerate the formation of bone (same final mineral density as mature bone, just appearing sooner)? Does the device actually integrate (osseointegration) with bone tissue or simply potentiate new/repaired trabecula to grow towards and inter-digitate into the screw threads?

Replay: Thank you for the comments. In order to answer the questions more clearly, we separate the questions and answer them point by point.

- a) For example, how does the amount of bone growth (Figure 6B-D) the authors achieve compare to other publications, other approaches, or to simply administering BMP alone?

Replay: Based on our review of other publications, we found that different amount of

bone growth was achieved with different modified approaches at different timepoints. We list representative results from some of these publications. However, these comparisons seem inclusive because of varied disease models and observation times. For example, Rahman et.al³² reported that BMP-2 with PLGA/PEG scaffolds into mouse calvarial model and achieved 55% of new bone area after 6 weeks of implantation. Zhao et.al³³ reported that silicon-doped titania nanotubes modified Ti screws enhanced new bone formation (BV/TV, 16%) at six weeks. In our approach, the Zn/BMP-2 co-modified bone implants achieved over 80% of new bone formed after 8 weeks of implantation which is several folds lower than the previously mentioned approaches. The probably due to the synergy of immunoactive Zn²⁺ and osteoinductive BMP-2 peptide, while a mono-modification (e.g., the Zn or BMP-2 group) might not provide the most favorable immunomodulatory microenvironment for bone regeneration.

Regarding to simply administering BMP alone, strategies often utilize soluble BMP-2 either physically adsorbed, covalently linked or encapsulated with biomaterials/implant to confer osteoinduction properties. However, depending on these strategies, there is two main issues, including impairing bioactivity and diffusing rapidly out of the biomaterials³⁵. What's more, high doses of BMP-2 have led to serious complications, such as ectopic bone formation, immunological response, and even tumorigenesis³⁶. These limitations hinder its wide application.

b) What is the big picture take away for readers?

Replay: The main idea of our paper was summarized in **Graphic abstract** to help readers clearly understand our work. Firstly, the clickable mussel-derived peptide was stably bound onto Ti screws via metal-catechol coordination. Then, an immunoactive metal zinc ion (Zn²⁺) capable of polarizing macrophages to the anti-inflammatory M2 phenotype was coordinated with the catechol residues to generate an immunoactive surface. For direct osteogenicity, a BMP-2-derived synthetic peptide capped with azido group was synthesized and conjugated with surface DBCO groups via a bioorthogonal cycloaddition chemistry. This Zn/BMP-2 co-modified coating would be more conducive to the regulation of macrophage phenotypic switch from M1 to M2, and the secretion anti-inflammatory cytokines would provide an optimum osteoimmunomodulatory microenvironment and lead to immuno-enhanced osteogenesis.

Graphic abstract: (A) Schematic illustration of the mussel-derived peptide for ion coordination and biomolecular click conjugation on a medical Ti screw. (B) In a bone implant model, the Zn^{2+} and BMP-2 peptide co-modified Ti screw shows osteoinductive and immunomodulatory dual functions in vivo, synergistically enhancing the interfacial osteogenesis and the intra-bone implant integration after implantation.

c) Would this replace bone graft or calcium phosphate augmentation?

Replay: Bone graft or calcium phosphate augmentation is used in bone defects in most cases^{37, 38}. While our co-modified coating is used in the bone-implant interface for improving osseointegration. They have a different application scope. Additionally, it is assumed that if the screw was changed by other substituent materials (e.g., 3D scaffolds), these substitutes may have the potential to replace bone graft or calcium phosphate in clinic as stabilizing structures for the injured bone, inducers of bone neof ormation and modulators of osteoimmunomodulatory.

d) How much BMP2 would you expect to “release”, or is this osteogenic effect only present at the screw/interface where a thin layer of cells/tissue benefit from the BMP-2 coating?

Replay: Generally, in this study, BMP-2 peptide stably grafted on the surface of titanium. And it doesn't have a clear release profile as zinc ion. Then, the durability of BMP-2 on the surface was evaluated by incubating the Zn/BMP-2 substrate in Dulbecco's modified Eagle's

medium/F12 (DMEM/F12, 37 °C) for 2 weeks. As shown in **Figure 1N**, the intensity of N 1s signal in XPS showed a slight decrease of less than 15 %. In addition, the durability of the coated clickable peptide labelled by a FITC probe was further checked to confirm the bioactivity. Despite of incubation in DMEM/F12 for 2 weeks, the intensity of fluorescence on the clickable peptide-modified TiO₂ surface (Zn/BMP-2 group) did not show significant reduction (**Figure S6**). Thus, it could be concluded that immobilized BMP-2 peptide is highly stable, probably due to the covalent bonding between DBCO group and azido group via bioorthogonal click chemistry. Therefore, we think this osteogenic effect majorly present at the screw/interface.

Figure 1. (N) Changes of N 1s signal in the XPS spectrum of the Zn/BMP-2 surface after incubated in DMEM for 2 weeks.

Figure S6. (A) Images of FITC-(2-Azido)-PEG₅-BMP-2 and Zn²⁺ co-modified surfaces:(green: (2-

Azido)-PEG₅-BMP-2) and (B) Intensity profile with regions of interest.

e) Would the authors recommend their approach for poorly growing bone?

Reply: Based on the answer to the first question above (**Significance 1**), we think the co-modified coating designed in our work was especially applicable for clinic situations like osteoporosis, diabetic and poorly healing bone.

f) Is it likely to increase the amount of bone growth (higher final density), or to simply accelerate the formation of bone (same final mineral density as mature bone, just appearing sooner)?

Reply: According to the results of Micro-CT (Figure 5B), it not only increased the amount of bone growth (BV/TV) but also improved final mineral density (BMD) compared with our control group (TiO₂).

Figure 5B Quantitatively evaluating the peri-implant bone generation according to the BMD, BV/TV, Tb.Sp, Tb.Th and Tb.N.

g) Does the device actually integrate (osseointegration) with bone tissue or simply potentiate new/repaired trabecula to grow towards and inter-digitate into the screw threads?

Reply: Actually, our device integrates (osseointegration) with bone tissue instead of simply potentiating new/repaired trabecula to grow towards and inter-digitate into the screw threads. Currently, screw implants for clinic use are mostly designed with porous structure for bone ingrowth, and neovascularization^{39, 40}. Due to its clinically important advantages, we adopted this kind of screws in our research. Hence, this open pore structure can create the opportunity for the new/repaired trabecula to ingrown and to interlock with the material.

Data

Comment 1: In supplementary figure S3 it is unclear how strong the osteogenic response is. Please include picture of control well (cells + osteogenic media without BMP). Including the control picture should be standard, and the reader should easily be able to exclude the possibility of pathological cell behavior (e.g. false positive from overactive osteogenesis,

which can only be determined by comparing treated to controls). The authors do a good job of showing this in Figure 4C, but it feels incomplete in Figure S3. It would also help the reader if you had a quantitative analysis- approximately how much more did the BMP2 peptide stimulate ALP, or Alizarin-chelated mineralization, and how much did the conjugated BMP-2 peptide stimulate ALP/Alizarin binding? In the results section this information should be written at the point where Figure S3 is first mentioned.

Reply: Thank you for the comments. We showed the pictures of our concurrent control well to help the readers clearly understand how strong the osteogenic response is. Then, we added a quantitative analysis on the stimulated ALP using the percentage of positive cells, and optical density to quantitate Alizarin-chelated mineralization. It showed that our conjugated BMP-2 peptide presented obvious osteogenic response. The conjugated BMP-2 peptide and rhBMP-2 we used to stimulate ALP/Alizarin in our study were both 50 ng/ml. So, we have replaced the original picture (**Figure S3**), the new image is as follows.

Changed in the revised supplementary information (Page 3, highlighted):

Figure S3. (A) ALP staining and (B) Alizarin Red S (ARS) staining of BM-MSCs cultured in osteogenic medium supplemented with rhBMP-2 or (2-Azido)-PEG₅-BMP-2; (C, D) quantitate results of ALP and ARS staining.

Comment 2: SEM/EDS: EDS is typically only semi-quantitative, unless using very rigorous methodology and calibration standards. Please include the standard deviation and number of measurements for all EDS results in the text.

Reply: Thank you for the comment and reminding. EDS in our lab is generally calibrated for quantification using elemental standards and followed general rules of analytical methods for scanning electron microscope in China. As you mentioned, variations between analysis tools can lead to values that differ considerable, so we have included the standard deviation for all EDS results as follows to eliminated errors inherent in EDS quantification. Element scanning and mapping were conducted on the surface for at least 25000 counts. We calculated the results of elemental composition of Zn/BMP-2 co-modified coating based on the EDS analysis. Three different samples from each group were used for testing. The result is as follows.

Table Semi-quantitative results of elemental composition based on EDS analysis (n=3, independent sample)

Element	Atomic (%)				
	TiO ₂	TiO ₂ -DBCO	Zn	BMP-2	Zn/BMP-2
C	7.69±0.06	9.53±0.68	9.55±1.46	10.75±0.12	10.84±0.88
O	48.54±0.45	46.65±0.59	46.04±0.65	44.94±0.56	44.38±0.98
Ti	7.79±0.2	6.55±0.20	6.66±0.89	6.7±0.62	4.75±0.14
Si	35.98±0.58	35.20±0.78	34.31±0.71	34.37±0.7	35.75±0.60
N	--	2.07±0.25	2.37±0.32	3.24±0.42	3.22±0.37
Zn	--	--	1.07±0.23	--	1.06±0.09

Comment 3: Figure 1O: In the results text please explicitly state what the minimum concentration needed to elicit a positive response, *in vitro* and *in vivo*, so the reader can immediately understand the significance of your release concentrations (based on prior literature). Ideally, in Figure 1O, put a baseline (dotted or dashed line) at the PPM value of Zn that elicits a relatively well known *in vitro* response, and one for *in vivo* response (e.g. concentration where >40% of culture Raw cells would be expected to convert to M2 cytokine secretion). At this point in the text the reader may ask, is releasing 0.04 PPM of Zn as steady state good enough to elicit an *in vitro*, and an *in vivo* change?

Reply: Thank you for the comments. Based on prior literatures, 1.25×10^{-6} M (0.08 ppm)

zinc is enough to decrease the expression of pro-inflammatory cytokines; however, a higher Zn^{2+} amounts (more than 6.5 ppm) could increase in the number of TRAP-positive cells ⁴¹, even evoke immune diseases. Therefore, the range of zinc amount conducive to macrophage differentiation is between 0.08-6.5 ppm. We put a dotted line at 0.08 ppm on the release **Figure S7** to point out the concentration that may elicit an immunomodulatory *in vitro* response.

Figure S7 Zn^{2+} release profiles of the Zn/BMP-2 surface in PBS solution, FBS-free DMEM and 10% FBS-DMEM. (A) non-accumulative Zn^{2+} release; (B) accumulative Zn^{2+} release.

According to the above description, releasing 0.04 ppm of Zn^{2+} is not good enough to elicit an immune response *in vitro*, and an *in vivo*. The y-axis of Fig.10 disappeared, leading to a misunderstanding. In our present study, zinc concentration was 0.145 ppm on the first day, and the total zinc ion released from Zn/BMP-2 dual-effect coating was 0.575 ppm (ug/ml) after 28 days' incubation (Figure 10), which is much lower than 6.5 ppm. Therefore, in our study, zinc ions at the concentration of 0.145 ppm (>0.08 ppm) could elevate the proliferation of both macrophage and BM-MSCs (Figure 2), activate the macrophage phenotype switch from M1 to M2 (Figure 3) and enhance the osteogenic differentiation (Figure 4).

Comment 4: The authors should make a supplementary figure showing representative force/displacement curves (Figure 6F), ideally superimposing all curves from each group, in a separate graph for each group. That way the reader can get more information about the mechanical behavior during pull-out, including stiffness, proper failure mechanics, etc. This does not require additional testing- you should already have this data saved in your mechanical test machine, and it should only require exporting, and replotting in excel/origin/or any software.

Reply: Thank you for the comment and reminding. As you required, we exported this data of the pull-out curve from mechanical test machine as follows. The maximal pull-out

forces in the Zn, BMP-2 and Zn/BMP-2 groups were all significantly improved as compared with the TiO₂ control, The Zn/BMP-2 group showed the highest maximum pull-out force.

Changed in the revised supplementary information (Page 15, highlighted):

Figure S15. Biomechanical pull-out testing curves of different peptide-treated and untreated Ti screws (n=3, independent sample per group).

Methodology

Comment 1: In the cell culture, please explicitly describe how you selected for BM-MSCs from the many cell populations collected from the bone marrow. Did you use magnetic beads? Was it a mixed population of cells? How did you confirm the purity, using FACS? The methodology in this section is lacking. Please provide more detailed information, and if this is a routine or common procedure in your laboratory, please reference/cite your prior work so the reader can get some idea of the purity of your MSC population.

Reply: Thank you for the comment and reminding. Media selection was responsible for the isolation of a nearly homogeneous population of BM-MSCs without magnetic beads⁴². The BM-MSCs immunophenotype and purity were confirmed to be CD29⁺, CD90⁺, CD34⁻ and CD45⁻ using FACS. The results were as follows:

Figure S8. Immunophenotypic characterization of BM-MSCs. FACS results showed that these cells were homogenously positive for mesenchymal markers CD29 and CD90; negative for hematopoietic markers CD34 and CD45.

The methodology and results of this section are added in the corresponding part in our revision.

Changed in the revised manuscript (Page 31 and 11, highlighted):

4. Experimental Section (Page 31 Line 644-663, highlighted)

Cell Culture

Bone marrow-derived mesenchymal stem cells (BM-MSCs) were isolated from the 4-week-old male Sprague Dawley (SD) rats (Shanghai Jihui Experimental Animal Center, Shanghai, China) according to a previous protocol ⁶⁶. All animal experiments were approved by the Animal Research Committee of Shanghai Jiaotong University School of Medicine. Briefly, the femur and tibia were collected and separated from muscle and connective tissue. After cutting off both ends of the bone, the bone marrow suspensions were flushed out and suspended in Dulbecco's modified Eagle's medium: F-12 (DMEM/F12, HyClone) containing 10% FBS (Gibco) and 100 U/ml of penicillin/streptomycin. The cell suspension was filtered by 70 μ m filters (Millipore, Ireland). The cells were incubated at 37°C under 5% CO₂ atmosphere, the medium was refreshed every 2-3 days. When the cells arrived at 80%-90% confluence, the cells were detached from the culture dish by 0.25% trypsin/EDTA. BM-MSCs were evaluated by flow cytometry in identifying the surface specific markers and confirming their purity before any experiment *in vitro*. Briefly, passage 3 BM-MSCs were suspended in PBS (pH=7.2) at a density of 1×10^6 cells/ml and then were stained with FITC anti-rat/mouse

CD90 (BioLegend, 202503), PE anti-mouse/rat CD29 (BioLegend, 102207), FITC anti-rat CD45 (BioLegend, 202205) and APC mouse anti-rat CD34 (Novus, NB600-1071) flow cytometry antibodies at 4°C for 30 min. Cells without any staining was used as a negative control. Cells were washed in PBS and stained with 7-amino-actinomycin D (7AAD, 559925, Biosciences, BD) according to the manufacture instructions. Quantitative fluorescence analysis was performed with a flow cytometer (LSRFortessa™ X-20, BD, USA), and FlowJo™ software (Version 10.7.1) was used to quantified the expression levels of surface markers.

2. Results and discussion (Page 11 Line 250-255, highlighted)

2.2. Surface Cytocompatibility *In vitro*

After passage 3, the cells were detached with trypsin, and then the cell surface markers were determined by flow cytometry (LSRFortessa™ X-20, BD, USA) analysis to identify the purity of BM-MSCs. As shown in Figure S8, high expressions of CD29 (99.8%) and CD90 (98.9%) and extremely low expressions of the hematopoietic marker CD45 (1.7%) and CD34 (3.1%) were detected, indicating the high purity of BM-MSCs as well as the feasibility of employing these cells in following studies.

Figure S8. Immunophenotypic characterization of BM-MSCs. FACS results showed that these cells were homogenously positive for mesenchymal markers CD29 and CD90; negative for hematopoietic markers CD34 and CD45.

References:

66. Zhu H, *et al.* A protocol for isolation and culture of mesenchymal stem cells from mouse compact bone. *Nat Protoc* **5**, 550-560 (2010).

Comment 2: Cytocompatibility Figure 2- how long did you let cells attach, did you wash off unattached cells?

Reply: Thank you for the comment and reminding. BM-MSCs and RAW264.7 cells were separately cultured on samples with different surface treatments for 24 hours and gently rinsed with sterilized PBS solution for 3 times. We have added these details in the section of methodology.

Changed in the revised manuscript (Page 32 Line 665-666, highlighted):

Methods:

Cytocompatibility:

BM-MSCs and RAW264.7 cells were separately cultured on samples with different surface treatments for 24 hours and gently rinsed with sterilized PBS solution for 3 times. A Live/Dead cell staining kit (Yeasen, China) consisting of Calcein-AM (green fluorescence) and Propidium iodide (PI, red fluorescence) was used to assess the cell viability. The fluorescent images were acquired by a fluorescence microscope (Nikon, Japan).

Comment 3: Osteogenic differentiation- just to clarify, did you collect a large pooled stock of conditioned media, freeze it, and thaw/use every 2-3 days during the differentiation process? Did you centrifuge it and filter through a 0.2 um filter to ensure that no large cell debris carried over? Mention these details!

Reply: Thank you for the comment and reminding. We are sorry for our ambiguous description. In this work, the supernatant of different groups was collected after centrifuge at 800 rpm for 5 min to move cells and frozen at -80°C for further use. The supernatant, removed the debris with a 0.22 µm filter (Millipore, USA), was mixed with fresh DMEM/F-12 medium at a ratio of 1:2 to obtain macrophage conditioned medium (MCM). We have added these details in the section of methodology.

Changed in the revised manuscript (Page 34, line 722-726 highlighted):

Method:

Osteogenic differentiation in vitro:

To investigate whether Zn²⁺-loaded substrates could affect the differentiation of BM-MSCs through modulating the polarization of macrophage, the supernatants of RAW264.7 cells cultured on different surfaces in inflammatory conditions was collected. **Then, the supernatant was centrifuged at 800 rpm for 5 min to move any remained cells and frozen at -80° C for**

further use. In addition, the supernatant filtered with a 0.22 μm filter (Millipore, Ireland) was mixed with fresh DMEM/F-12 medium at a ratio of 1:2 to obtain macrophage conditioned medium (MCM).

Comment 4: In the microCT section the methods information is lacking, and this is quite troubling! What settings did you use: voxel size? source voltage? current? Filter material and thickness??? exposure time?? did you frame average? Did you use rotation during scanning? What threshold settings did you use to differentiate between bone and metal? What reconstruction software did you use, and what version of software? How can others considering reproducing or building on your work without this kind of information? This is very important because it is easy to unintentionally reach erroneous conclusions on how much bone has formed with incorrect thresholding, especially if the metal screw cause artifacts (very common). Usually, you cannot correctly threshold/differentiate any closer than 10-100 μm from the screw surface, but as a reader I cannot even determine whether the results are correct/reliable, because you have not provided the resolution/voxel size! In authors figures, Supplementary Figure 6-B3, it very clearly appears that there is some artifacts arising from the screw. The authors must include a new image in S.Figure 6, “B4” showing a representative image (3D and 2D reconstruction) that indicates any artifacts, and shows exactly their choice of thresholding and where the delineation of bone/screw occurs. Please include in that image a scale bar that is the VOXEL size. This does not require any new experiments.

Reply: Thank you for the comment and reminding. We have added these details in the revised manuscript.

In this study, to minimize artifacts, we selected appropriate scanning parameters (threshold for new bone=50-80) and using artifact reduction tools (NRecon, Bruker, Kontich, Belgium) available in the image reconstruction step (**Figure A in the blow**). Figure S12(B4) show the ideal conditions, without ring artifact. On the 2D tomogram shows exactly where the delineation of bone occurs. Semi-automated image segmentation was used to define the boundary where new bone occurs (yellow) on a 2D tomogram. The volume of interest is outside the boundary. We also presented 3D rendering of the entire volume of interest and a corresponding longitudinal cut-away view for a representative specimen.

Figure A. (a1-a2) Image showed the selected appropriate scanning parameters (threshold for new bone=50-80); (a3) The 3D Images were reconstructed using NRecon software (Version1.7.3.0, Bruker, Kontich, Belgium) with correction for misalignment and ring artefacts.

Figure S12. (B4) Upper: Semi-automated image segmentation was used to define the boundary where new bone occurs (yellow) on a 2D tomogram. The volume of interest is outside the boundary; Lower: 3D rendering of the entire volume of interest and a corresponding longitudinal cut-away view for a representative specimen.

Changed in the revised manuscript (Page 35, 36 Line 769-783, highlighted):

Methods:

Micro-CT evaluation:

After 8 weeks of implantation, the rats (n=5 per group) were sacrificed. The femurs were isolated and fixed in 70% alcohol for 3 days. Then, those femurs were scanned by a Micro-CT system (SkyScan1172 Ex-Vivo Micro-CT, Belgium) with a rotation step of 0.15°, a pixel size of 17.81 μm , 80 kV source voltage, 112 μA source current, 0.5 mm Al filter optimizing the contrast, and 370 ms exposure time. Multilevel thresholding procedure (threshold for new bone=50-80, threshold for implant=130) was applied to distinguish bone from other tissues. The volume of interest (VOI) included the trabecular compartment between the outer diameter and inner diameter from the longitudinal axis of the screw. Specifically, the VOI was selected in an axisymmetric cuboid with a circular plane 2 mm ($d=17.81 \mu\text{m} \times 113 \text{ layer}$) from the top view (B3 in **Figure S12**) and a depth of 6 mm ($L=17.81 \mu\text{m} \times 340 \text{ layer}$) along the longitudinal axis of the screw (B1 and B2 in **Figure S12**). The 3D images were reconstructed by NRecon software (Version1.7.3.0, Bruker, Kontich, Belgium) with correction for misalignment and ring artefacts. Diversity index of bone regeneration including bone tissue volume/total tissue volume (BV/TV), bone mineral density (BMD), trabecular thickness (Tb.Th), trabecular number (Tb.N), and trabecular separation (Tb.Sp) were calculated by supporting analyzing software (CTAn, Bruker, Kontich, Belgium).

Figure S12. (A) Scheme for implantation surgery for *in vivo* tests and treatment process; (B) At each 3D location (shown as cross-hair), three orthogonal views are retrieved from the whole dataset and shown as (B1) coronal view (the normal images, in x-y plane), (B2) transaxial view (x-z plane), (B3) sagittal view (z-y plane); (B) At each 3D location (shown as cross-hair), three orthogonal views are retrieved from the whole dataset and shown as (B1) coronal view (the normal images, in x-y plane), (B2) transaxial view (x-z plane), (B3) sagittal view (z-y plane). (B4) Upper: Semi-automated image segmentation was used to define the boundary where new bone occurs (yellow) on a 2D tomogram. The volume of interest is outside the boundary; Lower: 3D rendering of the entire volume of interest and a corresponding longitudinal cut-away view for a representative specimen.

Analytical approach and statistics:

Comment 1: No comments, authors were rigorous and comprehensive in their analytical approach.

Suggested improvements: See details provided in other sections.

Reply: We thank the reviewer for the positive response.

Clarity and context:

Comment 1: The results and discussion text was clear and concise, but lacked a broader context (e.g. compare to the results of other publication, similar models, use of same drug via alternative methods, etc.). The introduction and abstract must be revised. The conclusion should also be revised.

Reply: Thank you for the comments. According to your suggestions, some broader context has been added in the sections of results and discussion. In addition, the abstract, introduction and conclusion have also been comprehensively revised in the manuscript. And the abstract and introduction are showed in the **comment 1** of the abstract and the **comment 1, 4, 5 and 6** of introduction in the section of *Specific suggestions*, respectively. And the conclusion has been revised in the manuscript as follows:

Changed in the revised manuscript (Page 29 Line 582-593, highlighted):

3. Conclusion:

In summary, we here reported a dual-effect coating on bone implants with both immunomodulatory and osteoinductive activities by a mussel adhesion-mediated ion coordination and molecular clicking strategy. The strategy could provide a simple method for co-modification of titanium bone implants with immunoactive metal ions (e.g., Zn^{2+}) and osteoinductive growth factors (e.g., BMP-2 peptide). The Zn^{2+} and BMP-2 peptide co-modified implants could elicit a favorable osteoimmune microenvironment by macrophage switch from M1 to M2 phenotypes that facilitates the osteogenic differentiation of BM-MSCs, thus enhancing osseointegration at the bone-implant interface and improving their mechanical stability in vivo. Overall, the dual-effect coating could be utilized to regulate macrophage phenotypic conversion and create a favorable immunomodulatory microenvironment for bone regeneration and osseointegration, providing a new idea of bone tissue engineering implants with immunoactivity and osteoinductivity.

References:

The references are sufficient.

Specific suggestions

• Abstract

Comment 1 “We anticipate this study would provide new ideas and solutions for engineering implants with immunoactivity and tissue inductivity to precisely adapt tissue regeneration microenvironment.” Strongly recommend avoiding such broad claims, especially in the abstract. The abstract is your one chance to give specific information BEFORE readers will read your full manuscript. The prior text in the abstract suggest you are using well understood approaches to improve the material properties (e.g. releasing bioactive or immunomodulatory ions), how does your work actually provide a new idea, or solution? Explain, specifically, how it does that. For example, you use click chemistry- but click chemistry is routinely employed to make new materials. How, specifically, have you created a new solution and idea using click chemistry?

Reply: According to the reviewer’s kind suggestion, we have revised the abstract as shown below.

Changed in the revised manuscript (Page 2 line 20-35, highlighted):

Abstract

Immune action and new tissue formation are two distinct but overlapping stages involved in tissue regeneration process. However, current biomaterial designs are mostly trapped into a one-sided consideration with either modulating immune response or inducing the formation of new tissues, particularly in biomodification. Here, a dual-effect coating with immunomodulatory and osteoinductive activities was designed by mussel adhesion-mediated ion coordination and molecular clicking strategy. This biomimetic surface engineering strategy provided a simple method to co-modify bone implants with immunoactive metal ions (e.g., Zn^{2+}) and osteoinductive growth factors (e.g., BMP-2 peptide). Compared to the bare TiO_2 group, Zn^{2+} could increase M2 macrophage recruitment by up to 92.5% *in vivo* and upregulate the expression of M2 cytokine IL-10 by 84.5%; while the dual-effect of Zn^{2+} and BMP-2 peptide could increase M2 macrophages recruitment by up to 124.7% *in vivo* and upregulate the expression of M2 cytokine IL-10 by 171%. These benefits eventually significantly enhanced bone-implant mechanical fixation (203.3 N) and new bone ingrowth (82.1%)

compared to the bare TiO₂ (98.6 N and 45.1%, respectively). Taken together, the dual-effect coating could be utilized to synergistically modulate osteoimmune microenvironment at the bone-implant interface, enhancing bone regeneration for successful implantation.

• **Introduction**

Comment 1: “Previous considerations for bone implants, however, mostly took a one-sided approach by either to minimize the immune actions or to induce direct osteogenesis at the bone-to-implants interfaces”.

Reply: Thank you for the comment. We have rewritten this part in the text as follows.

Changed in the revised manuscript (Page 3 Line 49-55, highlighted):

Traditional studies regarding bone implants predominantly focused on optimizing the osteogenic capacity, with some inert bone implants designed to evade immune response and others introduced with various bioactive moieties (e.g. peptides, growth factors, protein and even ions) for promoting osteogenesis *in vitro* and bone-to-implant osseointegration^{9, 10, 11}. However, these implants may not completely adapt to the *in vivo* microenvironment, thus leading to some inconsistent results *in vivo*¹² majorly due to the uncontrolled local immune responses triggered by exogenous biomaterials.

Comment 2: Did the authors do a pull-out test or push-out mechanical test? In one part they mention “push out” testing. Re-read the manuscript and check for these errors!

Reply: Thank you for reminding. In fact, I have done a pull-out testing, instead of push-out testing. We have revised them in the text.

Comment 3:

• Rewriting suggestions:

I hesitate to make specific suggestions as this is a personal choice, but please consider my suggestions below on how to improve your abstract and introduction sections. These are optional and not “required” revisions, though these text sections must be revised and improved in some way.

First in abstract:

- Why did you invent this “device”, try to describe a specific clinical problem that would benefit (if possible).....

- Specify the benefits (results) of your device- e.g. “Zn ion release increase M2 macrophage recruitment by up to X% *in vivo*, expression of M2 cytokine IL-10 by Y%, and increased total bone volume by Z%, while dual functionalized implants containing both Zn and BMP2 increased M2 macrophage recruitment by up to X% *in vivo*, expression of M2 cytokine IL-10 by Y%, and increased total bone volume by Z%. Similar benefits were also observed *in vitro*.” You only need to pick your most important result, in this case bone volume, mechanical strength, and inflammatory cell recruitment or expression, *in vivo*.

Reply: Thank you for the comment. We picked the most important results and presented this information in the section of abstract based on your suggestion. And the abstract is illustrated in the **comment 1** of the *Specific suggestions*.

Comment 4:

In the introduction:

- In the first sentence the authors introduce the most relevant work, Pan et al. but do not compare or contrast their work. The reader loses out of a lot of potential information. I recommend restructuring the introduction like so:
- What is the problem you seek to address? You write that orthopedic materials can fix some problems, but are not tailored to the specific needs of each clinical case (e.g. modulating the inflammatory response). Is this really a problem? You do not make clear to the reader that this is an unmet need/problem by citing literature that shows X number of patients could benefit from shifting the immune response towards M2.

Reply: Thank you for your suggestion. Clinically, while the integration of implants and bone tissues has a success rate of over 95%^{43,44}, there still exist problems of poorly healing bone and bone-implant integration failing. The rate of early implant failure was about 1.2% due to the development of fibrous tissue between the implants and the surrounding bone tissues in the healing period due to lack of appropriate osteoimmunology⁴⁵. This co-modified strategy reported in our paper can reduce the incidence of early implant failure because of its outstanding functions of immunoactivity and osteoinductivity.

Comment 5:

- Your assertions are broad. “These evidences indicated the two-sidedness of immune actions, in which macrophages and other immune cells can not only clear cell debris, combat microbes, activate inflammation and promote fibrosis, but also coordinate tissue healing processes by activating stem/progenitor cells and remodeling extracellular matrix for regeneration.” This is

informative, but a slightly improved statement might say something like, “Macrophages and other immune cells serve dual roles, activating inflammation and promoting fibrosis (M1 response?), but also coordinating tissue healing processes by activating stem/progenitor cells and remodeling extracellular matrix for regeneration (M2 response?).” Since your whole design relies on these 2 response type dichotomy, you should define and explain them, and why you want to trigger an M2 or shift from M1 to M2. That is unclear.

Reply: Thank you for your suggestion. According to your suggestions, we have revised the unclear statement in corresponding part and listed as follows.

Changed in the revised manuscript (Page 3 Line 62-70, highlighted):

Introduction:

Recent studies on osteoimmunology further revealed that immune microenvironments also play an important role in bone tissue formation^{7, 13, 14}. **In different microenvironments, macrophages polarize into classically activated macrophages (M1) or alternatively activated macrophages (M2). The pro-inflammatory M1 macrophages activate inflammation and promote fibrosis while the pro-healing M2 macrophages coordinate tissue healing processes by activating stem/progenitor cells and remodeling extracellular matrix for regeneration^{15, 16}. As is found in previous studies, an efficient and timely switch from M1 to M2 macrophage phenotype was essential for bone healing and osteointegration around bone implants, creating a favorable osteoimmune environment via the increased production of anti-inflammatory (e.g., IL-10) and pro-osteogenic (e.g., BMP-2 and VEGF) cytokines¹⁷.**

References:

15. Takayanagi H. Osteoimmunology in 2014: Two-faced immunology-from osteogenesis to bone resorption. *Nat Rev Rheumatol* **11**, 74-76 (2015).
16. Oishi Y, Manabe I. Macrophages in inflammation, repair and regeneration. *Int Immunol* **30**, 511-528 (2018).
17. Zhu Y, *et al.* Regulation of macrophage polarization through surface topography design to facilitate implant-to-bone osteointegration. *Sci. Adv.* **7**, eabf6654 (2021).

Comment 6:

• In paragraph 3 and 4 this would be a perfect place to discuss how your approach differentiates from Pan et al., and other studies in the field.

Reply: Thank you for your suggestion. According to your suggestions, the differences between Pan’s and our works are discussed in the section of introduction which also be listed below.

Changed in the revised manuscript (Page 5 Line 107-114, highlighted):

Introduction:

Likewise, catechol-rich poly(dopamine) can also achieve robust molecular adhesion to virtually all kinds of substrates⁴². In addition, the catechol residues on the surface enable not only simple conjugations of amino- or thiol-containing biomolecules via Michael addition or Schiff base reaction but also spontaneous coordination with bioactive metal ions^{43,44,45}. These advantages indicate the mussel-inspired surface strategy has the potential to co-modify bone implants with osteoinductive biomolecules and immunomodulatory active ions. However, the critical problem of this strategy is the random consumption of active groups (e.g., amino and thiol), which would interfere with the functions of conjugated biomolecules^{38,46}. Recently, Pan et.al¹ designed two mussel-derived biomimetic peptides for simple biomodification of Ti implants through robust catechol/titanium dioxide (TiO₂) coordinative interactions. The highly biomimetic peptides capped with RGD- or OGP-derived sequences could improve not only the biocompatibility of Ti implants but also the efficiency of osteogenicity, osseointegration and mechanical stability *in vivo*. The strategy provides a clear chemical binding on implants surfaces yet the uncontrolled biomolecular conjugation, particularly for multi-modification, which is not conducive to the reproducibility of a multi-bioactive surface. Given this, we designed an improved biomimetic strategy by combining mussel-like adhesion with bioorthogonal click reaction, a specific and biocompatible chemistry^{47,48,49}. We hypothesize this strategy which involves a bioclickable way for biomolecular conjugation and a coordination means for ion loading would provide a promising solution for surface engineering of osteoinductive and immunomodulatory bone implants.

Reference:

1. Pan G, *et al.* Biomimetic design of mussel-derived bioactive peptides for dual-functionalization of titanium-based biomaterials. *J Am Chem Soc* **138**, 15078-15086 (2016).
38. Yu H, *et al.* Nitric oxide-generating compound and bio-clickable peptide mimic for synergistically tailoring surface anti-thrombogenic and anti-microbial dual-functions. *Bioact Mater* **6**, 1618-1627 (2021).
46. Ding YH, Floren M, Tan W. Mussel-inspired polydopamine for bio-surface functionalization. *Biosurf Biotribol* **2**, 121-136 (2016).

Reviewer #3 (Remarks to the Author):

General comments:

In this manuscript, the Ti substrate coated with Zn²⁺ and BMP-2 peptide was used to achieve dual functions of (1) immunomodulation and (2) osseointegration. The study introduced commonly used chemical surface coating method of mussel-mediated chemistry, but the difference was emphasized by using bioorthogonal click reaction to solve the random consumption of active groups on peptides (e.g., amino and thiol). Despite advantages, some of results and discussion needs supportive experiments or further explanation. Please find the detail comments.

Reply: We appreciate the Reviewer's suggestion. According to your suggestions, we have revised the manuscript and the detailed changes are shown below.

Major comments:

Comment 1: I am not convinced with the statement of “the current biomaterial design is trapped into a one-sided consideration with either focusing on the regulation of immune response or paying attention to induction of new tissue formation.”. Previously, several approaches already endeavored to address both immune response and bone formation at the same time for natural tissue-like bone regeneration.

Reply: Thank you for the comment and reminding. Based on further systematic literature review on current approaches, we have rewritten the section of abstract in the revised paper. The revised version is as follows.

“However, current biomaterial designs are mostly focus on unilaterally either modulating immune response or inducing the formation of new tissues particularly by biomodification.”

Changed in the revised manuscript (Page 2 Line 21-23, highlighted):

Abstract:

“However, current biomaterial designs are mostly trapped into a one-sided consideration with either modulating immune response or inducing the formation of new tissues, particularly in biomodification.”

Comment 2: Recently, a number of studies on osteoimmunomodulation have been developed by immobilizing osteoinductive factors and immunomodulatory factors at the same time (e.g. growth factors, polyphenols, metal ions). In contrast to those, the significant difference of this

study is not clear.

Reply: Thank you for the comment and reminding. We have searched the relevant literatures based on the keywords “osteoinductive”, “immunomodulatory”, “growth factor”, and “metal ions”. Three representative literatures were selected for comparison.

(1). *Zhao D-W, et al. Interleukin-4 assisted calcium-strontium-zinc-phosphate coating induces controllable macrophage polarization and promotes osseointegration on titanium implant. Mater Sci Eng C 118, 111512 (2021).*

In this paper, a calcium-strontium-zinc-phosphate (CSZP) coating was fabricated on a Ti implant surface by phosphate chemical conversion (PCC) technique, which modified the surface topography and element constituents. Then, they envisioned an accurate immunomodulation strategy via delivery of interleukin (IL)-4 to promote CSZP-mediated bone regeneration.

(2). *Chen L, et al. Synergistic effects of immunoregulation and osteoinduction of ds-block elements on titanium surface. Bioactive Materials 6, 191-207 (2021).*

In this work, three ds-block elements, Zn, Cu, and Ag, were introduced on the titanium surface by PIII method to investigate their immune response of macrophages, their osteogenic differentiation of BMSCs, and further study the correlation between macrophages and mBMSCs on sample surfaces.

(3). *Bai J, et al. Biomimetic osteogenic peptide with mussel adhesion and osteoimmunomodulatory functions to ameliorate interfacial osseointegration under chronic inflammation. Biomaterials 255, 120197 (2020).*

In this work, the mussel-inspired peptide ((DOPA)₄-S₅-YGFGG) with adhesive and osteogenic properties is designed and applied to titanium implants. The biomimetic peptide coating provides a favorable milieu that can inhibit the inflammatory response of activated M1 macrophages, synergistically regulate bone immunomodulation, restore the balance between osteogenic differentiation and osteoclast activation, and improve osseointegration of the implants significantly.

In the first article, IL-4 was injected through the skin directly over the scaffolds, which was difficult to ensure the long-term bioactive of IL-4. In the second article, their strategy does not take growth factors or biomimetic peptides into account. And it was difficult to graft growth factors or biomimetic peptides on the surface of bone implants. In addition, their strategies, that were in the first and second article, involve tedious chemical reactions as well as sophisticated surface treatment technologies. The complex procedures make them to be hardly applied for multicomponent modification due to low controllability and poor operability.

In the third article, the critical problem of this mussel-inspired strategy is the random consumption of active groups (e.g., amino and thiol), which would impede the functions of conjugated biomolecules^{2, 7}, leading to be unable to display long-term bioactivity. What's more, they do not combine with the unique biological activities of the inorganic metal ions (e.g., Zn^{2+} , Mg^{2+} , Cu^{2+}) into their system.

In this context, we design a simple and biocompatible surface approach capable of efficient conjugating biomimetic peptide and metal ions for design of dual-functional bone implants with both osteoinductive and immunomodulatory functions to adapt mechanism of bone regeneration. Our paper highlights three major novelties that distinguish from other studies. **Firstly**, giving that previous works of surface modification on bone-implants focus either on osteoinduction or immunomodulation, it is the first time to combine the molecule and ion dual-functions in the field of surface modification on bone-implants by mussel adhesion-mediated approach in our work. **Secondly**, it is difficult to co-grafted zinc ions and BMP-2 peptide for the traditional methods^{1, 2, 3}. In our study, we combined the metal ion (e.g., Zn^{2+}) with bioactive peptide (e.g., BMP-2-derived peptide) using a mussel adhesion-mediated ion coordination and molecular clicking strategy. It not only ensures the long-term bioactivity of peptide, but also combine unique biological activities of the inorganic metal ions (Zn^{2+}) with BMP-2 peptide to meet the vary needs of biological materials. **Thirdly**, BMP-2-derived peptide or zinc ion could be replaced by other biomimetic peptides (e.g., VEGF, AMP) and metal ions (e.g., Cu^{2+} , Mg^{2+}) to synthesize varieties of multifunctional coatings for satisfying different clinical requirements. In a word, our study provides a promising solution for engineering implants with immunoactivity and tissue inductivity to precisely adapt tissue regeneration microenvironment.

Comment 3: Why did the authors choose ' Zn^{2+} ' as an immunomodulatory factor? Excluding the other factors such as Sr^{2+} or immune cytokines, which were addressed in the introduction section. The authors should elaborate the function of Zinc ion.

Reply: Thank you for the comment and reminding. Among these trace metallic ions, zinc is a nutritionally essential trace element for some key enzymes and transcription factors and is well documented to be indispensable for the development of the adaptive immune system^{46, 47}. A clinical study reported that patients with inflammatory diseases (e.g., rheumatoid arthritis, RA) have been found to be low serum levels of zinc and a corresponding increased TNF- α production; the process can be reversed by the supplementation of zinc⁴⁸. Besides, some previous reports have already suggested that zinc exerts modulatory effects on

macrophage phenotype from M1 to M2, inducing anti-inflammatory responses and inhibiting the pro-inflammatory. Therefore, the addition of zinc has a potential positive effect on osteoimmunomodulation.

Although zinc ions play an important and beneficial role in cell behaviors and immune functions, high doses of Zn^{2+} could be detrimental to osteoblast differentiation, disruption of mineralization process, even may evoke immune diseases⁴⁶. However, a high dose of Zn^{2+} could be detrimental to osteoblast differentiation, disruption of mineralization process, even may evoke immune diseases. A study reported that 1.25×10^{-6} M (0.08 ppm) zinc is enough to decrease the expression of pro-inflammatory cytokines; however, a low concentration of Zn^{2+} (less than 6.5 ppm) have no osteoclast activity, while higher Zn^{2+} amounts increased in the number of TRAP-positive cells⁴¹. In the present study, zinc concentration was up to 0.145 ppm on the first day, and the total zinc ion was released from Zn/BMP-2 dual-effect coating at 0.575 ppm (ug/ml) after 28 days (Figure 10), which is much lower than 6.5 ppm. Furthermore, zinc ions at the concentration of 0.145 ppm could elevate the proliferation of both macrophage and BM-MSCs (Figure 2), activate the macrophage phenotype switch from M1 to M2 (Figure 3) and enhance the osteogenic differentiation (Figure 4).

According to your suggestions, we have added the relevant explanation in the section of results and discussion.

Changed in the revised manuscript (Page 15 and 16, highlighted):

2. Results and discussion

2.3. Macrophage Phenotypic Switching *In vitro* (Page 15 Line 308-316, highlighted)

Regarding bioactive ions, zinc, an essential trace element for some key enzymes and transcription factors, is considered to be indispensable for the development of the adaptive immune system^{58,59}. A clinical study reported that patients with inflammatory diseases (e.g., rheumatoid arthritis, RA) had low serum levels of zinc and corresponding increased levels of pro-inflammatory TNF- α ; the process could be reversed by the supplementation of zinc⁶⁰. Besides, some previous reports have already suggested that zinc exerts modulatory effects on macrophage phenotype from M1 to M2, inducing anti-inflammatory responses and inhibiting the pro-inflammatory^{27, 28}. Therefore, the addition of zinc has a potential positive effect on osteoimmunomodulation.

Refences:

27. Zhao DW, *et al.* Strontium-zinc phosphate chemical conversion coating improves the osseointegration of titanium implants by regulating macrophage polarization. *Chem Eng J* **408**, 127362 (2021).
28. Liu W, *et al.* Zinc-modified sulfonated polyetheretherketone surface with immunomodulatory function for guiding cell fate and bone regeneration. *Adv Sci* **5**, 1800749 (2018).
58. Bonaventura P, Benedetti G, Albarède F, Miossec P. Zinc and its role in immunity and inflammation. *Autoimmun Rev* **14**, 277-285 (2015).
59. Maares M, Haase H. Zinc and immunity: An essential interrelation. *Arch Biochem Biophys* **611**, 58-65 (2016).
60. Chen Z, *et al.* Osteoimmunomodulation for the development of advanced bone biomaterials. *Mater Today* **19**, 304-321 (2016).

2.3. Macrophage Phenotypic Switching *In vitro* (Page 16 Line 331-336, highlighted)

Although zinc plays an important and beneficial role in immune functions, its effects depend on the concentration of zinc⁵⁸. A study reported that 1.25×10^{-6} M (0.08 ppm) zinc was enough to inhibit the expression of pro-inflammatory cytokines; however, a low concentration of Zn²⁺ (less than 6.5 ppm) had no osteoclast activity, while higher Zn²⁺ amounts increased osteoclast activity⁶⁵. In the present study, zinc concentration reached 0.145 ppm on the first day, and the total zinc ion released from Zn/BMP-2 dual-effect coating was 0.575 ppm after 28 days (Figure 10), which is much lower than 6.5 ppm. Furthermore, zinc ions at the concentration of 0.145 ppm could elevate the proliferation of both macrophages and BM-MSCs (Figure 2), activate the macrophage phenotypic switch from M1 to M2 (Figure 3) and enhance the osteogenic differentiation (Figure 4).

References:

58. Bonaventura P, Benedetti G, Albarède F, Miossec P. Zinc and its role in immunity and inflammation. *Autoimmun Rev* **14**, 277-285 (2015).
65. Holloway WR, Collier FM, Herbst RE, Hodge JM, Nicholson GC. Osteoblast-mediated effects of zinc on isolated rat osteoclasts: Inhibition of bone resorption and enhancement of osteoclast number. *Bone* **19**, 137-142 (1996).

Comment 4: Were the peptides ((DOPA)₆-PEG₅-DBCO and (2-Azido)-PEG₅-BMP-2) stably coated on Ti surface? Although the chemical analysis including EDS and AFM demonstrated the successful coating initially, the long term evaluation is needed to prove the stability of the peptides coating (release or detachment).

Reply: Thank you for the comment and reminding. Per the Reviewer's suggestion, we further determined the durability of surface modified BMP-2 peptide by incubation of the Zn/BMP-2 substrate in Dulbecco's modified Eagle's medium (DMEM, 37 °C) for 2 weeks.

As shown in **Figure 1N**, the intensity of N 1s signal in XPS showed a slight decrease of less than 15 %. In addition, the durability of the coated clickable peptide labelled by a FITC probe was further checked to confirm the bioactivity. Despite of incubation in DMEM/F12 for 2 weeks, the intensity of fluorescence on the clickable peptide-modified TiO₂ surface (Zn/BMP-2 group) did not show significant reduction (**Figure S6**). These results indicate that BMP-2 peptide can be grafted stably onto the TiO₂ surface.

Figure 1. (N) Changes of N 1s signal in the XPS spectrum of the Zn/BMP-2 surface after incubated in DMEM for 2 weeks.

Figure S6. (A) Images of FITC-(2-Azido)-PEG₅-BMP-2 and Zn²⁺ co-modified surfaces:(green: (2-Azido)-PEG₅-BMP-2) and (B) Intensity profile with regions of interest.

We have added some discussion in the section of results and discussion in the revised manuscript.

Changed in the revised manuscript (**Page 11 Line 231-237, highlighted**):

2. Results and discussion

2.1. Mussel-Derived Peptide Synthesis and Surface Modification

Then, the durability of BMP-2 on the surface was evaluated by incubating the Zn/BMP-2 substrate in Dulbecco's modified Eagle's medium/F12 (DMEM/F12, 37 °C) for 2 weeks. As shown in **Figure 1N**, the intensity of N 1s signal in XPS showed a slight decrease of less than 15 %. In addition, the durability of the coated clickable peptide labelled by a FITC probe was further checked to confirm the bioactivity. Despite of incubation in DMEM/F12 for 2 weeks, the intensity of fluorescence on the clickable peptide-modified TiO₂ surface (Zn/BMP-2 group) did not show significant reduction (**Figure S6**). Thus, it could be concluded that immobilized BMP-2 peptide is highly stable, probably due to the covalent bonding between DBCO group and azido group via bioorthogonal click chemistry

Figure S6. (A) Images of FITC-(2-Azido)-PEG₅-BMP-2 and Zn²⁺ co-modified surfaces:(green: (2-Azido)-PEG₅-BMP-2) and (B) Intensity profile with regions of interest.

Comment 5: In page 10, line 6, XPS results are not sufficient to explain changes in amount of immobilized BMP-2 because DOPA also showed similar N1s peak as shown in figure 1K.

Release profile of BMP-2 would be more appropriate for this purpose.

Reply: Thank you for the comment and reminding. In agreement with the Reviewer's opinion, XPS results are not sufficient to explain changes in amount of immobilized BMP-2. In this study, the purpose of XPS detection in this section is not to quantify the immobilized BMP-2, but to analyze the chemical composition of the surface. For the release profile of BMP-2 you mentioned, we checked the durability of BMP-2 which was labelled by a FITC probe and traced the fluorescence intensity. During the process, we found that BMP-2 was relatively stable with low release (**Figure S6**). While density of BMP-2 peptide on titanium surface was measured by QCM for the purpose of quantitative analysis (**Figure S4**). These results jointly indicate the BMP-2 peptide could be grafted on the TiO₂ surface. What's more, QCM result showed the maximal grafting amount for (2-Azido)-PEG₅-BMP-2 was 140 ng/cm².

Figure S6. (A) Images of FITC-(2-Azido)-PEG₅-BMP-2 and Zn²⁺ co-modified surfaces:(green: (2-Azido)-PEG₅-BMP-2) and (B) Intensity profile with regions of interest.

Figure S4. (A) Real-time monitoring of the binding of $(DOPA)_6$ -PEG₅-DBCO on a TiO₂-coated chip determined by QCM. (B) $(2\text{-Azido})\text{-PEG}_5\text{-BMP-2}$ co-grafting process on the $(DOPA)_6$ -PEG₅-DBCO-bound chips.

Comment 6: What is ‘Zn concentration’ in Figure 10? Additionally, explanation about different y-axis for each line should be mentioned.

Reply: Thank you for the comment and reminding. In the original Figure 10, Zn concentration is non-accumulative Zn²⁺ release amount. For a better understanding, we remodified the Figure 10 as shown in the following.

Figure 10. Zn²⁺ release profiles of the Zn/BMP-2 surface in PBS solution. Red (left) and blue (right) represent the non-accumulative and accumulative Zn²⁺ release, respectively.

Comment 7: In 2.2., further quantitative data such as DNA assay or cell counting should be presented to confirm reduction in dead cells from surface modified groups, which was only explained via live/dead staining.

Reply: Thank you for the comment and reminding. We have quantitated cell counting using flow cytometry (LSRFortessa™ X-20, BD, USA). And the results were as follows:

To further quantify the cell viability, RAW264.7 and BM-MSCs were stained by Annexin

V/Propidium iodide (PI) staining and analyzed with the flow cytometry (LSRFortessa™ X-20, BD, USA). These results showed the mean living cell (Annexin V-, PI-) percentage on Zn/BMP-2 co-modified surface was up to 94% (RAW264.7) and 97.9% (BM-MSCs), respectively (Figure S9).

Figure S9. Apoptotic cell death was analyzed by flow cytometry on 24 hours. (A) Annexin V-FITC and PI staining patterns are shown as dot plots in quadrants. Annexin-/PI- were accounted for living cells; (B-C) Quantitative analysis of living cells (RAW264.7 and BM-MSCs) is plotted as bar graphs (n=3,

independent samples. Data are reported as mean ± SD and analyzed with a one-way ANOVA with a Tukey's post hoc test).

We have added some details in the revised manuscript.

Changed in the revised manuscript (Page 32 and Page 11-12, highlighted):

4. Experimental Section

Cytocompatibility (Page 32 Line 669-672, highlighted)

A Live/Dead cell staining kit (Yeasen, China) consisting of Calcein-AM (green fluorescence) and Propidium iodide (PI, red fluorescence) was used to assess the cell viability. The fluorescent images were acquired by a fluorescence microscope (Nikon, Japan). **To further investigate cell viability, cells were collected and incubated with Annexin V-APC and PI. After incubation, binding buffer was added and the cells were analyzed by flow cytometry. All data were analyzed using FlowJo™ software.**

Results and discussion (Page 11 Line 26-29, Page 12 Line 260-264, highlighted)

2.2. Surface Cytocompatibility *In vitro*

To further quantify the cell viability, RAW264.7 and BM-MSCs were stained by Annexin V/Propidium iodide (PI) staining and analyzed with the flow cytometry (LSRFortessa™ X-20, BD, USA). These results showed the mean living cell (Annexin V-, PI-) percentage on Zn/BMP-2 co-modified surface was up to 94% (RAW264.7) and 97.9% (BM-MSCs), respectively (Figure S9).

Comment 8: During bone regeneration process including immunomodulation, tissue formation, and remodeling, the dose of adapted biomolecules is an important parameter to enhance the natural-like tissue reconstruction. Overdose of inductive proteins or peptides often caused the abnormally and bulky new bone formation. So what is the coating density of Zn²⁺ and (2-Azido)-PEG₅-BMP-2) on the Ti surface? The author would provide quantified amount of each. What is your comment on the concentration-dependent immunomodulation or osteogenesis?

Reply: Thank you for the comment and reminding. The amount of Zn²⁺ that adhered on the surface was measured by ICP- AES. The measured value was divided by the total amount and size of each sheet (1.76 cm²) to calculate ug/cm². The total amount of zinc is about the accumulative Zn²⁺ release because the release is steady-state (0.04 ppm) after 3 weeks, which is neglect. Therefore, the average grafting density of zinc could be estimated by using the

following equation: $\rho=C*V/S$, where, C is the accumulative Zn^{2+} release; V is total volume of solution for incubating the Zn/BMP-2 sample (10 mL). ρ is the average density of the grafted peptide film. S is the area of the TiO_2 -coated quartz substrates. Therefore, the average grafting density of zinc is about 3.24 ug/cm^2 .

In addition, we measured the density of the peptide, which grafted on the TiO_2 surface, by quartz-crystal microbalance QCM (Q-sense AB, Sweden). The results shown in the following **FigureS4**. It showed that $(DOPA)_6$ -PEG₅-DBCO could steady bind onto the QCM chips and the maximal grafting density was 489 ng/cm^2 , indicating the high efficiency and spontaneous adhesion onto TiO_2 -deposited quartz substrate surface. Then, the DBCO-modified TiO_2 substrates were incubated with azido-capped BMP-2-derived peptide for bioorthogonal conjugation. According to QCM analysis, the clickable reaction started in a few minutes, and the maximal grafting amount for $(2\text{-Azido})\text{-PEG}_5\text{-BMP-2}$ was 140 ng/cm^2 .

Figure S4. (A) Real-time monitoring of the binding of $(DOPA)_6$ -PEG₅-DBCO on a TiO_2 -coated chip determined by QCM. (B) $(2\text{-Azido})\text{-PEG}_5\text{-BMP-2}$ co-grafting process on the $(DOPA)_6$ -PEG₅-DBCO-bound chips.

My opinions on the concentration-dependent immunomodulation or osteogenesis were as follows:

Zinc is a nutritionally essential trace element for some key enzymes and transcription factors and is well documented to be indispensable for the development of the adaptive immune system^{46,47}. A clinical study reported that patients with inflammatory diseases (e.g., rheumatoid arthritis, RA) have been found to be low serum levels of zinc and a corresponding increased TNF- α production; the process can be reversed by the supplementation of zinc⁵⁸. Besides, some previous reports have already suggested that zinc exerts modulatory effects on macrophage phenotype from M1 to M2, inducing anti-inflammatory responses and inhibiting the pro-inflammatory. However, a high dose of Zn^{2+} could be detrimental to osteoblast differentiation, disruption of mineralization process, even may evoke immune diseases. A

study identified that a low concentration of Zn^{2+} (less than 6.5 ppm) have no osteoclast activity, while higher Zn^{2+} amounts increased in the number of TRAP-positive cells⁴¹. However, 1.25×10^{-6} M (0.08 ppm) zinc is enough to decrease the expression of pro-inflammatory cytokines. In the present study, zinc concentration was up to 0.145 ppm on the first day, and the total zinc ion was released from Zn/BMP-2 dual-effect coating at 0.575 ppm (ug/ml) after 28 days (Figure 1O), which is much lower than 6.5 ppm. According to *in vitro* results, furthermore, these results indicated that zinc ions at the concentration of 0.145 ppm could elevate the proliferation of both macrophage and BM-MSCs (Figure 2), while activate the macrophage phenotype switch from M1 to M2 (Figure 3), enhancing the osteogenic differentiation (Figure 4).

BMP-2 is a potent inducer of osteogenesis and could be demonstrated an impressive ability to induce new bone formation *in vivo*⁴⁹. Strategies often utilize soluble BMP-2 either physically adsorbed, covalently linked or encapsulated with biomaterials/implant to confer osteoinduction properties. However, depending on these strategies, there is two main issues, including impairing bioactivity and diffusing rapidly out of the biomaterials³⁵. What's more, high doses of BMP-2 have led to serious complications, such as ectopic bone formation, immunological response, and even tumorigenesis³⁶. These limitations may be overcome by using the osteogenic BMP-2 peptide. Short chain BMP-2 peptides have been developed to mimic the activity of BMP-2 proteins by binding to their cell receptors³⁶. Further studies demonstrated that incorporating the BMP-2 peptide in biomaterials can induce *in vitro* osteogenic differentiation and *in vivo* bone regeneration³⁵. In the present study, the maximal grafting amount of (2-Azido)-PEG₅-BMP-2 was 140 ng/cm² according to QCM analysis (Figure S4). What's more, we have demonstrated (2-Azido)-PEG₅-BMP-2 could be grafted stably onto TiO₂ surface (Figure S6).

According to *in vitro* and *in vivo* results, (2-Azido)-PEG₅-BMP-2 in our system could exert superior osteoinductive effect without causing other side effects.

Changed in the revised manuscript (Page 28 and 9, highlighted):

Method:

Characterizations (Page 30 Line 626-629, highlighted)

The chemical composition of different samples was characterized by energy-dispersive X-ray spectrometry (EDS, Sirion 200, FEI) and X-ray photoelectron spectroscopy (XPS, AXIS Ultra DLD, Japan). Surface wettability of different samples was analyzed by a contact

angle instrument Theta Lite (Biolin scientific, Finland). QCM-D (Q-sense AB, Sweden) was used to determine the mass of peptides modified on the TiO₂ surfaces. The concentrations of (DOPA)₆-PEG₅-DBCO and (2-Azido)-PEG₅-BMP-2 used for QCM-D analysis were the same as those used for the peptide coating and bio-orthogonal co-grafting process.

Results and discussion: (Page 9 Line 191-203, highlighted):

2.1. Mussel-Derived Peptide Synthesis and Surface Modification

Finally, (2-Azido)-PEG₅-BMP-2 was conjugated through bioorthogonal click chemistry to prepare a Zn²⁺ and BMP-2 peptide co-modified surface (noted as Zn/BMP-2). A BMP-2-modified surface without loading Zn²⁺ (noted as BMP-2) was also prepared as a control. Quartz-crystal microbalance (QCM) was used to monitor the peptide grafting densities of (DOPA)₆-PEG₅-DBCO and (2-Azido)-PEG₅-BMP-2⁵¹. As shown in **Figure S4A**, (DOPA)₆-PEG₅-DBCO could be steady bound onto the QCM chips and the maximal grafting density was about 489 ng/cm², indicating the high efficiency and spontaneous adhesion onto TiO₂-deposited quartz substrate surface. The grafting density in our study was higher than that of Pan's work (363 ng/cm²)¹, mainly due to the improved binding affinity of mussel-adhesion peptide resulting from the increased number of catechol groups. Then, the DBCO-modified TiO₂ substrates were incubated with azido-capped BMP-2-derived peptides for bioorthogonal conjugation (**Figure S4B**). The click reaction started in a few minutes, and the maximal grafting density for (2-Azido)-PEG₅-BMP-2 was 140 ng/cm², which was comparable to the results in previous reports on the immobilization of BMP-2 on chitosan-grafted titanium surfaces (50 ng/cm²)⁵² and the polydopamine-coated nanofibers (124 ng/cm²)⁵³, respectively.

Figure S4 (A) Real-time monitoring of the binding of (DOPA)₆-PEG₅-DBCO on a TiO₂-coated chip determined by QCM. (B) (2-Azido)-PEG₅-BMP-2 co-grafting process on the (DOPA)₆-PEG₅-DBCO-bound chips.

References:

1. Pan G, et al. Biomimetic design of mussel-derived bioactive peptides for dual-functionalization of titanium-based biomaterials. *J Am Chem Soc* 138, 15078-15086 (2016).
51. Ahmad N, et al. Peptide Cross-Linked Poly(2-oxazoline) as a Sensor Material for the Detection of Proteases with a Quartz Crystal Microbalance. *Biomacromolecules* 20, 2506-2514 (2019).
52. Shi Z, Neoh KG, Kang ET, Poh C, Wang W. Titanium with surface-grafted dextran and immobilized bone morphogenetic protein-2 for inhibition of bacterial adhesion and enhancement of osteoblast functions. *Tissue Eng Part A* 15, 417-426 (2009).
53. Cho H-j, et al. Effective Immobilization of BMP-2 Mediated by Polydopamine Coating on Biodegradable Nanofibers for Enhanced *in vivo* Bone Formation. *Acs Appl Mater Inter* 6, 11225-11235 (2014).

Comment 9: Why the initial cell adhesion and spreading of BM-MSCs were enhanced in Zn, BMP-2, Zn/BMP-2 groups than TiO₂ group? Was it associated with the Zn²⁺ and BMP-2, or other parameter? The authors should elaborate the increase in cell adhesion.

Reply: Thank you for the comment. Our results indicate that Zn²⁺ and BMP-2 co-modified surface was found beneficial to cell adhesion and spreading of BM-MSCs after one-day cultivation. Possible reasons can be concluded into two aspects. Hydrophilicity and roughness are known to impact the initial attachment of cells to various kinds of surfaces^{50, 51, 52, 53}. Previous works have confirmed that Zn or BMP-2 could improve the hydrophilicity of the bone implants, thus enhancing their biocompatibility^{54, 55} leading to more obvious cell adhesion and spreading. In our study, it could be found that the surface wettability showed significant improved after Zn²⁺ or BMP-2 peptide modification (**Figure 1G and 1H**), probably due to the hydrophilicity of surface chelated Zn²⁺ and the amino acid sequence of BMP-2 peptide.

The changes of surface roughness were also checked by atom force microscope (AFM) in each group (**Figure 1E and 1F**). BMP-2 and Zn/BMP-2 group showed increased roughness after modification which is benefit to cell adhesion and spreading. Taken the two aspects into consideration, there is no wonder that the initial cell adhesion and spreading of BM-MSCs were enhanced in BMP-2, Zn/BMP-2 groups.

Figure 1. (E, F) AFM images of different surfaces and the changes of surface roughness. (G, H) The water contact angles of different surfaces and the quantitative results.

Comment 10: In *in vivo* experiments, the inflammatory response was derived from the surgical process during the implantation of the screws because the host femoral bone was damaged, however, it was different from the immune response derived from LPS, which was a case of infection and demonstrated in *in vitro* analysis. Could Zn²⁺ reduce the inflammatory response in both cases with the same mechanism?

Reply: Thank you for the comment. Many biomaterials performed promisingly in setting of reductionist *in vitro*-models, but some inconsistent results were found due to the complex microenvironment *in vivo*, which is difficult to be simulation *in vitro*. Based on the relevant articles published previously^{56,57}, the addition of LPS into a cell culture medium was observed, aiming to create an inflammatory environment that activates the immune cells and skews them towards a M1 phenotype, simulating complex microenvironment *in vivo*. Therefore, LPS could be used to produce a set of activated inflammatory cells that can serve as a positive control⁵⁸ or to evaluate the anti-inflammatory properties of biomaterials⁵⁹.

In our study, we found that the Zn²⁺-modified surfaces showed the similar results *in vitro* and *in vivo*, modulating the polarization of macrophages from pro-inflammatory M1 phenotype to anti-inflammatory M2 phenotype and increasing the secretion of anti-inflammatory cytokine IL-10. So, we suspect that there may be a similar mechanism in both cases. Though the specific mechanisms that Zn²⁺-containing surfaces modulate macrophage polarization are not a major part of our study, there will be the focus of our future work.

Comment 11: In page 11, line 4, authors' implication about Figure 2D was not convincing because enhanced proliferation does not guarantee differentiation and immunoactivity of stem

cells.

Reply: Thank you for the comment and suggestion. We agreed with your comments and realized the overstatement of the conclusion that enhanced proliferation could not guarantee differentiation and immunoactivity of stem cells. We have rewritten this part as below. It could be found in the part of Results and Discussion 2.2 in the revised article.

Changed in the revised manuscript (Page 12 Line 282-285, highlighted):

Results and discussion

2.2. Surface Cytocompatibility *In vitro*

Interestingly, the Zn²⁺-containing surfaces (i.e., the groups of Zn and Zn/BMP-2) elicited the fastest proliferation of RAW 264.7 cells, while BMP-2 peptide-containing surfaces (i.e., the groups of BMP-2 and Zn/BMP-2) were inclined to enhance BM-MSCs proliferation (**Figure 2D**).

Comment 12: In page 11, line 6, positive effect of Zn²⁺ and BMP-2 peptide could not be explained through Figure 2E because there were no significant differences between groups.

Reply: Thank you for the comment. It was indeed overstated that the Zn²⁺ and BMP-2 peptide co-modified surfaces had positive effect on the growth of both macrophages and BM-MSCs. LDH activity was performed to determine cytotoxicity of the materials. These results, including LDH, live/dead and CCK-8, indicate that the Zn²⁺ and BMP-2 peptide co-modified surface possesses excellent biocompatibility and low cytotoxicity. For a better understanding, we have rewritten this content as follows.

Changed in the revised manuscript (Page 12 Line 264-270, highlighted):

Results and discussion

2.2. Surface Cytocompatibility *In vitro*

In addition, lactic dehydrogenase (LDH) released from cells incubated with Zn²⁺ or BMP-2 modified TiO₂ surfaces were detected to determine the cytotoxicity of the materials. After 24 h incubation, the amount of LDH from these cells was slightly lower than that in the bare TiO₂ group, indicating that there was no cytotoxicity (**Figure 2E**). These implied the poor biocompatibility of the bare TiO₂ surface was significantly improved by Zn²⁺ modification and BMP-2 peptide conjugation, promising further application in bone-implants.

Comment 13: There is no criterion to decide pancake-like cells in this manuscript. It was too

subjective that other objective method such as elongation factor would be proper to this purpose.

Reply: Thank you for the comment. We calculated the elongation factor according to the previous study^{60, 61}. Specifically, the long axis was defined as the longest length of the cell, and the short axis was defined as the length across the nucleus in a direction perpendicular to the long axis. The ratio of the two axes was determined to be the elongation factor. And the relevant result was presented as follows:

Figure S10. (A) Scheme of cell morphology switch and definition of elongation factor; (B) Quantitative data of RAW264.7 elongation factor.

Changed in the revised manuscript (Page 16 Line 331-336, highlighted):

Results and discussion

2.3. Macrophage Phenotypic Switching In vitro

Additionally, to define the shape of macrophages, the degree of cell elongation was further quantified by the ratio of the long axis to the short axis length^{61, 62}. The macrophages treated with Zn²⁺-containing coatings showed a significant higher rate of cellular elongation than those on the bare TiO₂ surface (control group) (**Figure S10**). Together, these data suggested that Zn²⁺-containing surfaces could influence the macrophages morphology and might have an impact on their macrophage phenotypic conversion.

References:

61. McWhorter FY, Wang T, Nguyen P, Chung T, Liu WF. Modulation of macrophage phenotype by cell shape. *Proceedings of the National Academy of Sciences* **110**, 17253-17258 (2013).

62. Kang H, *et al.* Immunoregulation of macrophages by dynamic ligand presentation via ligand–cation coordination. *Nat Commun* **10**, 1696 (2019).

Comment 14: In figure 5A, structure of bone was hard to distinguish that other staining method to show mature bone tissue structure, i.e. Goldner's trichrome staining would be appropriate for this study.

Reply: Thank you for the comment and reminding. To better show mature bone tissue structure, we have performed the Goldner's trichrome staining in our study. The relevant results are presented as follows.

Figure S13. Goldner's trichrome staining for the tissue around bone implants.

And We have added this part of description below in the section of discussion.

Changed in the revised manuscript (Page 22 Line 465-468, highlighted):

Results and discussion

2.5. Macrophage Phenotypic Switching *In vivo*

Goldner's trichrome analysis highlighted the calcified bone (green) significantly increased at the implantation site of Zn/BMP-2 co-modified group in comparison with others, indicating a higher extent of integration and a larger amount of newborn trabecular structures adjacent to the implant (**Figure S13**).

Comment 15: All the genes written in this manuscript should follow general nomenclature.

Reply: Many thanks for the reviewer's carefulness. We have corrected all our these in whole manuscript.

Comment 16: As a minor comment, the forward primer information of Runx-2 and CCR7 in Table S2 was duplicated.

Reply: Thank you for your comment and we are sorry for the mistake. We have corrected

it in Table S2.

Changed in the revised supplementary information (Page 18, highlighted):

Table S2. Primers used in the RT-PCR of BM-MSCs and BMMs cells.

Cell	Gene	Primers Sequence (5'-3')
BM-MSCs	Alp	F: ATGCTCAGGACAGGATCAAA R: CGGGACATAAGCGAGTTTCT
	Colla1	F: AGCTCGATACACAATGGCCT R: CCTATGACTTCTGCGTCTGG
	Runx2	F: ATCATTCAGTGACACCACCA R: GTAGGGGCTAAAGGCAAAAG
	Opn	F: GAACATGAAATGCTTCTTTCTCAG R: TCCATGAAGCCACAACTAAACTA
	β-actin	F: CCTCTATGACAACACAGT R: AGCCACCAATCCACACAG
RAW264.7	Tnf-α	F: GTTCCCAAATGGCCTCCC R: GTGCTCCTCACCCACACCG
	Il10	F: CCCTTTGCTATGGTGTCTCT R: GTGGCCAGTTTGTATTAT
	Cd206	F: TACTTGGACGGATAGATGGAGG R: CATAGAAAGGAATCCACGCAGT
	Ccr7	F: GGTGGCTCTCCTTGTCATTTTC R: AGGTTGAGCAGGTAGGTATCCG
	Vegf	F: AGGAGTCCCCGACGAGATAGA R: CACATCTGCTGTGCTGTAGGAA
	Bmp-2	F: AACGAGAAAAGCGTCAAGCC R: AGGTGCCACGATCCAGTCAT
	β-actin	F: GTGACGTTGACATCCGTAAAGA R: GTAACAGTCCGCCTAGAAGCAC

Comment 17: In figure legend for Figure 2A, typo in living/dead staining.

Reply: Many thanks for the reviewer's carefulness. We have corrected it in the legend of the Figure 2A.

Changed in the revised manuscript (Page 14, Figure 2 Legend, Line 293, highlighted):

Figure 2. (A) **Live/Dead staining** of BM-MSCs and RAW264.7 on the bare and modified TiO₂ surface (DBCO-TiO₂, Zn, BMP-2 and Zn/BMP-2).

Comment 18: In figure legend for Figure 6D, Van Gieson staining should be changed to Van Gieson

Reply: We have corrected it in the legend of the Figure 6D.

Changed in the revised manuscript (Page 26, Figure 6 Legend, Line 5777, highlighted):

Figure 6. (D) quantitative staining analysis; **Van Gieson** and bone implant contact (BIC).

We greatly thank all the reviewers' valuable comments. We hope that the revised manuscript will prove to be acceptable for publication in *Nature Communications*.

Sincerely,

Wenguo Cui, Ph.D./ Prof.

Regenerative Biomaterials Lab– Group Leader

Section Editor: BMC Biomedical Engineering (Spring Nature)

Editorial Board: Materials Science and Engineering: C (Elsevier)

Editorial Board: Mater Today Adv (Elsevier)

Editorial Board: VIEW (Wiley)

Department of Orthopaedics,

Shanghai Key Laboratory for Prevention and Treatment of Bone and Joint Diseases,

Shanghai Institute of Traumatology and Orthopaedics,

Ruijin Hospital,

Shanghai Jiao Tong University School of Medicine,

197 Ruijin 2nd Road, Shanghai 200025, P. R. China.

E-mail: wgcui80@hotmail.com; wgcui@sjtu.edu.cn

<https://scholar.google.com/citations?user=VXkkbqvIMgC&hl=en>

References:

1. Ren X, *et al.* Surface modification and endothelialization of biomaterials as potential scaffolds for vascular tissue engineering applications. *Chem Soc Rev* **44**, 5680–5742 (2015).
2. Yu H, *et al.* Nitric oxide-generating compound and bio-clickable peptide mimic for synergistically tailoring surface anti-thrombogenic and anti-microbial dual-functions. *Bioact Mater* **6**, 1618–1627 (2021).
3. Chen X, Gao Y, Wang Y, Pan G. Mussel-inspired peptide mimicking: An emerging strategy for surface bioengineering of medical implants. *Smart Materials in Medicine* **2**, 26–37 (2021).
4. Liu W, *et al.* Zinc-modified sulfonated polyetheretherketone surface with immunomodulatory function for guiding cell fate and bone regeneration. *Adv Sci* **5**, 1800749 (2018).
5. Shao S-y, *et al.* A titanium surface modified with zinc-containing nanowires: Enhancing biocompatibility and antibacterial property in vitro. *Applied Surface Science* **515**, 146107 (2020).
6. Que M, *et al.* Effects of Zn²⁺ ion doping on hybrid perovskite crystallization and photovoltaic performance of solar cells. *Chem Phys* **517**, 80–84 (2019).
7. Ding YH, Floren M, Tan W. Mussel-inspired polydopamine for bio-surface functionalization. *Biosurf Biotribol* **2**, 121–136 (2016).
8. Xiao Y, *et al.* A Versatile Surface Bioengineering Strategy Based on Mussel-Inspired and Bioclickable Peptide Mimic. *Research-China* **2020**, 7236946 (2020).
9. Loi F, *et al.* The effects of immunomodulation by macrophage subsets on osteogenesis in vitro. *Stem Cell Res Ther* **7**, 15 (2016).
10. Bai X, *et al.* Sequential macrophage transition facilitates endogenous bone regeneration induced by Zn-doped porous microcrystalline bioactive glass. *J Mater Chem B* **9**, 2885–2898 (2021).
11. Nohe A, Hassel S, Ehrlich M, Neubauer F, Knaus P. The Mode of Bone Morphogenetic Protein (BMP) Receptor Oligomerization Determines Different BMP-2 Signaling Pathways. *J Biol Chem* **277**, 5330–5338 (2002).
12. Yu Y, Chen R, Yuan Y, Wang J, Liu C. Affinity-selected polysaccharide for rhBMP-2-induced osteogenesis via BMP receptor activation. *Applied Materials Today* **20**, 100681 (2020).
13. von Bülow V, Rink L, Haase H. Zinc-Mediated Inhibition of Cyclic Nucleotide Phosphodiesterase Activity and Expression Suppresses TNF- α and IL-1 β Production in Monocytes by Elevation of Guanosine 3',5'-Cyclic Monophosphate. *The Journal of Immunology* **175**, 4697 (2005).
14. Driessen C, Hirv K, Kirchner H, Rink L. Zinc regulates cytokine induction by superantigens and lipopolysaccharide. *Immunology* **84**, 272–277 (1995).
15. Chen B, *et al.* Zn-Incorporated TiO₂(2) Nanotube Surface Improves Osteogenesis Ability Through Influencing Immunomodulatory Function of Macrophages. *International journal of nanomedicine* **15**, 2095–2118 (2020).
16. Huang X, *et al.* Sustained zinc release in cooperation with CaP scaffold promoted bone regeneration via directing stem cell fate and triggering a pro-healing immune

- stimuli. *Journal of Nanobiotechnology* **19**, 207 (2021).
17. Bai L, *et al.* A multifaceted coating on titanium dictates osteoimmunomodulation and osteo/angio-genesis towards ameliorative osseointegration. *Biomaterials* **162**, 154-169 (2018).
 18. Zhao DW, *et al.* Strontium-zinc phosphate chemical conversion coating improves the osseointegration of titanium implants by regulating macrophage polarization. *Chem Eng J* **408**, 127362 (2021).
 19. He Y, *et al.* Improved osteointegration by SEW2871-encapsulated multilayers on micro-structured titanium via macrophages recruitment and immunomodulation. *Applied Materials Today* **20**, 100673 (2020).
 20. Gao A, *et al.* Tuning the surface immunomodulatory functions of polyetheretherketone for enhanced osseointegration. *Biomaterials* **230**, 119642 (2020).
 21. Luo M-l, *et al.* Macrophages enhance mesenchymal stem cell osteogenesis via down-regulation of reactive oxygen species. *J Dent* **94**, 103297 (2020).
 22. Zhang Y, Böse T, Unger RE, Jansen JA, Kirkpatrick CJ, van den Beucken JJJP. Macrophage type modulates osteogenic differentiation of adipose tissue MSCs. *Cell Tissue Res* **369**, 273-286 (2017).
 23. Wang Z, *et al.* Switching on and off macrophages by a “bridge - burning” coating improves bone - implant integration under osteoporosis. *Adv Funct Mater* **31**, 2007408 (2020).
 24. Chen C, *et al.* Enhanced Osteoinductivity of Demineralized Bone Matrix with Noggin Suppression in Polymer Matrix. *Advanced Biology* **5**, 2000135 (2021).
 25. Murphy MP, *et al.* Articular cartilage regeneration by activated skeletal stem cells. *Nat Med* **26**, 1583-1592 (2020).
 26. Livshits MA, *et al.* Isolation of exosomes by differential centrifugation: Theoretical analysis of a commonly used protocol. *Sci Rep* **5**, 17319 (2015).
 27. Standal T, Borset M, Sundan A. Role of osteopontin in adhesion, migration, cell survival and bone remodeling. *Exp Oncol* **26**, 179-184 (2004).
 28. Xiang G, *et al.* In situ regulation of macrophage polarization to enhance osseointegration under diabetic conditions using injectable silk/sitagliptin gel scaffolds. *Adv Sci* **8**, 2002328 (2021).
 29. Lee SS, *et al.* Bone regeneration with low dose BMP-2 amplified by biomimetic supramolecular nanofibers within collagen scaffolds. *Biomaterials* **34**, 452-459 (2013).
 30. Budiraharjo R, Neoh KG, Kang E-T. Enhancing bioactivity of chitosan film for osteogenesis and wound healing by covalent immobilization of BMP-2 or FGF-2. *J Biomater Sci Polym Ed* **24**, 645-662 (2013).
 31. Chen S, Shi Y, Zhang X, Ma J. Evaluation of BMP-2 and VEGF loaded 3D printed hydroxyapatite composite scaffolds with enhanced osteogenic capacity in vitro and in vivo. *Mater Sci Eng C* **112**, 110893 (2020).
 32. Rahman CV, *et al.* Controlled release of BMP-2 from a sintered polymer scaffold enhances bone repair in a mouse calvarial defect model. *J Tissue Eng Regen Med* **8**, 59-66 (2014).
 33. Zhao X, *et al.* Enhanced Osseointegration of Titanium Implants by Surface Modification with Silicon-doped Titania Nanotubes. *Int J Nanomedicine* **15**, 8583-8594 (2020).

34. Pan G, *et al.* Biomimetic design of mussel-derived bioactive peptides for dual-functionalization of titanium-based biomaterials. *J Am Chem Soc* **138**, 15078–15086 (2016).
35. Weng L, Boda SK, Wang H, Teusink MJ, Shuler FD, Xie J. Novel 3D Hybrid Nanofiber Aerogels Coupled with BMP-2 Peptides for Cranial Bone Regeneration. *Adv Health Mater* **7**, 1701415 (2018).
36. Zhou X, *et al.* BMP-2 Derived Peptide and Dexamethasone Incorporated Mesoporous Silica Nanoparticles for Enhanced Osteogenic Differentiation of Bone Mesenchymal Stem Cells. *Acs Appl Mater Inter* **7**, 15777–15789 (2015).
37. Wang X, Yu Y, Ji L, Geng Z, Wang J, Liu C. Calcium phosphate-based materials regulate osteoclast-mediated osseointegration. *Bioactive Materials* **6**, 4517–4530 (2021).
8. Humbert P, *et al.* Immune Modulation by Transplanted Calcium Phosphate Biomaterials and Human Mesenchymal Stromal Cells in Bone Regeneration. *Front Immunol* **10**, 663 (2019).
39. Pałka K, Pokrowiecki R. Porous Titanium Implants: A Review. *Advanced Engineering Materials* **20**, 1700648 (2018).
40. Kaur M, Singh K. Review on titanium and titanium based alloys as biomaterials for orthopaedic applications. *Materials Science & Engineering C-Materials for Biological Applications* **102**, 844–862 (2019).
41. Holloway WR, Collier FM, Herbst RE, Hodge JM, Nicholson GC. Osteoblast-mediated effects of zinc on isolated rat osteoclasts: Inhibition of bone resorption and enhancement of osteoclast number. *Bone* **19**, 137–142 (1996).
42. Raghuvanshi P. Mesenchymal Stem Cells Derived from Rat Bone Marrow (rBM MSC): Techniques for Isolation, Expansion and Differentiation. (2018).
43. Bayliss LE, *et al.* The effect of patient age at intervention on risk of implant revision after total replacement of the hip or knee: a population-based cohort study. *The Lancet* **389**, 1424–1430 (2017).
44. Ferguson RJ, Palmer AJR, Taylor A, Porter ML, Malchau H, Glyn-Jones S. Hip replacement. *The Lancet* **392**, 1662–1671 (2018).
45. Wu X, Chen S, Ji W, Shi B. The risk factors of early implant failure: A retrospective study of 6113 implants. *Clin Implant Dent Relat Res* **n/a**, (2021).
46. Bonaventura P, Benedetti G, Albarède F, Miossec P. Zinc and its role in immunity and inflammation. *Autoimmunity Reviews* **14**, 277–285 (2015).
47. Maares M, Haase H. Zinc and immunity: An essential interrelation. *Arch Biochem Biophys* **611**, 58–65 (2016).
48. Chen ZT, *et al.* Osteoimmunomodulation for the development of advanced bone biomaterials. *Mater Today* **19**, 304–321 (2016).
49. Niu H, *et al.* Multicellularity-interweaved bone regeneration of BMP-2-loaded scaffold with orchestrated kinetics of resorption and osteogenesis. *Biomaterials* **216**, 119216 (2019).
50. Taubenberger AV, Woodruff MA, Bai H, Muller DJ, Hutmacher DW. The effect of unlocking RGD-motifs in collagen I on pre-osteoblast adhesion and differentiation. *Biomaterials* **31**, 2827–2835 (2010).
51. Markwardt J, *et al.* Experimental study on the behavior of primary human osteoblasts

- on laser-cused pure titanium surfaces. *Journal of Biomedical Materials Research Part A* **102**, 1422–1430 (2014).
52. Ferrari M, Cirisano F, Morán MC. Mammalian Cell Behavior on Hydrophobic Substrates: Influence of Surface Properties. *Colloids and Interfaces* **3**, 48 (2019).
 53. Nouri-Goushki M, *et al.* 3D-Printed Submicron Patterns Reveal the Interrelation between Cell Adhesion, Cell Mechanics, and Osteogenesis. *Acs Appl Mater Inter* **13**, 33767–33781 (2021).
 54. Wu J, *et al.* Micro-porous polyetheretherketone implants decorated with BMP-2 via phosphorylated gelatin coating for enhancing cell adhesion and osteogenic differentiation. *Colloids and Surfaces B: Biointerfaces* **169**, 233–241 (2018).
 55. Gao C, *et al.* Advances in the induction of osteogenesis by zinc surface modification based on titanium alloy substrates for medical implants. *Journal of Alloys and Compounds* **726**, 1072–1084 (2017).
 56. Zhao F, *et al.* Promoting in vivo early angiogenesis with sub-micrometer strontium-contained bioactive microspheres through modulating macrophage phenotypes. *Biomaterials* **178**, 36–47 (2018).
 57. Sadowska JM, Wei F, Guo J, Guillem-Marti J, Ginebra MP, Xiao Y. Effect of nano-structural properties of biomimetic hydroxyapatite on osteoimmunomodulation. *Biomaterials* **181**, 318–332 (2018).
 58. Behera S, *et al.* Hydroxyapatite reinforced inherent RGD containing silk fibroin composite scaffolds: Promising platform for bone tissue engineering. *Nanomedicine: Nanotechnology, Biology and Medicine* **13**, 1745–1759 (2017).
 59. Sadowska JM, *et al.* The effect of biomimetic calcium deficient hydroxyapatite and sintered β -tricalcium phosphate on osteoimmune reaction and osteogenesis. *Acta Biomaterialia* **96**, 605–618 (2019).
 60. McWhorter FY, Wang T, Nguyen P, Chung T, Liu WF. Modulation of macrophage phenotype by cell shape. *Proceedings of the National Academy of Sciences* **110**, 17253–17258 (2013).
 61. Kang H, *et al.* Immunoregulation of macrophages by dynamic ligand presentation via ligand-cation coordination. *Nat Commun* **10**, 1696–1696 (2019).

REVIEWERS' COMMENTS

Reviewer #1 (Remarks to the Author):

The revised manuscript has been improved when the authors have made a substantial efforts to explain their work with supplementary results. However, I am afraid that I am still not convinced by this research work in terms of novelty. Particularly, why dual functional implant surface is needed when either Zn²⁺ or BMP-2 is able to convince osteogenesis? A load of literatures indicated the role of Zn²⁺ on immunosteomodulation. If Zn²⁺ alone works, why BMP-2 is needed? "Immunosteomodulation" means that a substance added is able to modulate the immune microenvironment towards osteogenesis. If Zn²⁺ only exhibits "immunomodulatory" effect, I can understand why the addition of BMP-2 is considered. However, it seems this is not the case. Or is there any design to realise the sequential release of Zn²⁺ and BMP-2 in tissue microenvironment that may enhance the osteogenic effect of BMP-2??

Reviewer #2 (Remarks to the Author):

The authors have address all my concerns and have made significant improvements to the manuscript in this revision. I am satisfied and suggest publication of this manuscript in it's current form, however, I do have some comments and suggestions for the authors to consider before publishing, at their discrection:

Reviewer #2 Comment #1 in the Significance category- I still suggest the authors make small improvements to their text because it is still not clear enough, to readers, what the true novelty and significance of this work is, even after the new changes by the authors. I want to make clear that I am recommending publication in the manuscripts' current form, so the following comments are more of strong suggestions to improve their work, and not requirements that must be addressed before publication:

1. It is STILL unclear to the reader what the importance of this work is. For example, reviewer #1 in their very first comments say this- Zinc and BMP are known stimulators, why should this work be considered novel.

Here is an example of what I would write to address these concerns; please note I am not suggesting the authors use this text, only read and consider if they could clarify these points in their text.

"At present a number of medical devices have used Zinc ion-delivery approaches to improve bone formation by as much as X% in vivo [citations], after Y weeks. BMP-2 delivery devices have likewise increased bone formation (volume) by as much as Z% [citations]. However, no device has yet attempted to combine both of these approaches, and it is clear from prior studies that combining two successful, but distinct, stimulation effectors does not always produce a synergistic, or even additive improvement [citations showing that when two individually beneficial methods are combined there is no added benefit, or worsened outcomes], rather, antagonistic nullification is a very possible outcome. In this study, combining Zn and BMP-2 on a single device increased bone formation by as much as Q%, which is significantly greater than either stimulator alone [citations]. In fact, the most successful studies using similar devices only report half as much new bone growth (30-60%, this is a made up number) as we have produced in this study [citations]. Finally, the reader should note that in this model Zinc has been delivered as a acute/chronically released ion that diffused to act on cells distant from the device, while BMP2 has been fixed to the device chemically, which will prevent known side effects associated with unexpected changes in release

rates [citations showing soft tissue related toxicity of BMP2 when used in spine/cervical area, and lawsuit related articles regarding BMP-2 device related deaths, Medtronic, etc.] and, most importantly, ONLY act upon activated/recruited cells that are in direct surface contact with our implanted device. This two-step approach uses a far reaching modulator of immune cytokine production in the first stage, and very localized activation/differentiation of recruited target cell populations (e.g. osteogenic cells but not immune cells) in the second stage. Our two-stage approach is entirely novel, and designed to capitalize on the kinetics of cell recruitment, where zinc actively influences cell recruitment (i.e. cytokine mediate recruitment), while BMP-2 is surface bound and subsequently only act locally on the recruited cells. No other device or study has utilized, or studied the effects of, this type of approach.”

I do not encourage the authors to use this text, but I do suggest that this description of their work explains things that are unclear to readers, for example Reviewer #1. Reviewer #1, comments #1 suggests the work is not novel and the authors do not explain the novelty. The reviewer is correct. Reviewer #1, comments #2 and #3 show the reviewer was unaware (meaning it was not obvious to someone very skilled in the field, and it will therefore be even LESS obvious to general readers of this journal) that BMP-2 was fixated, chemically, and would not be diffusing away, or acting as a “drug delivery” type agent. This is a reoccurring theme in the reviewer comments- the authors have not clearly explained novelty or significance. This is paramount. In my opinion, the article novelty and significance is:

a. You combined two approaches and they worked additively, if not synergistically- explain why this is significant/novel because many others have tried to combine two or more approaches only to find out the effects are NOT additive.

b. You have real, significant bone growth improvement. How much? Is that a lot compared to other studies using BMP-2 alone? What is the SIGNIFICANCE of your result? To know that the reader must be able to compare your most important result (how much bone volume grew) to other studies and SEE how SIGNIFICANT your results were. (Please ignore the use of caps, I am just trying to emphasize particular words).

c. Your approach is actually novel- you are not just combining two drug/agents, you are combining a diffusible wide acting agent with a fixated locally acting agent, and agents which participate in distinct temporal/kinetic events of the healing process (e.g. Zn modulates which/quantity of cytokines at early stage, which in turn recruit MSC, etc., which in turn encounter BMP-2 at the recruitment site at mid/late stage).

In short, for this comment, I strongly suggest the authors add/revise 1 sentence in the abstract, and either a small section of text in the discussion/conclusion and/or introduction (you want the reader to identify your novelty and significance even BEFORE seeing your results, if possible), stating clearly, at a minimum, the three points above in a, b, and c. I see from the revised text that the authors have made significant changes, so I am not making this a required revision, just a strong suggestion.

Reviewer #2 comment #4, the authors have provided very good response/information, this is exactly what I would want to read, as a journal reader. Can they add this to the text somewhere, maybe discussion, in line with my suggestions above (e.g. “ Zinc ion-delivery approaches to improve bone formation by as much as X% in vivo [citations], after Y weeks. BMP-2 delivery devices have likewise increased bone formation (volume) by as much as Z% [citations]”).

Reviewer #3 (Remarks to the Author):

The authors now successfully addressed the comments raised by reviewers.

Reviewer #1 (Remarks to the Author):

Comment 1:

The revised manuscript has been improved when the authors have made a substantial efforts to explain their work with supplementary results. However, I am afraid that I am still not convinced by this research work in terms of novelty. Particularly, why dual functional implant surface is needed when either Zn^{2+} or BMP-2 is able to convince osteogenesis? A load of literatures indicated the role of Zn^{2+} on immunosteomodulation. If Zn^{2+} alone works, why BMP-2 is needed? "Immunosteomodulation" means that a substance added is able to modulate the immune microenvironment towards osteogenesis. If Zn^{2+} only exhibits "immunomodulatory" effect, I can understand why the addition of BMP-2 is considered. However, it seems this is not the case. Or is there any design to realise the sequential release of Zn^{2+} and BMP-2 in tissue microenvironment that may enhance the osteogenic effect of BMP-2??

Response: Many thanks for the reviewer's critical comments, which are very helpful for us to improve our study. Maybe we had not clearly explained the novelty in our previous response. But we think the novelty and significance can be summarized as follows.

Firstly, it is the novelty in our approach. The two agents act differently and not simply synergistically in this dynamic healing process. The mussel-like surface coating provides a fixated locally acting surface, and widely diffusible Zn modulates cytokines production, providing a favorable immunomodulatory microenvironment at early stage, which in turn recruit MSC, etc. to the fixated acting agent, encountering BMP-2 peptide, improving the osseointegration at bone-to-implant interfaces at later stage. Obviously, the two agents participate in distinct temporal/kinetic events of the healing process.

Secondly, it is difficult to co-grafted zinc ions and bioactive peptides with traditional methods^{1, 2, 3}. In our study, we combined the metal ion (e.g., Zn^{2+}) with bioactive peptide (e.g., BMP-2-derived peptide) using a mussel adhesion-mediated ion coordination and molecular clicking strategy to overcome the shortcomings of the traditional methods. It not only ensures the long-term bioactivity of peptide, but also

combine unique biological activities of the inorganic metal ions with bioactivity peptide to meet the various needs of biological materials.

Additionally, our approach has a wide range of applications. BMP-2-derived peptide or zinc ion could be replaced by other biomimetic peptides (e.g., VEGF, AMP) and metal ions (e.g., Cu^{2+} , Mg^{2+}) to synthesize varieties of multifunctional coatings for satisfying different clinical requirements, such as osteoporosis, diabetes, infection, and poor bone healing.

Given that previous works of surface modification on bone-implants focus either on osteoinduction or immunomodulation, it is the first time to combine the molecule and ion dual-functions in the field of biomodification on bone-implants by mussel adhesion-mediated approach in our work. This two-step approach successfully combines Zn^{2+} with BMP-2 derived peptide, which was used a far for reaching modulator of immune cytokine production in the first stage, providing the favorable immunomodulatory microenvironment and very localized activation/differentiation of recruited target cell populations (e.g., osteogenic cells but not immune cells) in the second stage, improving the osseointegration at bone-to-implant interfaces.

In this case, we have obtained a real, significant bone growth improvement. At present, some biomimetic implants have used Zinc ion-delivery approaches to improve bone formation (BV/TV) by as much as 28.8% in vivo 4, after 4 weeks; BMP-2 delivery devices have likewise increased bone formation by as much as 55% 5 after 6 weeks. In this study, however, we report that a mussel-like surface coating, which was immobilized immune modulating metal ions (Zn^{2+}) and growth factors (BMP-2-derived peptide), increase bone formation by as much as 80% at 8 weeks, which is significantly greater than either stimulator alone ^{4,5}. In fact, the most successful studies using similar implants only report that there is half as much new bone growth (28%-55%) as we have produced in this study ^{4,5,6}.

In this study, BMP-2 peptide has been fixed to the mussel-like surface coating chemically which can hard realize the sequential release of Zn^{2+} and BMP-2 in tissue microenvironment.

References:

1. Ren X, *et al.* Surface modification and endothelialization of biomaterials as potential scaffolds for vascular tissue engineering applications. *Chem Soc Rev* **44**, 5680-5742 (2015).
2. Yu H, *et al.* Nitric oxide-generating compound and bio-clickable peptide mimic for synergistically tailoring surface anti-thrombogenic and anti-microbial dual-functions. *Bioact Mater* **6**, 1618-1627 (2021).
3. Chen X, Gao Y, Wang Y, Pan G. Mussel-inspired peptide mimicking: An emerging strategy for surface bioengineering of medical implants. *Smart Materials in Medicine* **2**, 26-37 (2021).
4. Li Y, *et al.* Enhanced osseointegration and antibacterial action of zinc-loaded titania-nanotube-coated titanium substrates: In vitro and in vivo studies. *J Biomed Mater Res A* **102**, 3939-3950 (2014).
5. Rahman CV, *et al.* Controlled release of BMP-2 from a sintered polymer scaffold enhances bone repair in a mouse calvarial defect model. *J Tissue Eng Regen Med* **8**, 59-66 (2014).
6. Zhao X, Wang T, Qian S, Liu X, Sun J, Li B. Silicon-doped titanium dioxide nanotubes promoted bone formation on titanium implants. *Int J Mol Sci* **17**, 292 (2016).

Reviewer #2 (Remarks to the Author):

Comment 1:

The authors have address all my concerns and have made significant improvements to the manuscript in this revision. I am satisfied and suggest publication of this manuscript in it's current form, however, I do have some comments and suggestions for the authors to consider before publishing, at their discretion:

Response: We really appreciate the reviewer's constructive comments, which are very helpful for improving our study. Accordingly, we have revised our manuscript carefully and the point-by-point responses are provided below.

Comment 2:

Reviewer #2 Comment #1 in the Significance category- I still suggest the authors make small improvements to their text because it is still not clear enough, to readers, what the true novelty and significance of this work is, even after the new changes by the authors. I want to make clear that I am recommending publication in the manuscripts'

current form, so the following comments are more of strong suggestions to improve their work, and not requirements that must be addressed before publication:

1. It is **STILL** unclear to the reader what the importance of this work is. For example, reviewer #1 in their very first comments say this- Zinc and BMP are known stimulators, why should this work be considered novel.

Here is an example of what I would write to address these concerns; please note I am not suggesting the authors use this text, only read and consider if they could clarify these points in their text.

“At present a number of medical devices have used Zinc ion-delivery approaches to improve bone formation by as much as X% in vivo [citations], after Y weeks. BMP-2 delivery devices have likewise increased bone formation (volume) by as much as Z% [citations]. However, no device has yet attempted to combine both of these approaches, and it is clear from prior studies that combining two successful, but distinct, stimulation effectors does not always produce a synergistic, or even additive improvement [citations showing that when two individually beneficial methods are combined there is no added benefit, or worsened outcomes], rather, antagonistic nullification is a very possible outcome. In this study, combining Zn and BMP-2 on a single device increased bone formation by as much as Q%, which is significantly greater than either stimulator alone [citations]. In fact, the most successful studies using similar devices only report half as much new bone growth (30-60%, this is a made up number) as we have produced in this study [citations]. Finally, the reader should note that in this model Zinc has been delivered as a acute/chronically released ion that diffused to act on cells distant from the device, while BMP2 has been fixed to the device chemically, which will prevent known side effects associated with unexpected changes in release rates [citations showing soft tissue related toxicity of BMP2 when used in spine/cervical area, and lawsuit related articles regarding BMP-2 device related deaths, Medtronic, etc.] and, most importantly, **ONLY** act upon activated/recruited cells that are in direct surface contact with our implanted device. This two-step approach uses a far reaching modulator of immune cytokine production in the first stage, and very localized activation/differentiation of recruited target cell populations (e.g. osteogenic cells but

not immune cells) in the second stage. Our two-stage approach is entirely novel, and designed to capitalize on the kinetics of cell recruitment, where zinc actively influences cell recruitment (i.e. cytokine mediate recruitment), while BMP-2 is surface bound and subsequently only act locally on the recruited cells. No other device or study has utilized, or studied the effects of, this type of approach.”

I do not encourage the authors to use this text, but I do suggest that this description of their work explains things that are unclear to readers, for example Reviewer #1. Reviewer #1, comments #1 suggests the work is not novel and the authors do not explain the novelty. The reviewer is correct. Reviewer #1, comments #2 and #3 show the reviewer was unaware (meaning it was not obvious to someone very skilled in the field, and it will therefore be even LESS obvious to general readers of this journal) that BMP-2 was fixated, chemically, and would not be diffusing away, or acting as a “drug delivery” type agent. This is a reoccurring theme in the reviewer comments- the authors have not clearly explained novelty or significance. This is paramount. In my opinion, the article novelty and significance is:

- a. You combined two approaches and they worked additively, if not synergistically- explain why this is significant/novel because many others have tried to combine two or more approaches only to find out the effects are NOT additive.
- b. You have real, significant bone growth improvement. How much? Is that a lot compared to other studies using BMP-2 alone? What is the SIGNIFICANCE of your result? To know that the reader must be able to compare your most important result (how much bone volume grew) to other studies and SEE how SIGNIFICANT your results were. (Please ignore the use of caps, I am just trying to emphasize particular words).
- c. Your approach is actually novel- you are not just combining two drug/agents, you are combining a diffusible wide acting agent with a fixated locally acting agent, and agents which participate in distinct temporal/kinetic events of the healing process (e.g. Zn modulates which/quantity of cytokines at early stage, which in turn recruit MSC, etc., which in turn encounter BMP-2 at the recruitment site at mid/late stage).

In short, for this comment, I strongly suggest the authors add/revise 1 sentence in the

abstract, and either a small section of text in the discussion/conclusion and/or introduction (you want the reader to identify your novelty and significance even BEFORE seeing your results, if possible), stating clearly, at a minimum, the three points above in a, b, and c. I see from the revised text that the authors have made significant changes, so I am not making this a required revision, just a strong suggestion.

Response: Thank you very much for your sincere comment. We totally agree with your suggestions mentioned above. As you required, we have revised one sentence in the abstract. The details have presented as below.

“Immune response and new tissue formation are important aspects of tissue repair. However, only a single aspect was generally considered in previous biomedical interventions, and the synergistic effect is unclear.”

And we have also added a small section of text in the section of discussion.

Changed in the revised manuscript (Page 27, 567-581 and 589-595, highlighted):

Up to now, no relevant study has successfully combined metal ions and growth factors, and the synergistic effect on osteogenic differentiation or osteointegration is unclear. In this study, we reported on the use of a mussel-like surface coating with immobilized immunomodulatory metal ions (e.g., Zn^{2+}) and osteoinductive growth factors (e.g., BMP-2-derived peptide), and we demonstrated the improved *in vivo* outcomes. Zn/BMP-2 co-modified Ti implants increased bone formation by up to 80% at 8 weeks, which was significantly higher than either stimulator alone ^{66, 67}. For example, some biomimetic implants have improved bone formation (BV/TV) by 28.8% *in vivo* ⁶⁶ at 4 weeks via Zn^{2+} delivery; BMP-2 delivery strategies have likewise increased bone formation by 55% ⁶⁷. The possible reasons are as follows. Zinc has been delivered as an acute released ion that diffuses to act on cells (e.g., immune cells) distant from the implant, providing a favorable immunomodulatory microenvironment. Besides, BMP-2 peptide has been fixed to the implants chemically, so side effects associated with unexpected changes in release rates are prevented ^{68, 69}, and BMP-2 peptide will only act upon recruited/activated cells (e.g., osteogenic cell) that contact

implant surfaces directly, therefore improving the osseointegration at bone-to-implant interfaces.

The outlook of this study is that it provides a novel solution in a dual-functional implants with both osteoinductive and immunomodulatory activity for improving osseointegration by a mussel adhesion-mediated ion coordination and molecular clicking strategy to effectively improve mechanical fixation of the bone implants. This strategy involves combining the metal ion (e.g., Zn^{2+}) with bioactive peptide (e.g., BMP-2-derived peptide) to overcome the shortcomings of the traditional methods. It not only ensures the long-term bioactivity of peptide, but also combines unique biological activities of the inorganic metal ions with bioactive peptide to meet the various needs of biological materials. The two-step approach has successfully combined Zn^{2+} with BMP-2 derived peptide, acting as a distant modulator for immune cytokine production in the first stage and achieving local activation/differentiation of recruited target cell clusters (e.g., osteogenic cells instead of immune cells) in the second stage. Given that previous works on bone-implant surface modification focus either on osteoinduction or immunomodulation, our work is the first to combine the dual functions of molecules and ions for biomodification on bone implants via a mussel adhesion-mediated approach. Additionally, BMP-2-derived peptide or zinc ion could be replaced by other biomimetic peptides (e.g., VEGF, AMP) and metal ions (e.g., Cu^{2+} , Mg^{2+}) to synthesize varieties of multifunctional coatings for satisfying different clinical requirements. Although further exploration is still needed to understand the potential mechanisms of osteoimmunomodulation, these results have demonstrated a promising strategy towards bone regeneration and bone-implant osseointegration, which is in all probability utilized in future clinical practice and applied to orthopedic research. Furthermore, our mussel adhesion-mediated and molecular bioclickable strategy provides a favorable osseointegration approach to clinical applications in osteoporosis, diabetes, infection, and poor bone healing. The combination of inorganic metal ions with bioactive peptides and biomaterials will provide more opportunities for developing a new generation of engineering bone implants for orthopedic medicine.

References:

66. Li Y, *et al.* Enhanced osseointegration and antibacterial action of zinc-loaded titania-nanotube-coated titanium substrates: In vitro and in vivo studies. *J Biomed Mater Res A* **102**, 3939-3950 (2014).
67. Rahman CV, *et al.* Controlled release of BMP-2 from a sintered polymer scaffold enhances bone repair in a mouse calvarial defect model. *J Tissue Eng Regen Med* **8**, 59-66 (2014).
68. Shao N, *et al.* Development of Organic/Inorganic Compatible and Sustainably Bioactive Composites for Effective Bone Regeneration. *Biomacromolecules* **19**, 3637-3648 (2018).
69. Zhou X, *et al.* BMP-2 Derived Peptide and Dexamethasone Incorporated Mesoporous Silica Nanoparticles for Enhanced Osteogenic Differentiation of Bone Mesenchymal Stem Cells. *Acs Appl Mater Inter* **7**, 15777-15789 (2015).

Comment 3:

Reviewer #2 comment #4, the authors have provided very good response/information, this is exactly what I would want to read, as a journal reader. Can they add this to the text somewhere, maybe discussion, in line with my suggestions above (e.g. “Zinc ion-delivery approaches to improve bone formation by as much as X% in vivo [citations], after Y weeks. BMP-2 delivery devices have likewise increased bone formation (volume) by as much as Z% [citations]”).

Response: Thank you very much for your sincere comment. We totally agree with your suggestions mentioned above. As you required, we added them in the section of discussion in our revision.

Changed in the revised manuscript (Page 27, 573-575, highlighted):

“Zn/BMP-2 co-modified Ti implants increased bone formation by up to 80% at 8 weeks, which was significantly higher than either stimulator alone^{66, 67}. For example, some biomimetic implants have improved bone formation (BV/TV) by 28.8% *in vivo*⁶⁶ at 4 weeks via Zn²⁺ delivery; BMP-2 delivery strategies have likewise increased bone formation by 55%⁶⁷.”

References:

66. Li Y, *et al.* Enhanced osseointegration and antibacterial action of zinc-loaded titania-nanotube-coated titanium substrates: In vitro and in vivo studies. *J Biomed Mater Res A* **102**, 3939-3950 (2014).

67. Rahman CV, *et al.* Controlled release of BMP-2 from a sintered polymer scaffold enhances bone repair in a mouse calvarial defect model. *J Tissue Eng Regen Med* **8**, 59-66 (2014).

Reviewer #3 (Remarks to the Author):

Comment 1:

The authors now successfully addressed the comments raised by reviewers.

Response: We thank the reviewer for the positive response and all valuable suggestions in previous comments to help us improve our manuscript.

We greatly appreciate all the reviewers' valuable comments. We hope that the revised manuscript will prove to be acceptable for publication in Nature Communications.

Sincerely,

Wenguo Cui, Ph.D./ Prof.
Regenerative Biomaterials Lab– Group Leader
Section Editor: BMC Biomedical Engineering (Spring Nature)
Editorial Board: Materials Science and Engineering: C (Elsevier)
Editorial Board: Mater Today Adv (Elsevier)
Editorial Board: VIEW (Wiley)
Department of Orthopaedics,
Shanghai Key Laboratory for Prevention and Treatment of Bone and Joint Diseases,
Shanghai Institute of Traumatology and Orthopaedics,
Ruijin Hospital,
Shanghai Jiao Tong University School of Medicine,
197 Ruijin 2nd Road, Shanghai 200025, P. R. China.
E-mail: wgcui80@hotmail.com; wgcui@sjtu.edu.cn
<https://scholar.google.com/citations?user=VXkkbqbvIMgC&hl=en>